# The tetraspanin TSPAN5 regulates AMPAR exocytosis by interacting with the AP4 complex

Edoardo Moretto[1,2]*, Federico Miozzo[1], Anna Longatti[1], Caroline Bonnet[3], Francoise Coussen[3], Fanny Jaudon[4,5], Lorenzo A Cingolani[4,6], Maria Passafaro[1,2]*

[1]Institute of Neuroscience, CNR, Vedano al Lambro, Italy; [2]NeuroMI Milan Center for Neuroscience, University of Milano-Bicocca, Milan, Italy; [3]University of Bordeaux, Interdisciplinary Institute for Neuroscience, Bordeaux, France; [4]Department of Life Sciences, University of Trieste, Trieste, Italy; [5]IRCCS Ospedale Policlinico San Martino, Genoa, Italy; [6]Center for Synaptic Neuroscience and Technology (NSYN), Istituto Italiano di Tecnologia (IIT), Genoa, Italy

**Abstract** Intracellular trafficking of AMPA receptors is a tightly regulated process which involves several adaptor proteins, and is crucial for the activity of excitatory synapses both in basal conditions and during synaptic plasticity. We found that, in rat hippocampal neurons, an intracellular pool of the tetraspanin TSPAN5 promotes exocytosis of AMPA receptors without affecting their internalisation. TSPAN5 mediates this function by interacting with the adaptor protein complex AP4 and Stargazin and possibly using recycling endosomes as a delivery route. This work highlights TSPAN5 as a new adaptor regulating AMPA receptor trafficking.

## Editor's evaluation

Glutamate receptor trafficking to synapses plays a crucial role in adjusting the efficacy of information flow in the nervous system. Here the authors show in hippocampal neurons that TSPAN5, a tetraspanin family protein facilitates the delivery of AMPA-type glutamate receptors to the cell surface by interacting with adaptor proteins, AP-4, and stargazin. The work provides evidence supporting a novel mechanism that contributes to the regulation of AMPA receptor traffic and is of interest to the molecular neuroscience and cell biology community.

*For correspondence:
edoardo.moretto@in.cnr.it (EM);
maria.passafaro@in.cnr.it (MP)

Competing interest: The authors declare that no competing interests exist.

## Introduction

Tetraspanins are transmembrane proteins conserved in metazoans that present four transmembrane domains, a small and a large extracellular loop, and intracellular N- and C-termini (*Berditchevski, 2001*). Tetraspanins have the peculiar ability to organise tetraspanin enriched microdomains, membrane domains in which they accumulate (*Charrin et al., 2002*). Tetraspanins have been proposed to function as molecular facilitators by promoting physical proximity between proteins that belong to signalling complexes (*Charrin et al., 2014*). To date, 33 tetraspanins have been described in mammals, with functions in cell-cell adhesion, sperm-egg fusion, cell motility, and proliferation (*Hemler, 2005*). TSPAN5 is part of the C8 subgroup of tetraspanins and was previously shown to regulate the intracellular trafficking and activity of the protease ADAM-10 (*Dornier et al., 2012*; *Eschenbrenner et al., 2020*; *Haining et al., 2012*; *Jouannet et al., 2016*; *Noy et al., 2016*; *Saint-Pol et al., 2017*).

A previous study from our laboratory showed that, in hippocampal pyramidal neurons, TSPAN5 is enriched in dendritic spines and promotes their morphological maturation during synaptogenesis

(*Moretto et al., 2019*). This action is mediated by controlling the surface mobility of the postsynaptic adhesion molecule neuroligin-1 via an interaction occurring on the plasma membrane. A few other studies have investigated the function of tetraspanins at the synapse identifying their role in intracellular trafficking of neurotransmitter receptors in neurons (*Bassani et al., 2012*; *Lee et al., 2017*; *Murru et al., 2018*; *Murru et al., 2017*).

Here, we report a significant increase in the intracellular pool of TSPAN5 in dendritic spines upon neuronal maturation. We demonstrate that in mature neurons TSPAN5 does not participate in dendritic spine maturation but has the main function of controlling surface delivery of α-amino-3-hydroxy-5-methyl-4-isoxazolepropionic acid (AMPA) receptors (AMPARs). AMPARs are tetrameric complexes that mediate most of the fast excitatory transmission in response to the neurotransmitter glutamate in neurons (*Henley and Wilkinson, 2016*). AMPAR surface levels are directly responsible for synapse weakening or strengthening during synaptic plasticity (*Huganir and Nicoll, 2013*); their intracellular trafficking is an extremely complex phenomenon involving several auxiliary proteins (*Anggono and Huganir, 2012*; *Moretto and Passafaro, 2018*) that can be impaired in neurological and neurodevelopmental disorders (*Henley and Wilkinson, 2016*; *Moretto et al., 2016*; *Moretto et al., 2018*).

Importantly, we found that TSPAN5 exerts this function by interacting with AP4, a member of the adaptor protein complex family (*Boehm and Bonifacino, 2001*; *Bonifacino, 2014*; *Robinson and Bonifacino, 2001*), which coding genes are mutated in a syndrome characterised by spastic paraplegia and intellectual disability (*Sanger et al., 2019*). AP4 was previously found to regulate the intracellular trafficking and sorting of several transmembrane proteins in neurons including the stargazin-AMPARs complex (*Matsuda et al., 2008*), the glutamate receptor δ2 (*Yap et al., 2003*), the autophagy regulator ATG9 (*Ivankovic et al., 2020*), and DAGLB, an enzyme involved in the production of the endocannabinoid 2-AG (*Davies et al., 2022*).

Our data identify a novel function of TSPAN5 at the synapse and highlight AMPARs defective trafficking as a possible mechanism for intellectual disability symptoms in the AP4 deficiency syndrome.

## Results

### TSPAN5 intracellular pool interacts with the AP4 complex

In our previous work (*Moretto et al., 2019*), we observed the existence of a substantial intracellular pool of TSPAN5 in mature neurons. We thus performed crosslinking experiments using bis(sulfosuccinimidyl) suberate (BS3) on rat cultured hippocampal neurons. This crosslinker is not permeable to membranes and, as such, if applied to living cells will only crosslink plasma membrane proteins which will appear as high molecular weight bands upon western blot analysis. In contrast, the intracellular pool will not be crosslinked, thereby running at the expected molecular weight. We analysed TSPAN5 levels in BS3 experiments and looked at DIV12 and DIV19. At DIV12 synaptogenesis is prominent in rat cultured neurons (*Chanda et al., 2017*) while at DIV19, primary neurons are considered functionally mature. As shown in *Figure 1A*, TSPAN5 appears as a complex pattern of bands. This is probably due to the association of this protein with cholesterol-rich membranes which makes it poorly soluble in standard lysis buffers (*Charrin et al., 2014*). We previously demonstrated that all these bands are specific (*Moretto et al., 2019*) and thus they were all included in the quantification. We observed an increase in the intracellular levels of TSPAN5 from DIV12 to DIV19, which was not accompanied by a concomitant increase in plasma membrane levels (*Figure 1A*), suggesting that increased intracellular levels of TSPAN5 do not necessarily imply increased delivery of this protein to the plasma membrane. The transferrin receptor showed a more stable distribution across these time points. It needs to be mentioned that it is possible that a fraction of TSPAN5 present on the plasma membrane does not interact with any other protein. This fraction would not be crosslinked and run as a monomer. However, this eventuality is quite unlikely, especially considering that the main function of tetraspanins is exerted by homo- and heterotypic interactions (*Charrin et al., 2014*). To test if the increase in intracellular TSPAN5 could be related to a different function compared to its previously described role in dendritic spines maturation (*Moretto et al., 2019*), we transfected cultured hippocampal neurons at DIV13 with scrambled, Sh-TSPAN5, and rescue (coding for the Sh-TSPAN5 and an ShRNA-resistant form of TSPAN5) constructs. A reduction of TSPAN5 at this time point is unlikely to affect dendritic spine maturation as synaptogenesis is already underway. We analysed dendritic spine density and morphology at DIV21 (*Figure 1B*) and observed that dendritic spine density was reduced, but to a lower extent compared to our previous observations when knocking down TSPAN5 at DIV5 (20% compared to more than 65% reduction, respectively) (*Moretto*

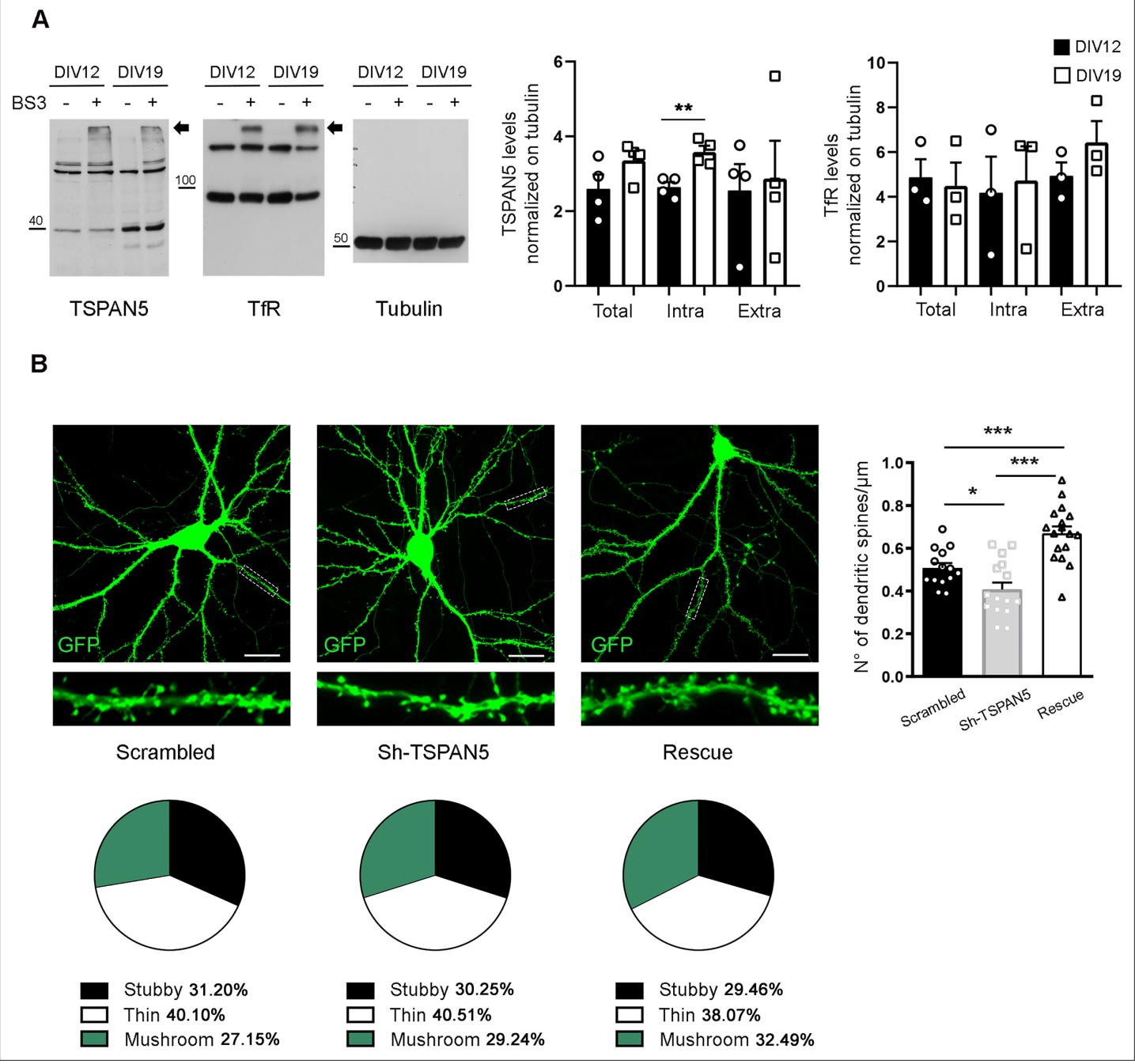

**Figure 1.** TSPAN5 intracellular levels increase with neuronal maturation. (**A**) Bis(sulfosuccinimidyl)suberate (BS3) crosslinking experiment on cultured rat hippocampal neurons at DIV12 and -19 blotted for TSPAN5, transferrin receptor (TfR), and tubulin. Arrows indicate the higher molecular weight bands present in the BS3+lanes that represent the plasma membrane pool of the proteins. Tubulin was used as a loading control; TfR was used as a crosslinking positive control (TSPAN5: total/tubulin: DIV12 2.599±0.38, DIV19 3.357±0.25; intra/tubulin: DIV12 2.643±0.14, DIV19 3.582±0.16; extra/tubulin: DIV12 2.552±0.70, DIV19 2.871±1.01; TfR: total/tubulin: DIV12 4.87±0.81, DIV19 4.48±1.05; intra/tubulin: DIV12 4.18±1.61, DIV19 4.73±1.53; extra/tubulin: DIV12 4.94±0.6, DIV19 6.44±0.95). n = 3–4 independent cultures per condition. Unpaired Student T test. (**B**) Left panels: Confocal images of DIV20 cultured rat hippocampal neurons transfected at DIV13 with either scrambled, Sh-TSPAN5, or rescue (expressing simultaneously both the Sh-TSPAN5 and an ShRNA-resistant form of TSPAN5) constructs, all co-expressing GFP. Scale bar = 20 μm. Inserts (25 μm wide) show higher magnification of the dendrites highlighted in white. Right panel: Quantification of dendritic spine density represented as histograms. Dendritic spine density (no. of dendritic spines/μm: scrambled 0.51±0.02; Sh-TSPAN5 0.41±0.03; rescue 0.67±0.03). Pie charts (bottom panels) show quantification of dendritic spine morphology. Dendritic spine morphology (%: stubby: scrambled 31.20±1.52, Sh-TSPAN5 30.25±2.02, rescue 29.46±1.38; thin: scrambled 40.10±2.45, Sh-TSPAN5 40.51±1.96, rescue 38.07±2.5; mushroom: scrambled 27.15±2.25, Sh-TSPAN5 29.24, rescue 32.49±2.44). n = scrambled, 14; Sh-TSPAN5, 16;

*Figure 1 continued on next page*

*Figure 1 continued*

rescue, 17 neurons. One Way ANOVA, Newman-Kulspost hoc multiplecomparison test. Values represent the mean ± SEM. *=p < 0.05, **=p < 0.01, ***=p < 0.001.

The online version of this article includes the following source data for figure 1:

**Source data 1.** individual data values for the bar graphs and pie charts in panels A and B.

**Source data 2.** Raw images and images with cropped areas highlighted of the blots in panel A.

*et al., 2019*). Even more interestingly, the morphology of dendritic spines was completely unaffected by TSPAN5 knockdown at this time point. In contrast, our previous results had shown a strong reduction (50%) in mature mushroom dendritic spines in favour of less mature thin dendritic spines when TSPAN5 levels were downregulated from DIV5 (*Moretto et al., 2019*). These data support a more prominent role of TSPAN5 for dendritic spine maturation at early stages of development and suggest that TSPAN5 might be involved in other functions at more mature stages. We decided to explore whether the intracellular pool of TSPAN5 could have a role in regulating intracellular trafficking given previous evidence on the role of this and other tetraspanins (*Dornier et al., 2012*; *Haining et al., 2012*; *Jouannet et al., 2016*; *Noy et al., 2016*; *Saint-Pol et al., 2017*; *Bassani et al., 2012*).

The only portions of TSPAN5 exposed to the cytosol are the N- and C-termini (*Berditchevski, 2001*). The C-terminus of other tetraspanins has been shown to regulate the intracellular trafficking of other proteins (*Bassani et al., 2012*). We thus decided to perform a yeast two-hybrid screen using the C-terminal tail of TSPAN5 as a bait. Among the clones identified (the full list is presented in *Figure 2—source data 1*), four of them coded for amino acids 1–102 of the protein AP4σ, one of the subunits of the adaptor protein complex AP4 (*Boehm and Bonifacino, 2001*; *Bonifacino, 2014*; *Robinson and Bonifacino, 2001*). This complex is an obligate tetramer of four different subunits (β, μ, ε, and σ), which readily assemble and are almost undetectable as single subunits (*Hirst et al., 2013*). The AP4 complex has been previously shown to participate in intracellular trafficking of transmembrane proteins, including AMPARs via direct interaction of its epsilon subunit with the auxiliary AMPAR subunit Stargazin (*Matsuda et al., 2008*). We validated the interaction between TSPAN5 and AP4 by GST pulldown on rat brain lysates (cortices and hippocampi) using the C-terminus of TSPAN5 fused to GST (GST-Ct) which precipitated AP4ε (*Figure 2A*), one of the subunits of the AP4 complex. In addition, we confirmed the interaction via co-immunoprecipitation experiments by immunoprecipitating TSPAN5, AP4σ, or AP4ε from rat brain lysates (cortices and hippocampi) and found that all three proteins were associated (*Figure 2B*).

## TSPAN5 can form a complex with GluA2 and Stargazin and localises in recycling endosomes

Given the previously shown interaction of AP4 with Stargazin and AMPARs (*Matsuda et al., 2008*), we performed GST-pulldown experiments to investigate whether TSPAN5 could be part of the same protein complex. By using the C-terminus of TSPAN5 for GST-pulldown experiments on rat brain lysates, we were able to confirm that the C-terminal tail of TSPAN5 is sufficient to precipitate Stargazin, GluA1, and GluA2/3 (*Figure 2C*). The NMDA receptor subunit GluN2A instead was not detected in the precipitates, supporting the specificity of the interaction. Interestingly, GluA2/3 appeared to be pulled down more efficiently than GluA1, suggesting a preferential association between TSPAN5 and GluA2/3-containing AMPARs (*Figure 2D*). To further characterise the interaction, we performed GST pulldown using the C-terminal tail of Stargazin, which was previously identified to be the region responsible for the interaction with AP4 (*Matsuda et al., 2008*). With this experiment, we detected GluA2/3 and TSPAN5 in the precipitate (*Figure 2E*). As a negative control, CD81, another member of the tetraspanin family, was not precipitated by the GST-Ct-Stargazin.

The formation of the complex was also confirmed in immunoprecipitation experiments in Hela cells, which endogenously express AP4, transfected with TSPAN5-GFP, Stargazin-HA, and GluA2 (*Figure 2—figure supplement 1*).

AP4, Stargazin, and AMPARs were previously shown to interact in heterologous cells, which have little to no expression of TSPAN5 (*Matsuda et al., 2008*). This suggests that TSPAN5 is not necessary for the formation of the complex, and we hypothesised that it could be participating in directing these proteins to a specific cellular compartment.

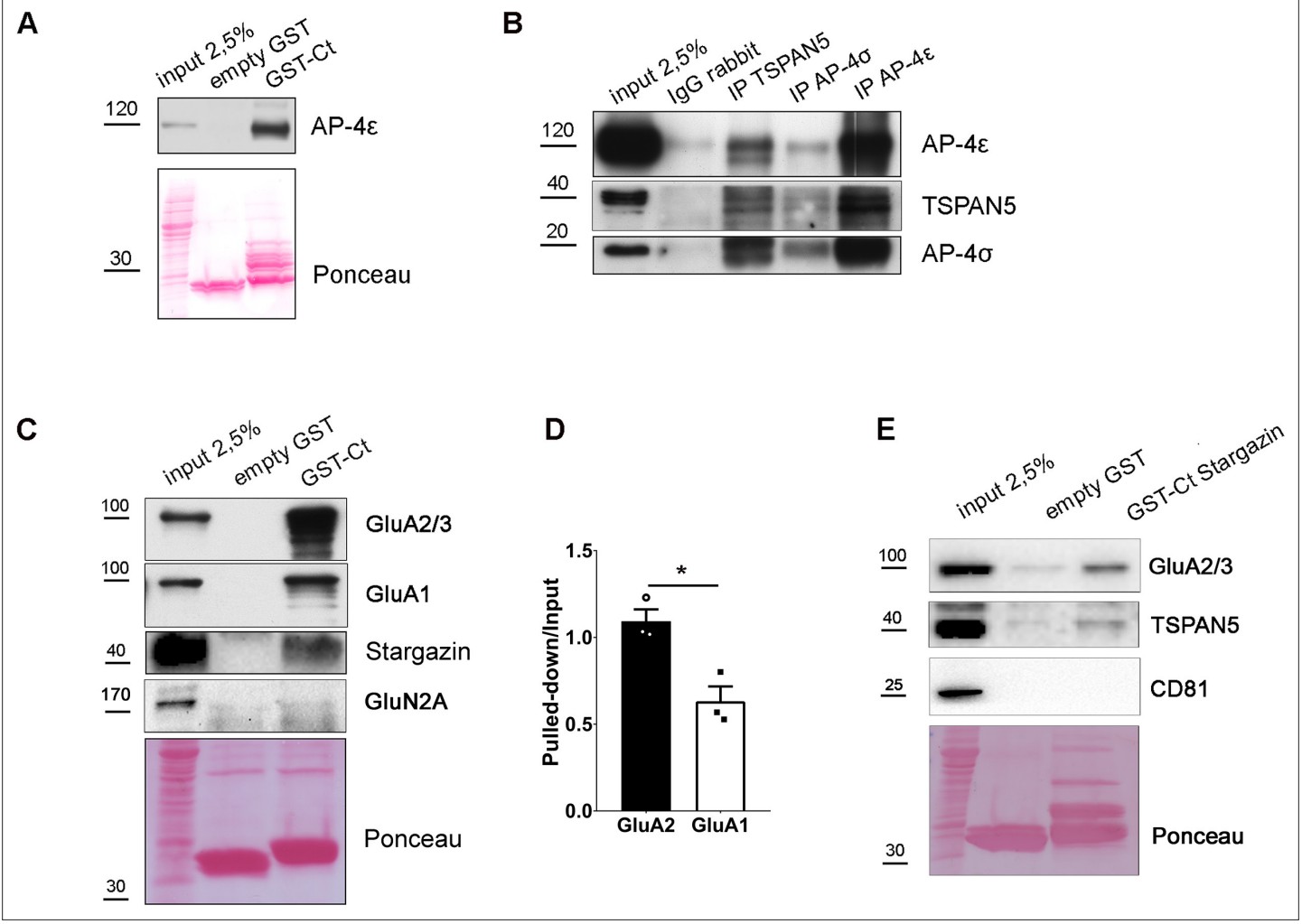

**Figure 2.** TSPAN5 interacts with AP4 and forms a complex with GluA2 and Stargazin. (**A**) GST-pulldown experiment on adult rat hippocampus and cortex lysates using empty GST or GST fused to TSPAN5 C-terminus (GST-Ct). Input: 2.5% of pulldown volume. Blots probed for AP4ε. Red Ponceau shows the GST-bound fragments. (**B**) Co-immunoprecipitation experiment on adult rat hippocampus and cortex lysates. Input: 2.5% of the immunoprecipitated volume. Immunoprecipitation: α-rabbit IgG, α-TSPAN5, α-AP4σ, or α-AP4ε. Blots probed for TSPAN5, AP4σ, and AP4ε. (**C**) GST-pulldown experiments on adult rat hippocampus and cortex lysates using empty-GST or GST fused to the C-terminus of TSPAN5 (GST-Ct). Input: 2.5% of pulldown volume. Blots probed for GluA2/3, GluA1, Stargazin, and NMDAR subunit GluN2A. n = 3 independent experiments. Unpaired Student T test. (**D**) Quantification of experiment in panel C: intensity of the pulldown band for GluA2/3 and GluA1 each normalised on their input (pulldown/input: GluA2, 1.09±0.07; GluA1, 0.63±0.09). (**E**) GST-pulldown experiments on adult rat hippocampus and cortex lysates using empty-GST or GST-fused to the C-terminal of Stargazin (GST-Ct Stargazin). Input: 2.5% pulldown volume. Blots probed for GluA2/3, TSPAN5, and CD81. Values represent the mean ± SEM. *=p < 0.05, **=p < 0.01, ***=p < 0.001.

The online version of this article includes the following source data and figure supplement(s) for figure 2:

**Source data 1.** List of the proteins identified with the yeast two-hybrid screening performed with full-length or C-terminal tail of TSPAN5 and individual data values for the bar graphs in panel D.

**Source data 2.** Raw images and images with cropped areas highlighted of the blots in panels A, B, C, and E.

**Figure supplement 1.** Immunoprecipitation experiment on Hela cells, transfected with TSPAN5-GFP, Stargazin-HA, and GluA2-myc.

**Figure supplement 1—source data 1.** Raw images and images with cropped areas highlighted of the blots presented.

Thus, we next addressed where this association takes place. Given that the intracellular pool of TSPAN5 must reside in intracellular vesicles, we prepared synaptosomes from rat brains (cortices and hippocampi) and loaded their content on a linear sucrose gradient to separate different populations of vesicles (*Rao et al., 2011*; *Figure 3A and B*). We observed that AP4ε, TSPAN5, Stargazin, GluA1, and GluA2/3 are all present at significant levels in the heaviest fractions which are positive for the recycling endosomes

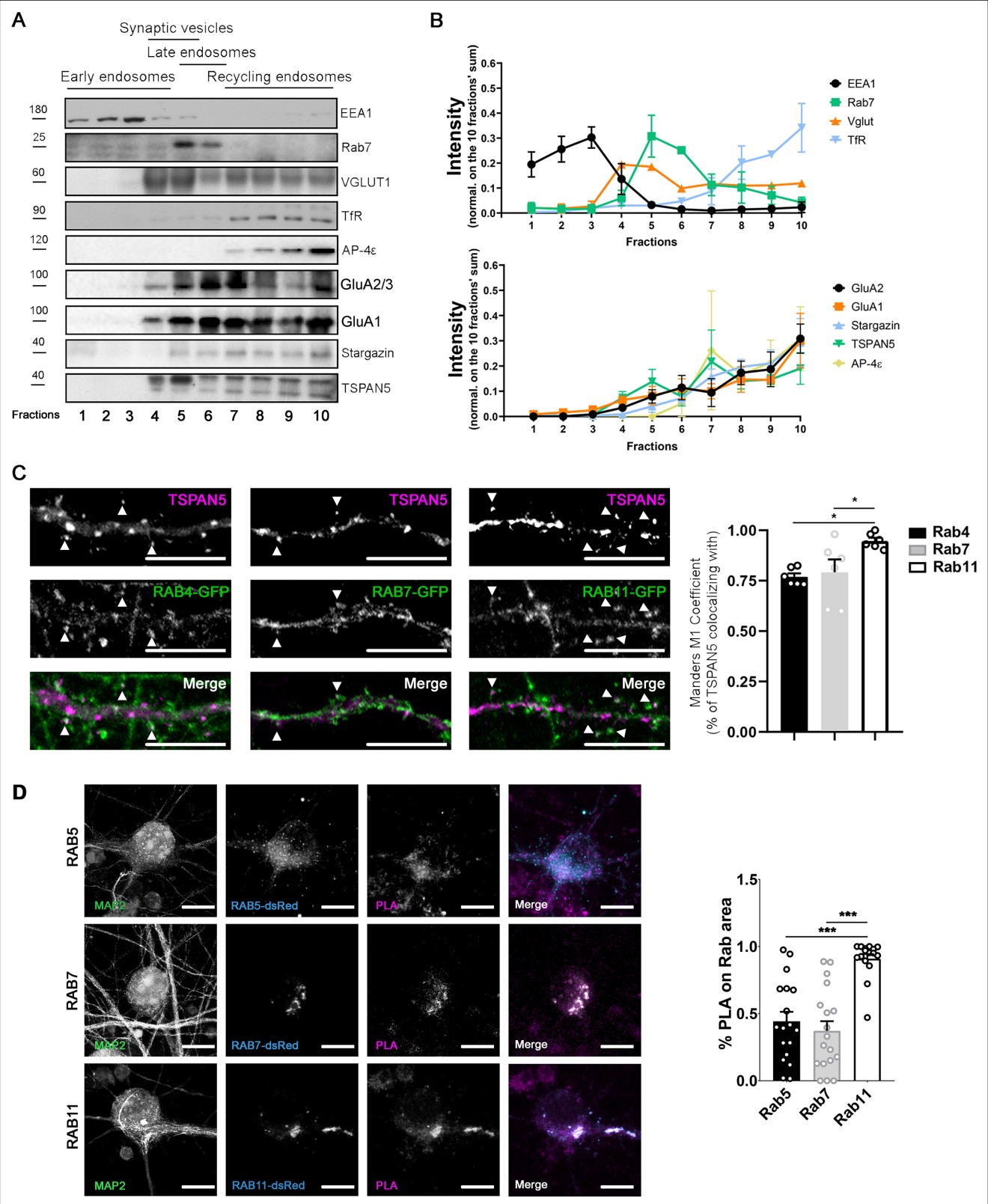

**Figure 3.** TSPAN5 complex with AP4, Stargazin, and α-amino-3-hydroxy-5-methyl-4-isoxazolepropionic acid receptors (AMPARs) localises in recycling endosomes. (**A**) Vesicles fractionation from synaptosomes obtained from adult rat hippocampus and cortex. Ten isovolumetric fractions were isolated. Blots were probed for: EEA1 for early endosomes, Rab7 for late endosomes, VGLUT1 for synaptic vesicles, TfR for recycling endosomes, AP4ε, GluA1, GluA2/3, Stargazin, and TSPAN5. (**B**) Quantification of the experiment in panel A: the intensity of each band was normalised over the sum of the

*Figure 3 continued on next page*

*Figure 3 continued*

intensity of the bands in the 10 fractions. n = 3 separate experiments. (**C**) Top panel: Confocal images of DIV20 cultured rat hippocampal neurons transfected at DIV12 with plasmids encoding either Rab4-GFP, Rab7-GFP, or Rab11-GFP and immunolabelled for TSPAN5 (magenta). Colocalising puncta are highlighted by white arrowheads. Scale bar = 10 µm. Bottom panel: Quantification of TSPAN5 colocalisation (Mander's M1 coefficient) with RAB4-GFP, RAB7-GFP, and RAB11-GFP (Mander's M1 coefficient: Rab4, 0.77±0.02; Rab7, 0.79±0.06; 0.94±0.02). n = 6 neurons per condition. One Way ANOVA, Newman-Kulspost hoc multiple comparison test. (**D**) Left panel: Confocal images of DIV20 cultured mouse hippocampal neurons transfected at DIV12 with plasmids encoding either Rab5-DsRed, Rab7-DsRed, or Rab11-DsRed, immunolabelled for MAP2 (green) and subjected to proximity ligation assay (PLA) on TSPAN5 and GluA2 antibodies, with far red detection probe (magenta). DsRed signal is shown in cyan. Scale bar = 10 µm. Right panel: Quantification of PLA signal colocalisation with Rab5-DsRed, Rab7-DsRed, and Rab11-DsRed (Mander's M1 coefficient: Rab5, 0.44±0.07; Rab7, 0.37±0.07; Rab11, 0.91±0.03). n = 16-18 neurons. One Way ANOVA, Newman-Kulspost hoc multiple comparison test. Values represent the mean ± SEM. *=p < 0.05, **=p < 0.01, ***=p < 0.001.

The online version of this article includes the following source data and figure supplement(s) for figure 3:

**Source data 1.** Individual data values for the graphs in panels B, C, and D.

**Source data 2.** Raw images and images with cropped areas highlighted of the blots in panel A.

**Figure supplement 1.** Control experiments for the proximity ligation assay (PLA).

marker transferrin receptor (***Figure 3A and B***). This experiment suggests that the intracellular pool of TSPAN5 could associate with AP4, Stargazin, GluA1, and GluA2 in recycling endosomes. We also analysed the localisation of TSPAN5 in cultured hippocampal neurons by evaluating its colocalisation with over-expressed GFP-tagged Rabs: Rab4, Rab7, and Rab11, markers of early, late, and recycling endosomes, respectively (***Figure 3C***). TSPAN5 showed a high level of colocalisation with all three Rabs. This is not surprising as TSPAN5 is likely to be transported in the endolysosomal pathway, similarly to many other transmembrane proteins that can localise in the plasma membrane. However, colocalisation with Rab11-positive endosomes was significantly higher than with the other two Rabs. To further clarify the location where the complex forms, we transfected DIV12 rat hippocampal neurons with dsRed-tagged Rab5, Rab7, or Rab11 and performed proximity ligation assay (PLA) using antibodies directed against TSPAN5 and GluA2 (***Figure 3D***). We then measured the colocalisation of the PLA signal, corresponding to sites of proximity between TSPAN5 and GluA2, with each of the Rabs. Although some level of colocalisation was visible with all three Rabs, this was more pronounced with Rab11 (***Figure 3D***), strongly pointing towards Rab11-positive organelles as the location in which the complex between TSPAN5, AP4, Stargazin, and AMPARs preferentially forms. Control experiments with only one or the other antibody are shown in ***Figure 3— figure supplement 1***.

## TSPAN5 depletion affects surface and total levels of AMPAR subunits GluA2 and GluA1

Our data so far suggest that TSPAN5 could participate in the intracellular trafficking of AMPARs, a tightly regulated process that is crucial to maintain a correct level of receptors at the synapse membrane and to ensure efficient synaptic transmission (***Anggono and Huganir, 2012***).

To investigate this possibility, we transfected rat hippocampal neurons at DIV12 with either scrambled, Sh-TSPAN5, or rescue constructs and measured the surface levels of the two most abundant subunits of AMPARs GluA2 (***Figure 4A***) and GluA1 (***Figure 4B***) at DIV20. We observed that knockdown of TSPAN5 induced a reduction of surface GluA2 levels that was reversed in the rescue condition (***Figure 4A***). In contrast, GluA1 appeared to be increased upon TSPAN5 knockdown (***Figure 4B***), an effect that was reversed in the rescue condition. We also analysed surface levels of both GluA2 and GluA1 specifically in the postsynaptic compartment, by restricting the analysis on dendritic spines or dendritic shafts as identified in the GFP channel by morphological criteria. The reduction of GluA2 and the increase in GluA1 were present in both compartments (***Figure 4A and B***), suggesting that these effects are not restricted to dendritic spines.

We confirmed these results by BS3 crosslinking in hippocampal neurons that were transduced with lenti-viral particles carrying scrambled, Sh-TSPAN5, or rescue DNA (***Figure 4C and D***). In these experiments, we observed a significant reduction in plasma membrane and total levels of GluA2/3, possibly suggesting an increased degradation of the receptor in addition to its reduced plasma membrane localisation. Similarly, the increase in GluA1 was observed both in the plasma membrane fraction and in the total level, suggesting a potential compensatory effect driven by increased protein synthesis or reduced degradation.

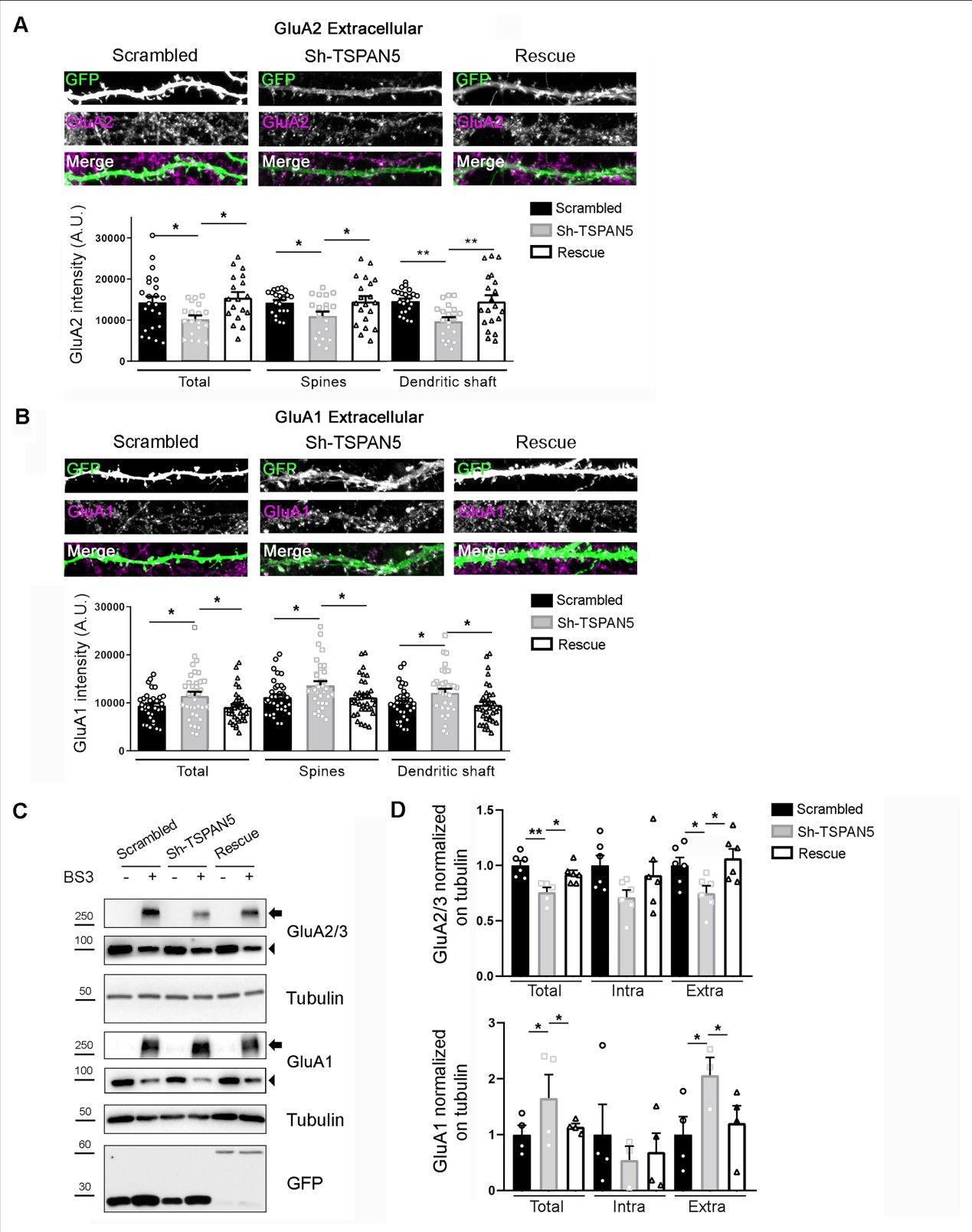

**Figure 4.** TSPAN5 depletion affects surface and total levels of the AMPAR subunits GluA2 and GluA1. (**A**) Top panel: Confocal images of dendrites from cultured rat hippocampal neurons transfected at DIV12 with either scrambled, Sh-TSPAN5, or rescue constructs, all co-expressing GFP and immunostained at DIV20 with an antibody against an extracellular epitope of GluA2 (magenta) in non-permeabilising condition. Boxes are 20 μm wide. Full neurons are shown in *Figure 4—figure supplement 1*. Bottom panel: Quantification of surface GluA2 signal mean intensity on the whole

*Figure 4 continued on next page*

*Figure 4 continued*

GFP-positive area (GluA2 intensity [A.U.] total: scrambled 14302±1430, Sh-TSPAN5 10250±884, rescue 15476±1352); GluA2 mean intensity in dendritic spines (scrambled 14290±593, Sh-TSPAN5 11006±1055; rescue 14544±1293); GluA2 mean intensity in dendritic shafts (scrambled 14579±610; Sh-TSPAN5 9730±921; rescue 14512±1482). N = scrambled, 23; Sh-TSPAN5, 19; rescue, 18 neurons. One Way ANOVA, Newman-Kulspost hoc multiple comparison test. (**B**) Top panel: Confocal images of dendrites from cultured rat hippocampal neurons transfected at DIV12 with either scrambled, Sh-TSPAN, or rescue constructs, all co-expressing GFP and immunostained at DIV20 with an antibody against an extracellular epitope of GluA1 (magenta) in non-permeabilising condition. Boxes are 20 µm wide. Full neurons are shown in the *Figure 4—figure supplement 1*. Bottom panel: Quantification of GluA1 signal mean intensity on the whole GFP-positive area (GluA1 intensity [A.U.] total: scrambled 9404±494, Sh-TSPAN5 11492±817, rescue 9167±565); GluA1 mean intensity in dendritic spines (scrambled 11232±599; Sh-TSPAN5 13711±831; rescue 11185±634); GluA1 mean intensity in dendritic shafts (scrambled 9914±563; Sh-TSPAN5 12128±808; rescue 9610±678). N = scrambled, 35; Sh-TSPAN5, 36; rescue, 35 neurons. One Way ANOVA, Newman-Kulspost hoc multiple comparison test. (**C**) BS3 crosslinking on DIV20 cultured rat hippocampal neurons infected at DIV12 with lentiviral particles encoding for scrambled, Sh-TSPAN5, or rescue all co-expressing GFP. Blots probed for AMPARs subunits GluA2/3 and GluA1. Tubulin was used as a loading control, GFP was used as a control for infection. Arrowheads indicate total and intracellular bands, arrows indicate crosslinked plasma membrane bands. Full blots are shown in the *Figure 4—figure supplement 1*. (**D**) Quantification relative to panel C (GluA2/3: total/tubulin: scrambled 1±0.04, Sh-TSPAN5 0.76±0.06, rescue 0.91±0.04; intra/tubulin: scrambled 1±0.09, Sh-TSPAN5 0.71±0.09, rescue 0.91±0.14; extra/tubulin: scrambled 1±0.06, Sh-TSPAN5 0.75±0.09, rescue 1.08±0.08; GluA1: total/tubulin: scrambled 1±0.15, Sh-TSPAN5 1.65±0.22, rescue 1.14±0.0; intra/tubulin: scrambled 1±0.48, Sh-TSPAN5 0.5±0.4, rescue 0.69±0.44; extra/tubulin: scrambled 1±0.29, Sh-TSPAN5 2.06±0.14, rescue 1.21±0.23). n = 4/6 independent cultures. One Way ANOVA, Newman-Kulspost hoc multiple comparison test. Values represent the mean ± SEM. *=p < 0.05, **=p < 0.01, ***=p < 0.001.

The online version of this article includes the following source data and figure supplement(s) for figure 4:

**Source data 1.** Individual data values for the graphs in panels A, B, and D.

**Source data 2.** Raw images and images with cropped areas highlighted of the blots in panel C.

**Figure supplement 1.** Full images and blots related to *Figure 4*.

## TSPAN5 and AP4 regulate surface GluA2 levels without affecting its internalisation

To investigate whether the TSPAN5-AP4 complex is responsible for the regulation of GluA2 surface levels, we evaluated the GluA2 surface levels in neurons transfected with a construct carrying the Sh-TSPAN5 and the cDNA for the human TSPAN5 lacking the C-terminus (rescue ΔC) (*Figure 5A and B*), since this region is the one interacting with AP4 (*Figure 2A*). We found that the TSPAN5- ΔC was unable to rescue the knockdown of TSPAN5, confirming the importance of the TSPAN5-AP4 interaction for maintaining the correct surface levels of GluA2.

We then designed guide RNAs (gRNAs) to knock down AP4β and -ε via CRISPR/Cas9 and tested them by generating lentiviral particles and infecting cultured rat hippocampal neurons. Both RT-PCR and western blotting showed efficient reduction in the levels of AP4 (*Figure 5—figure supplement 1*). In line with previous findings (*Matsuda et al., 2008*; *Hirst et al., 2013*), targeting one subunit reduced the expression also of the others. We then transduced cultured rat hippocampal neurons at DIV12 with lentiviral particles coding for either a control, an AP4β, or an AP4ε gRNA and simultaneously with lentiviral particles coding for scrambled or TSPAN5 shRNA. We analysed GluA2 surface levels at DIV20 and observed that the knockdown of either AP4 or TSPAN5 reduced GluA2 levels to the same extent. In addition, the simultaneous knockdown of TSPAN5 and AP4 did not induce any further reduction (*Figure 5C*), supporting the hypothesis that the two proteins participate in the same pathway.

According to the hypothesised complex arrangement, in which AP4 mediates the interaction between TSPAN5 and Stargazin-AMPARs, we reasoned that removal of AP4 would induce a reduction in the surface levels of GluA2 due to its impossibility to engage in the complex with TSPAN5. To demonstrate this, we performed PLA between TSPAN5 and GluA2 in rat hippocampal neurons transfected at DIV12 with the Ctrl, AP4β, or AP4ε gRNAs (*Figure 5D*). As expected, reducing AP4 levels compromised the interaction between TSPAN5 and GluA2.

Another tetraspanin, TSPAN7, was previously shown to regulate AMPAR internalisation (*Bassani et al., 2012*). To explore whether TSPAN5 could have a similar role, we evaluated the internalisation of AMPARs using an antibody-feeding assay. Rat hippocampal neurons transfected at DIV12 with either scrambled or Sh-TSPAN5 were exposed to an α-GluA2 antibody directed against a surface epitope and incubated for different time points. Both scrambled- and Sh-TSPAN5-transfected neurons exhibited a significant increase in the GluA2 intracellular/total ratio after 5 min, suggesting that AMPAR internalisation is not affected by TSPAN5 knockdown. Interestingly, in scrambled-transfected neurons,

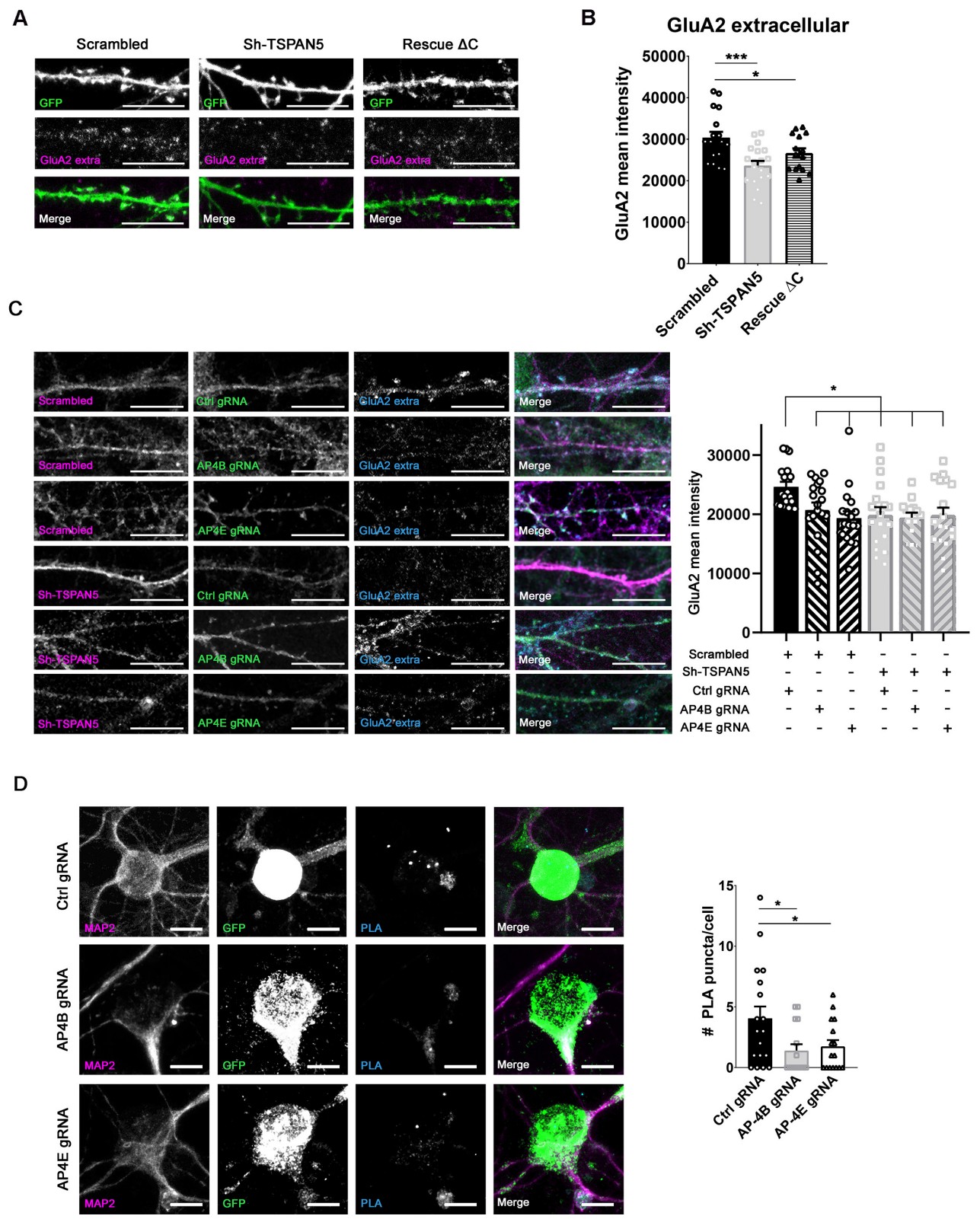

**Figure 5.** TSPAN5 and AP4 regulate surface GluA2 levels without affecting its internalisation. (**A**) Confocal images of DIV20 cultured rat hippocampal neurons transfected with vectors coding for GFP (green) and either a scrambled ShRNA, the Sh-TSPAN5, and a construct carrying the ShTSPAN5 and the cDNA for TSPAN5 lacking the C-terminus (rescue ΔC), and surface stained at DIV20 for GluA2 (magenta). Scale bar = 10 μm. (**B**) Quantification of the intensity of the surface GluA2 signal: GluA2 mean intensity (scrambled 30384±1390; Sh-TSPAN5 23654±1113; rescue ΔC 26686±1116). n = 15/19

*Figure 5 continued on next page*

*Figure 5 continued*

neurons. One Way ANOVA, Newman-Kulspost hoc multiple comparison test. (**C**) Left panel: Confocal images of DIV20 cultured rat hippocampal neurons transduced at DIV12 with lentiviral particles coding for an mCherry (magenta) and either scrambled or Sh-TSPAN5 and with lentiviral particles coding for a GFP (green), CRISPR/Cas9, and either a control guide RNA (Ctrl gRNA) or gRNAs directed against AP4B (AP4B gRNA) or AP4E (AP4E gRNA), respectively, and immunostained at DIV20 with an antibody against an extracellular epitope of GluA2 (cyan). Scale bar = 10 μm. Right panel: Quantification of the intensity of the GluA2 signal: GluA2 mean intensity (scrambled-Ctrl gRNA 24627±840; scrambled-AP4B gRNA 20737±1236; scrambled-AP4E gRNA 19339±1165; Sh-TSPAN5-Ctrl gRNA 19864±1331; Sh-TSPAN5-AP4B gRNA 19407±836; Sh-TSPAN5-AP4E gRNA 19836±1279). n = 18 neurons from three independent experiments. One Way ANOVA, Newman-Kulspost hoc multiple comparison test. (**D**) Left panel: Confocal images of DIV20 culture rat hippocampal neurons transfected at DIV12 with plasmid coding for a GFP (green), CRISPR/Cas9, and either a control guide RNA (Ctrl gRNA) or gRNAs directed against AP4B (AP4B gRNA) or AP4E (AP4E gRNA), respectively, immunostained for MAP2 (magenta) and subjected to proximity ligation assay (PLA) on TSPAN5 and GluA2 antibodies, with red detection probe (cyan). Scale bar = 10 μm. Right panel: Quantification of the density of PLA signal per cell (# PLA puncta/cell: Ctrl gRNA 4.1±1; AP4B gRNA 1.4±0.5; AP4E gRNA 1.8±0.5). n = 18, 15, 16 neurons, respectively from three independent experiments. One Way ANOVA, Newman-Kulspost hoc multiple comparison test. Values represent the mean ± SEM. *=p < 0.05, **=p < 0.01, ***=p < 0.001.

The online version of this article includes the following source data and figure supplement(s) for figure 5:

**Source data 1.** Individual data values for the graphs in panels B, C, and D.

**Figure supplement 1.** CRISPR/Cas9 knockdown of AP-4 subunits in hippocampal neurons.

**Figure supplement 1—source data 1.** Individual data values for the graphs presented in panels A and B.

**Figure supplement 1—source data 2.** Raw images and images with cropped areas highlighted of the blots in panel B.

**Figure supplement 2.** Left panel: Confocal images of secondary dendrites from DIV20 cultured rat hippocampal neurons transfected at DIV12 with either scrambled or Sh-TSPAN5 constructs, both co-expressing GFP.

**Figure supplement 2—source data 1.** Individual data values for the graphs presented.

recycling of the receptor at 10 min post incubation brought GluA2 back to the surface, with levels of the intracellular/total ratio similar to those at time point 0; by contrast, the Sh-TSPAN5-transfected neurons maintained higher levels of internalised receptor at the 10 min time point. This potentially points to defects in GluA2 exocytosis.

## TSPAN5 regulates exocytosis of GluA2-containing AMPARs

Given the possible localisation of the TSPAN5, AP4, Stargazin, and AMPAR complex in Rab11-positive organelles, which have been shown to mediate receptor recycling back to the plasma membrane, we decided to directly evaluate GluA2 recycling. To this end, we relied on an overexpression model as recycling levels of endogenous GluA2 receptors are too low to be detected by an antibody-feeding approach. We decided to use super-ecliptic pHluorin (SEP)-tagged GluA2, where the SEP has been inserted in the extracellular domain of GluA2 (*Ashby et al., 2004*; *Sankaranarayanan et al., 2000*; *Hildick et al., 2012*). SEP is extremely useful for intracellular trafficking studies as it is only fluorescent at a neutral pH, allowing for the visualisation of the receptor only when it is exposed to the neutral extracellular environment. Instead, its fluorescence is quenched in the acidic intracellular vesicles. We verified the functionality of SEP-GluA2 by exposure to imaging media at pH 6, which completely abolished the signal, or to media containing $NH_4Cl$ that alkalinise also intracellular vesicles (*Figure 6—figure supplement 1*). To avoid interferences from endoplasmic reticulum (ER)-contained GluA2, we only evaluated the signal in individual dendritic spines, which are virtually devoid of ER (*Rathje et al., 2013*; *Rathje et al., 2014*; *Wilkinson et al., 2014*). We applied a FRAP-FLIP protocol in which a portion of dendrite (ROI) is bleached and then imaged over 300 s while repetitively bleaching the flanking regions of the ROI to eliminate interference from receptors laterally diffusing into the ROI (*Hildick et al., 2012*; *Figure 6A*). This protocol allows for the selective visualisation of receptors that were present in intracellular compartments at the time of the initial bleaching. These receptors are quenched and therefore not affected by bleaching, retaining the ability to fluoresce once exposed to the neutral extracellular environment. As such, the protocol allows for the visualisation of newly synthesised receptors and internalised receptors recycling back to the plasma membrane. Application of cycloheximide before the experiment allowed us to block synthesis of new receptors, thereby restricting the analysis to recycling receptors only (*Figure 5A*). To our surprise, there were no differences in the levels or kinetics of SEP-GluA2 recycling between scrambled-, Sh-TSPAN5-, or rescue mCherry-transfected neurons (*Figure 6B–D*). Considering the reduction in GluA2 surface levels shown before (*Figure 3*), the absence of differences in this experiment could be either due to cycloheximide blocking the synthesis of other proteins necessary for TSPAN5-dependent recycling of AMPARs, thus masking the effect of TSPAN5 knockdown, or

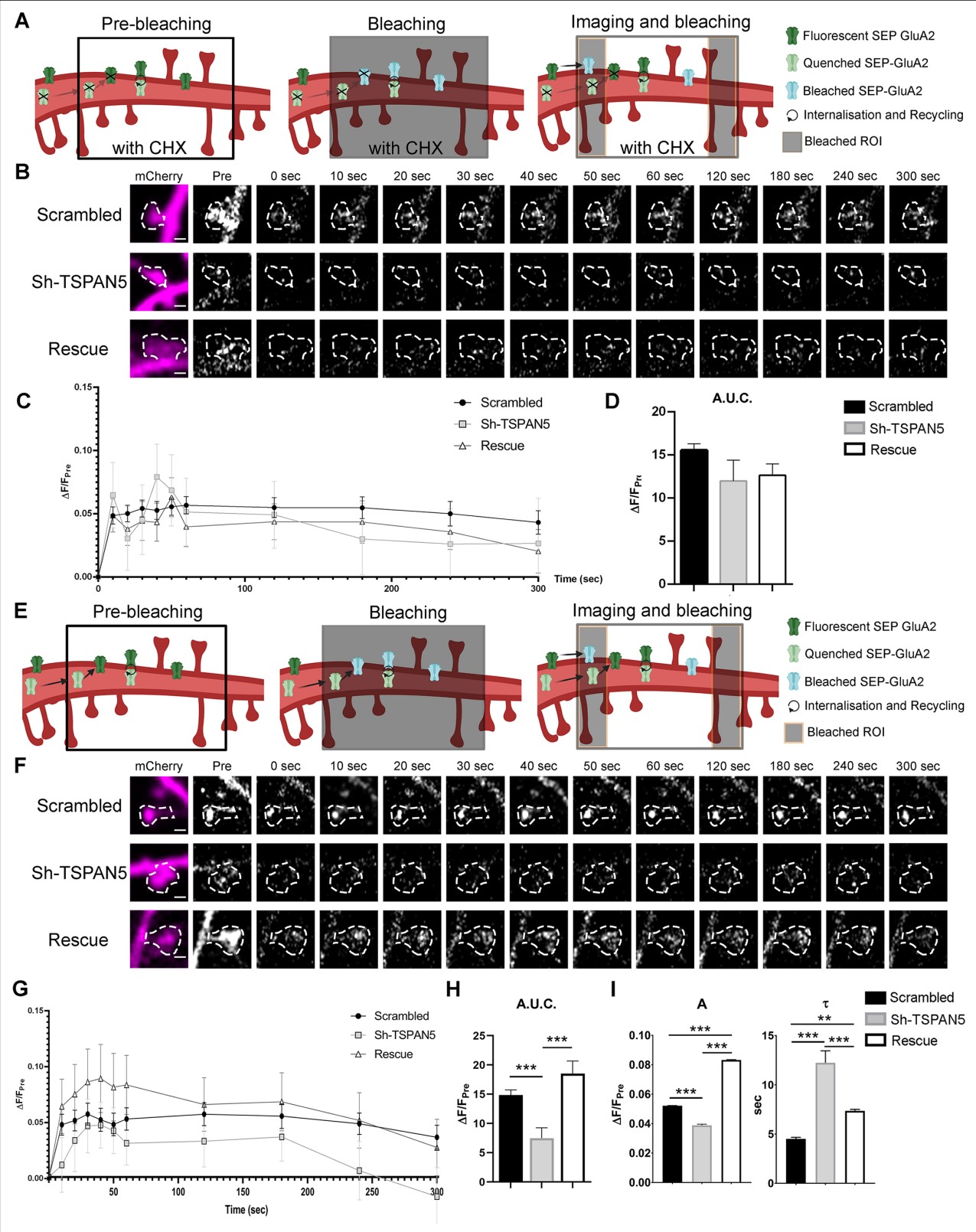

**Figure 6.** TSPAN5 regulates exocytosis of GluA2-containing α-amino-3-hydroxy-5-methyl-4-isoxazolepropionic acid receptors (AMPARs). (**A**) Schematic of the FRAP-FLIP experiment presented in panel **B**. SEP-GluA2 in pre-bleaching condition is either fluorescent (green) if exposed to the extracellular media or quenched (light green) if in intracellular compartments. A region of the dendrite is bleached (black box). SEP-GluA2 that was fluorescent (and so extracellularly exposed) at the time of bleaching becomes bleached (light blue). Quenched SEP-GluA2 is not affected by the bleaching.

*Figure 6 continued on next page*

*Figure 6 continued*

During imaging, the ROI flanking regions are continuously bleached (black lateral boxes), thus lateral diffusing SEP-GluA2 will be bleached. Receptors that have been internalised and directed for recycling are exocytosed and become fluorescent. Newly synthesised receptors would not be present due to the application of cycloheximide (CHX) (crossed out receptors). Controls for pH sensitivity of the SEP signal are shown in *Figure 6—figure supplement 1*. (**B**) Live confocal images of individual dendritic spines from DIV20 cultured rat hippocampal neurons transfected at DIV12 with SEP-GluA2 and either scrambled, Sh-TSPAN5, or rescue construct co-expressing mCherry. Neurons were treated for 2 hr with 200 µg/ml of cycloheximide to inhibit protein synthesis and then imaged under a FRAP-FLIP protocol for 5 min to isolate the recycling receptors. mCherry (magenta) and SEP-GluA2 (white) images (time points: prebleach, postbleach, 10, 20, 30, 60, 120, 180, 240, and 300 s) are shown. The dendritic spine mask is depicted with white dashed line. Scale bar =1 µm. (**C**) Quantification of the $\Delta F/F_{pre}$ for SEP-GluA2 over time for scrambled-, Sh-TSPAN5-, and rescue-transfected neurons. (**D**) Quantification of the area under the curve relative to panel B (area under the curve [A.U.]: scrambled 15.56±0.74, Sh-TSPAN5 11.99±2.51, rescue 11.77±1.31). n = scrambled, 56; Sh-TSPAN5, 53; rescue, 53 dendritic spines. One Way ANOVA, Newman-Kulspost hoc multiple comparison test.(**E**) Schematic of the FRAP-FLIP experiment presented in panel F. SEP-GluA2 at basal condition is either fluorescent (green) if exposed to the extracellular media or quenched (light green) if in intracellular compartments. A region of the dendrite is bleached (black box). SEP-GluA2 that was fluorescent (and so extracellularly exposed) at the time of bleaching becomes bleached (light blue). Quenched SEP-GluA2 is not affected by the bleaching. During imaging the ROI flanking regions are continuously bleached (black box), thus lateral diffusing SEP-GluA2 will be bleached. Receptors that have been internalised and directed for recycling are exocytosed and become fluorescent. Newly synthesised receptors could also travel in intracellular vesicles to be exocytosed and become fluorescent. (**F**) Confocal images of individual dendritic spines from DIV20 cultured rat hippocampal neurons transfected at DIV12 with SEP-GluA2 and either scrambled, Sh-TSPAN5, or rescue construct co-expressing mCherry. Neurons were imaged under a FRAP-FLIP protocol for 5 min to analyse receptor exocytosis. mCherry (magenta) and SEP-GluA2 (white) images (time points: prebleach, postbleach, 10, 20, 30, 60, 120, 180, 240, and 300 s) are shown. The dendritic spine mask is depicted with white dashed line. Scale bar =1 µm. (**G**) Quantification of the $\Delta F/F_{pre}$ for SEP-GluA2 over time for scrambled-, Sh-TSPAN5-, and rescue-transfected neurons. (**H**) Quantification of the area under the curve relative to panel E (area under the curve [A.U.]: scrambled 14.85±0.89, Sh-TSPAN5 7.49±1.77, rescue 18.5±2.18). One Way ANOVA, Newman-Kulspost hoc multiple comparison test. (**I**) Quantification of the parameters A and $\tau$, representative of the steady state $\Delta F/F_{pre}$ and of the time constant of the exocytosis kinetics, based on the fitting of the first eight time points with the exponential function ($\Delta F/F_{pre} = A \left( 1 - e^{-\frac{t}{\tau}} \right)$ a ($\Delta F/F_{pre}$): scrambled, 0.0522±0.0002; Sh-TSPAN5, 0.0388±0.0008; rescue, 0.0832±0.0003). ($\tau$ (s): scrambled, 4.5±0.2; Sh-TSPAN5, 12.2±1.2; rescue, 7.4±0.2). n = scrambled, 56; Sh-TSPAN5, 35; rescue, 29 dendritic spines. One Way ANOVA, Newman-Kulspost hoc multiple comparison test. Values represent the mean ± SEM. *=p < 0.05, **=p < 0.01, ***=p < 0.001.

The online version of this article includes the following source data and figure supplement(s) for figure 6:

**Source data 1.** Individual data values for the graphs presented in panels C and G.

**Figure supplement 1.** Control experiments for SEP-GluA2 sensitivity to pH.

because TSPAN5 regulates the exocytosis of newly synthesised receptors. We thus used the same FRAP-FLIP approach but without application of cycloheximide; this experimental setup allows for the simultaneous observation of recycling receptor and exocytosis of newly synthesised receptor (*Figure 6E*). In this experiment, we observed a significant reduction of the recovery after photobleaching in individual dendritic spines of Sh-TSPAN5-transfected neurons compared to scrambled-transfected neurons as measured by the area under the curve (*Figure 6F–H*). Given this change, we decided to analyse the amplitude and kinetic of exocytosis. To do this, we fitted an exponential curve ($\Delta F/F_{pre} = A \left( 1 - e^{-\frac{t}{\tau}} \right)$) onto our data according to *Hildick et al., 2012*. We then extrapolated the values for A, corresponding to the steady state $\Delta F/F_{pre}$, and $\tau$, which represents a time constant related to the kinetic of exocytosis (*Figure 6I*). For both parameters the Sh-TSPAN5 neurons presented significant differences compared to the scrambled condition with smaller A and greater $\tau$, suggesting lower steady-state recovery of GluA2 and slower kinetics (*Figure 6I*). These defects were completely reversed in rescue-transfected neurons, even showing a potentiation of the recovery. These data strongly suggest that exocytosis of newly synthesised GluA2 receptors is regulated by TSPAN5. However, our experiments do not exclude the possibility that TSPAN5 could also regulate the recycling of GluA2-containing AMPARs, an effect that would be masked by the application of cycloheximide in the experiment presented in *Figure 6A–D* which could cause the loss of rapidly turning over factors needed for this process.

## TSPAN5 regulates exocytosis of newly synthesised AMPARs possibly by avoiding their degradation via the lysosomal pathway

To further confirm the role of TSPAN5 in AMPAR exocytosis, we took advantage of an ER retention system called ARIAD (*Hangen et al., 2018*; *Rivera et al., 2000*). In this system, the ARIAD-GluA2 is synthesised in the ER similarly to endogenous GluAs, but the presence of a conditional aggregation domain (CAD) results in its retention in this compartment. The protein can be released in a controlled manner by application of the ARIAD ligand that causes the disassembly of the CAD allowing the protein to continue along the secretory

pathway. The fusion protein also presents a myc tag on the extracellular side allowing detection of the exocytosed receptor. As such, by applying an anti-myc antibody in the culture media after exposing the cells to the ARIAD ligand, one can assess the levels of plasma membrane inserted ARIAD-GluA2 directly coming from the ER site of synthesis (*Figure 7A, B*). As expected from our previous results, TSPAN5 knockdown resulted in a reduction in the surface levels of ARIAD-GluA2 90 min after application of the ARIAD ligand, an effect that was rescued by re-expression of the Sh-resistant form of TSPAN5 (*Figure 7C*). We also analysed dendritic transport of ARIAD-tdTomato-GluA2 via live imaging of neurons 30 min after addition of the ARIAD ligand. Here, we did not detect any change in the average speed of transport of GluA2-containing vesicles in either the anterograde or retrograde direction (*Figure 7—figure supplement 1A, B*), nor in the average number of vesicles (*Figure 7—figure supplement 1C*). These results suggest that there is either a lower amount of GluA2 loaded into each of these vesicles directed for exocytosis or that these vesicles fail to deliver their content to the plasma membrane of dendrites and might be directed for degradation as a result.

To test this second possibility, we assessed the total levels of GluA2/3 via immunofluorescence in DIV20 neurons transfected at DIV12 with either scrambled or Sh-TSPAN5 and treated with the lysosomal inhibitor leupeptin (*Figure 7D and E*), since AMPARs are mostly degraded via this pathway (*Ehlers, 2000*). Leupeptin treatment increased GluA2/3 to similar levels in scrambled- and Sh-TSPAN5-transfected neurons, suggesting that AMPARs degradation is increased in the absence of TSPAN5 (*Figure 7D and E*). However, this experiment does not directly demonstrate that GluA2-containing AMPARs are rerouted towards degradation. We also tested whether the exocytosis of newly synthesised GluA1 is regulated by TSPAN5 by using the same ARIAD system (*Figure 7F*). Silencing TSPAN5 also reduced the surface levels of newly synthesised GluA1, which was rescued by re-expressing an ShRNA-resistant TSPAN5. This strengthen the hypothesis that the overall increase in the surface GluA1 levels (*Figure 4B–D*) is a compensatory mechanism.

It is important to note that these experiments still do not exclude a possible regulation of TSPAN5 on recycling AMPARs.

Altogether, our data support a model whereby the association of TSPAN5 with GluA2, occurring via AP4 and Stargazin, promotes the exocytosis of AMPARs, potentially via Rab11/TfR-positive recycling endosomes (*Figure 8*).

## Discussion

In this work, we have identified an intracellular pool of TSPAN5 that participates in the delivery of newly synthesised AMPARs to the plasma membrane. We showed that TSPAN5 forms a complex with AP4, Stargazin, and AMPARs and that this interaction could take place in recycling endosomes. Although the main function of recycling endosomes is to redirect endocytosed receptors back to the plasma membrane, they have also been shown to participate in a non-canonical secretory pathway. Proteins synthesised in the dendritic ER are trafficked to an ER-Golgi intermediate compartment before being loaded to recycling endosomes for insertion in the plasma membrane (*Bowen et al., 2017*; *Hirling, 2009*). As a result, the receptors would bypass the Golgi compartment, which is poorly present in dendrites and dendritic spines. The molecular regulators of this process are not well defined. However, our data do not fully identify the nature of the organelles involved, therefore further investigations are required.

In addition, our experiments cannot fully exclude that TSPAN5 could also regulate the recycling of AMPARs. It is thus possible that TSPAN5 could modulate the delivery to the plasma membrane of both newly synthesised and recycling AMPARs.

Although our experiments show a differential effect of TSPAN5 knockdown on surface levels of GluA2 and GluA1 at the steady state, TSPAN5 appears to regulate the exocytosis of both GluA2 and GluA1. This is in line with the fact that both can interact with Stargazin (*Chen et al., 2000*), and thus with AP4 and TSPAN5 (*Figure 2*). The differences at the steady state could be due to a compensatory potentiation of a secretory pathway that does not rely on TSPAN5 and that is responsible for GluA1-containing AMPAR delivery exploited to maintain normal synaptic activity. The trafficking of GluA2 and GluA1 was previously shown to be partially regulated by separate mechanisms for example with GluA2 delivery and recycling being a constitutive process, whereas GluA1 exocytosis to the plasma membrane is more dependent on synaptic plasticity (*Passafaro et al., 2001*; *Shi et al., 2001*). Our findings do not elucidate whether AMPARs exhibit a different subunit composition upon TSPAN5 knockdown. However, the fact that we observe a reduction in GluA2 and GluA3 and an increase in GluA1 potentially suggests that there could be an overall reduction

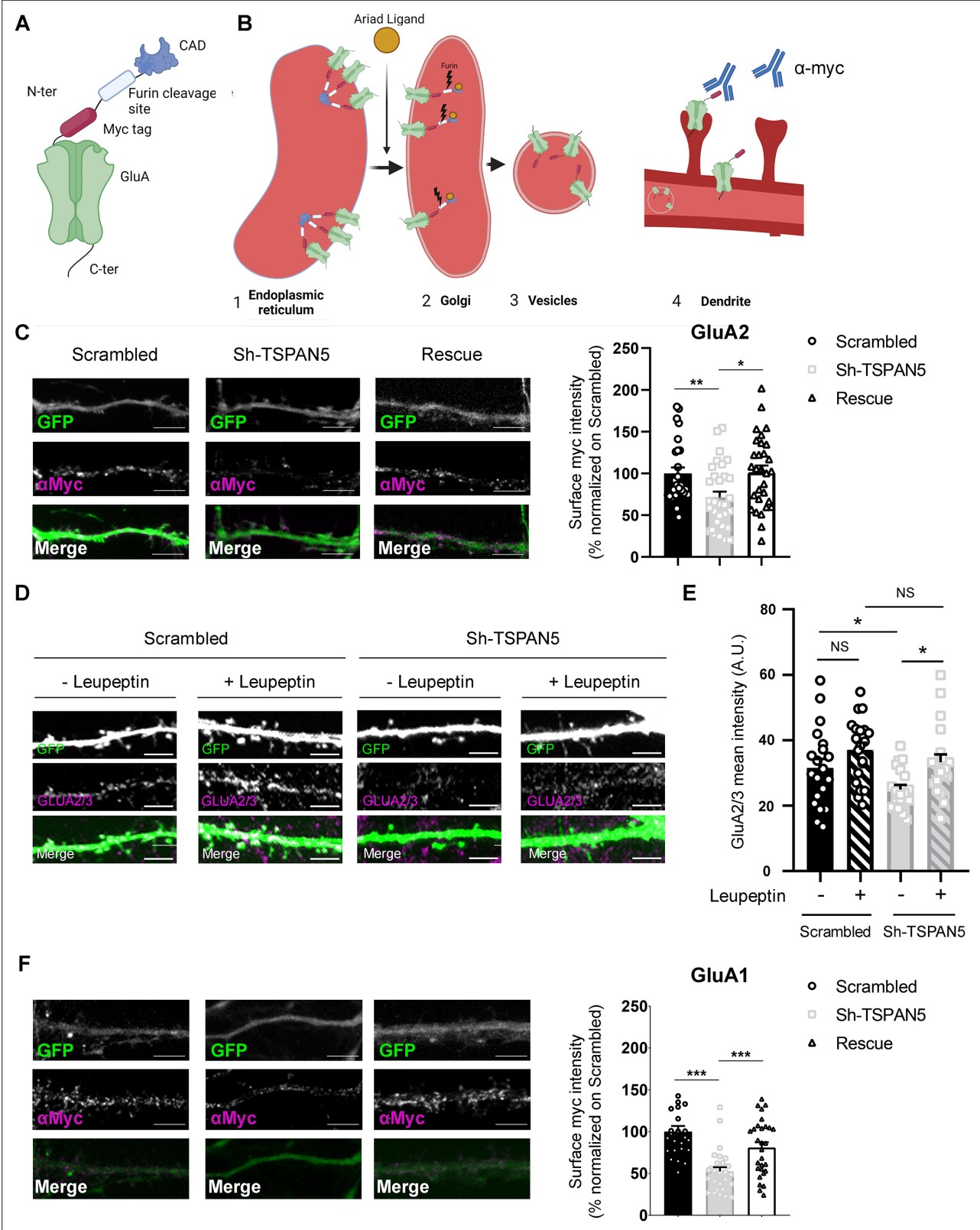

**Figure 7.** TSPAN5 regulates exocytosis of newly synthesised α-amino-3-hydroxy-5-methyl-4-isoxazolepropionic acid receptors (AMPARs), possibly by preventing their degradation via the lysosomal pathway. (**A**) Schematic of the ARIAD-GluA2 construct. (**B**) In basal conditions (**Berditchevski, 2001**), ARIAD-GluA2 is retained in the endoplasmic reticulum (ER) due to the self-assembly properties of the conditional aggregation domain (CAD). Upon application (**Charrin et al., 2002**), the ARIAD ligand binds to CAD, inhibits self-assembly and allows the ARIAD-GluA2 to move to the Golgi where

*Figure 7 continued on next page*

*Figure 7 continued*

the endogenous Furin protease cleaves the CAD. ARIAD-GluA2 can now be loaded onto secretory vesicles (*Charrin et al., 2014*), transported along the dendrites, and subsequently exocytosed (*Hemler, 2005*). Application of an anti-myc antibody in the culture medium allows for the detection of the plasma membrane pool of GluA2 that was released from the ER after application of the ARIAD ligand. (**C**) Left panel: Confocal images of DIV20 rat cultured hippocampal neurons transfected at DIV12 with the ARIAD-myc-GluA2 construct and with a plasmid coding for GFP (green) and either scrambled, Sh-TSPAN5, or rescue, and immunostained with an anti-myc antibody in live staining conditions (magenta) 90 min after the application of the ARIAD ligand. Scale bar = 5 µm. Right panel: Quantification of the surface anti-myc mean intensity normalised to scrambled (scrambled 100±7.14; Sh-TSPAN5 71.44±6.81; rescue 101.4±7.92). n = 27–31 neurons per condition. One Way ANOVA, Newman-Kulspost hoc multiple comparison test. (**D**) Confocal images of secondary dendrites from DIV20 rat cultured hippocampal neurons transfected at DIV12 with either scrambled or Sh-TSPAN5 constructs, both co-expressing GFP. Neurons were treated for 90 min with either vehicle ($H_2O$) or leupeptin (100 µM), fixed and immunostained for GLUA2/3 (magenta). Scale bar = 5 µm. (**E**) Relative quantification of GluA2/3 staining mean intensity (GluA2/3 mean intensity: scrambled vehicle 31.25±2.43; scrambled leupeptin: 36.95±2.25: Sh-TSPAN5 vehicle: 24.51±1.35; Sh-TSPAN5 leupeptin 33.3±2.22). n = scrambled vehicle, 20; scrambled leupeptin, 20; Sh-TSPAN5 vehicle, 20; Sh-TSPAN5 leupeptin, 20 neurons. One Way ANOVA, Newman-Kulspost hoc multiple comparison test. (**F**) Left panel: Confocal images of DIV20 rat cultured hippocampal neurons transfected at DIV12 with the ARIAD-myc-GluA1 construct and with a plasmid coding for GFP (green) and either scrambled, Sh-TSPAN5, or rescue, and immunostained with an anti-myc antibody in live staining conditions (magenta) 90 min after the application of the ARIAD ligand. Scale bar = 5 µm. Right panel: Quantification of the surface anti-myc mean intensity normalised to scrambled (scrambled 95.32±5.07; Sh-TSPAN5 52.49±4.95; rescue 81.064±6.2). n = 27–29 neurons per condition. One Way ANOVA, Newman-Kulspost hoc multiple comparison test. Values represent the mean ± SEM. *=p < 0.05, **=p < 0.01, ***=p < 0.001.

The online version of this article includes the following source data and figure supplement(s) for figure 7:

**Source data 1.** Individual data values for the graphs presented in panels C, E, and F.

**Figure supplement 1.** TSPAN5 knockdown does not affect intracellular transport speed or number of vesicles containing GluA2.

**Figure supplement 1—source data 1.** Individual data values for the graphs presented in panels A, B, and C.

in GluA2/3 tetramers and that the remaining GluA2 could potentially be redirected to GluA1/2 tetramers. In addition, an increase of GluA1 homomers could also occur. This could partially explain our previous observation that AMPAR-mediated mEPSCs are not affected by TSPAN5 knockdown in either their amplitude or frequency, but display altered kinetics (*Moretto et al., 2019*) which can be due to a change in the receptor subunit composition (*Lu et al., 2009*).

Our results also shed new light on AP4 function. AP complexes select transmembrane proteins via interaction through typical sorting motifs and promote their insertion into specific vesicles (*Robinson, 2004*). A role for AP4 in AMPARs intracellular trafficking was previously shown (*Matsuda et al., 2008*; *Matsuda et al., 2013*): AP4 was found to restrict AMPARs from being directed towards the axonal compartment. AP4β knockout mice presented with mislocalisation of AMPARs to the axon, which accumulated in autophagosomes. The authors did not detect a reduction of dendritic AMPARs in AP4β knockout neurons, however this could have been due to compensatory mechanisms arising in vivo upon constitutive knockout of the AP4 complex or because only levels of overexpressed AMPARs were analysed. It remains possible that other AP complexes could compensate for the loss of AP4. In particular, AP1 was found to regulate sorting and exocytosis of membrane proteins (*Bonifacino, 2014*). Interestingly, the involvement of AP4 in AMPAR exocytosis could potentially explain the intellectual disability phenotype of AP4 deficiency syndrome; an imbalance between GluA2 and GluA1 subunits in the composition of AMPARs was previously shown to cause changes in how neurons respond to synaptic plasticity events, thus impacting on learning and memory functions (*Moretto et al., 2018*).

As the association between AP4, Stargazin, and AMPARs was shown to occur in heterologous cells, with little to no expression of TSPAN5 (*Matsuda et al., 2008*), we believe that TSPAN5 is not necessary for the formation of the complex but that it could rather be involved in directing the complex to the correct organelle for its delivery to the plasma membrane.

Together with our previous work, these data highlight the importance of TSPAN5 for neuronal function. TSPAN5 appears to act on two independent pathways; on the one hand, its localisation at the plasma membrane is crucial for the maturation of dendritic spines during neuronal development (*Moretto et al., 2019*); on the other hand, TSPAN5 localisation in intracellular vesicles in mature neurons regulates exocytosis of AMPARs enabling correct synaptic function.

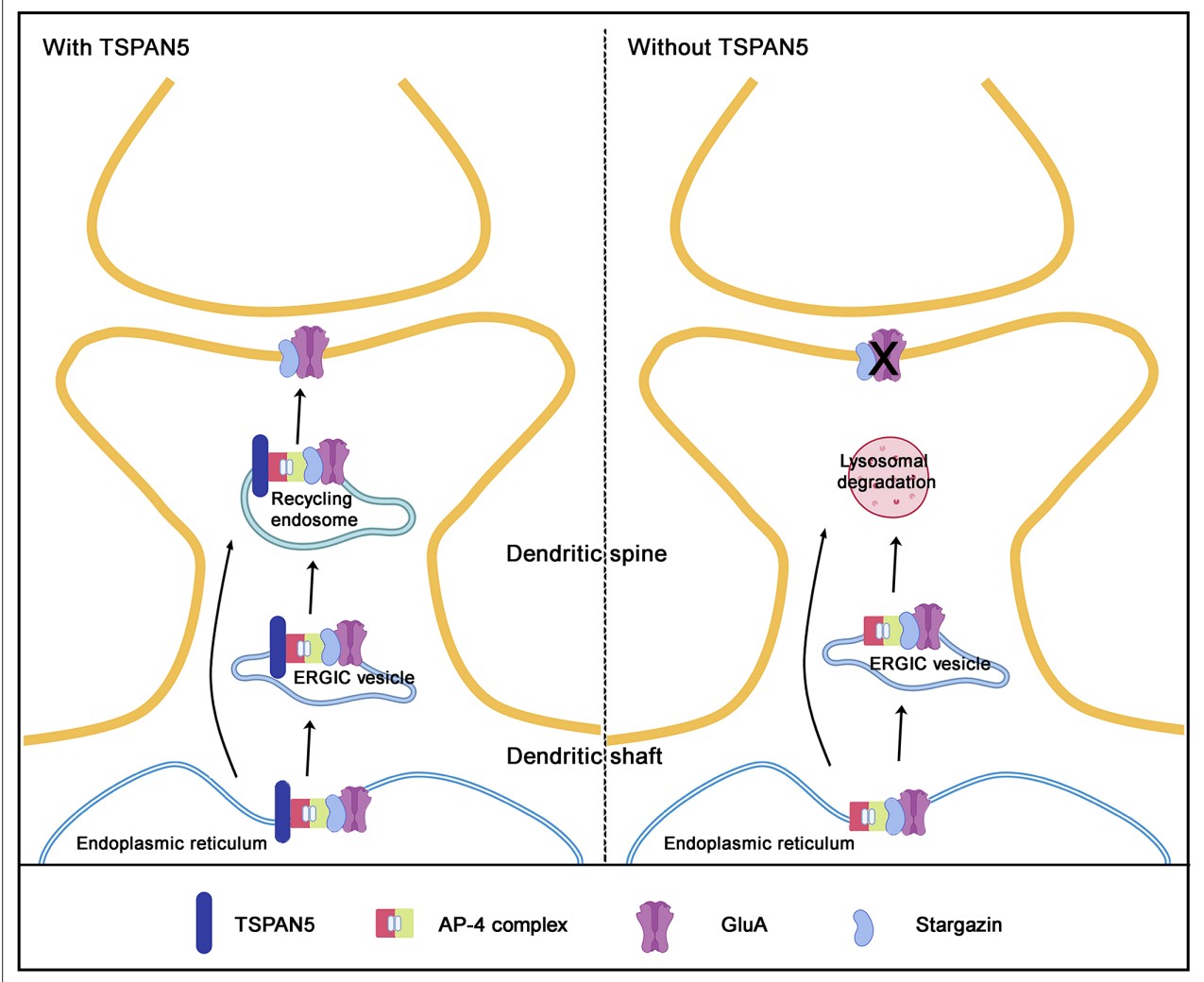

**Figure 8.** TSPAN5 regulates α-amino-3-hydroxy-5-methyl-4-isoxazolepropionic acid receptors (AMPARs) exocytosis through recycling endosomes by the formation of a tetrameric complex with AP4 and Stargazin. Working model of TSPAN5 function in mature neurons (left) and TSPAN5 silencing effects (right). TSPAN5 forms a complex with Stargazin and AMPARs in the endoplasmic reticulum or in endoplasmic reticulum-Golgi intermediate compartment (ERGIC) vesicles. The presence of TSPAN5 is necessary to direct the GluA2 and Stargazin complex to the plasma membrane. TSPAN5 silencing in neurons induces the redirection of GluA2-containing vesicles to lysosomal degradation.

## Experimental models

Animal procedures were performed in accordance with the European Community Council Directive of November 24, 1986 (86/609/EEC) on the care and use of animals. Animal procedures were approved by the Italian Ministry of Health (Protocol Number N° 2D46AN.463).

The HEK293 cell line (293 [HEK-293] CRL-1573 from ATCC, confirmed by STR profiling) used to generate the lentiviruses and the HeLa cells (HeLa CCL-2 from ATCC, confirmed by STR profiling) were grown in DMEM supplemented with 10% FBS, 1% L-glutamine, 0.1% gentamycin. All cell lines were tested for mycoplasma and confirmed negative.

Primary hippocampal neurons were prepared from Wistar E18 rat brains or form C57/BL6 E16 mouse brains (*Folci et al., 2014*; *Zapata et al., 2017*; *Valnegri et al., 2011*). Neurons were plated onto coverslips coated overnight with 0.25 mg/ml poly-D-lysine (Sigma-Aldrich) at 75,000 per well and grown in Neurobasal medium supplemented with 2% B27 prepared as in *Chen et al., 2008*, 0.25% L-glutamine, 1% penicillin/streptomycin, and 0.125% glutamate (Sigma-Aldrich).

Three-month-old male Wistar rats were used for hippocampus and cortex lysates.

## Methods

### Plasmids

pLVTHM-scrambled, pLVTHM-Sh-TSPAN5, pSicor-TSPAN5-GFP (rescue), pSicor-TSPAN5-ΔC-GFP (rescue ΔC), pSicor-scrambled-mCherry, pSicor-Sh-TSPAN5-mCherry, pSicor-TSPAN5-mCherry, pGEX4T1-TSPAN5-Ct, and TSPAN5-GFP have been characterised in our previous work (*Moretto et al., 2019*). Rab4-GFP, Rab7-GFP, and Rab11-GFP are kind gifts from Prof. G Schiavo. pCl-SEP-GluA2 was obtained from Addgene #24001 (*Kopec et al., 2006*). DsRed-Rab5, DsRed-Rab7, and DsRed-Rab11 were obtained from Addgene (#13050, #12661, #12679) (*Sharma et al., 2003*; *Choudhury et al., 2002*). Stargazin-HA and GluA2-myc plasmids were kind gifts of Dr Francoise Coussen. The pLenti-U6-(BsmBI)-hSyn-SaCas9-P2A-EGFP vector allowing the expression of *Staphylococcus aureus* Cas9 and a gRNA for the knockdown of AP4β and AP4ε were constructed by replacing the EF-1α promoter in the pLenti_SaCRISPR-EGFP plasmid (gift from Christopher Vakoc; Addgene #118636) with the hSyn promoter from the pAAV-hSyn-EGFP plasmid (gift from Bryan Roth; Addgene #50465). The gRNA sequences were designed as previously described (*Jaudon et al., 2020*; *Jaudon et al., 2022*; *Riccardi et al., 2022*) and were inserted downstream of the U6 promoter using BsmbI cloning sites. EGFP expression was used for visualisation of the transduced neurons. The gRNAs were CCGGTAGCGCAGCCTATCAGC and TTGATGAATCCTTACGAAGAG for AP4β and -ε, respectively. The control non-targeting gRNA sequence was GTTCCGCGTTACATAACTTA.

### Yeast two-hybrid screening

For yeast two-hybrid experiments, a fragment corresponding to the TSPAN5 C-terminal tail (aa 254–268) was cloned in frame with the GAL4 DNA-binding domain (pGBKT7 vector) and used as bait to screen a human adult brain cDNA library (Clonetech, Mate and Plate Library). Positives clones (3+) grew on plates containing X-α-GAL and Aureobasidin A (QDO/X/A plates) and expressed all four integrated reporter genes: HIS3, ADE2, AUR1C, and MEL1 under the control of three distinct Gal4-responsive promoters. cDNA plasmids from positive clones were recovered via DH5a *Escherichia coli* (*E. coli*) and sequenced.

### Transfection and infection

For lentivirus production, HEK293FT cells were transfected using the calcium phosphate method. Briefly, DNA was mixed with 130 mM $CaCl_2$ in $H_2O$. One volume of HEBS buffer (280 mM NaCl, 100 mM HEPES, 1.5 mM $Na_2HPO_4$, pH 7.11) was added to the DNA and thoroughly mixed to produce air bubbles. The mix was added to the cells and left for 5 hr before washing and changing the medium.

Rat hippocampal neurons were transfected with Lipofectamine 2000 (Invitrogen) following the manufacturer's instructions or infected with lentiviral particles produced as previously described (*Lois et al., 2002*).

### BS3 crosslinking

Experiments were carried out according to *Boudreau et al., 2012*. Briefly, primary hippocampal neurons were washed twice with PBS supplemented with 0.1 mM $CaCl_2$ (Sigma-Aldrich) and 1 mM $MgCl_2$ (Sigma-Aldrich) at 37°C. Neurons were then exposed to PBS supplemented with 0.1 mM $CaCl_2$ and 1 mM $MgCl_2$ with or without the BS3 crosslinker (1 mg/ml, Thermo Fisher) at 4°C for 10 min. Neurons were then rapidly washed first with TBS supplemented with 0.1 mM $CaCl_2$ and 1 mM $MgCl_2$ plus 50 mM glycine (Sigma-Aldrich) at 4°C and subsequently with TBS supplemented with 0.1 mM $CaCl_2$ and 1 mM $MgCl_2$ at 4°C prior to lysis with BS3 buffer (50 mM Tris-HCl, 150 mM NaCl, 1 mM EDTA, pH 7.4, 1% SDS plus protease inhibitors). 3× Laemmli sample buffer was then added and samples were analysed by SDS-PAGE and western blotting. Crosslinked proteins present in the plasma membrane appeared as high molecular bands. All the other bands, which were also present in the non-crosslinked reaction, were considered as part of the intracellular pool and their intensity quantified to generate the graphs in *Figure 1A*, according to our previous results (*Moretto et al., 2019*). Extracellular and intracellular intensities were normalised on tubulin intensity.

### Vesicles purification

Hippocampi and cortices were collected from adult Wistar rats and homogenised with glass-teflon homogeniser in homogenisation buffer (0.32 M sucrose, 20 mM HEPES-NaOH, protease inhibitors,

pH 7.4). The total homogenate was centrifuged at 1000 × $g$ for 10 min at 4°C. The supernatant S1 was further centrifuged at 10,000 × $g$ for 15 min at 4°C. The resulting pellet, corresponding to crude synaptosomal fraction, was resuspended in homogenisation buffer and centrifuged again at 10,000 × $g$ for 15 min at 4°C to wash the synaptosomes. Crude synaptosomes were lysed using hypotonic shock by resuspension in $H_2O$. The resulting vesicles were layered on a 9 ml 50–1000 mM sucrose gradient (in $H_2O$) and centrifuged in an SW40Ti Beckman rotor at 65,000 × $g$ for 3 hr. After centrifugation, 10 equal fractions were collected from the top of the gradient, and protein precipitation was performed using 6% trichloroacetic acid and 0.02% deoxycholate. 3× sample buffer was then added and the samples analysed by SDS-PAGE and western blot.

## Immunoprecipitation

For immunoprecipitation experiments on hippocampi and cortices, these were dissected from adult rat brains, lysed in RIPA buffer (50 mM Tris, 150 mM NaCl, 1 mM EDTA, 1% NP40, 1% Triton X-100, pH 7.4, protease inhibitor) with a tephlon-glass homogeniser, rotated for 1 hr at 4°C and centrifuged at 10,000 × $g$ for 30 min at 4°C. Supernatants were incubated with antibodies at 4°C overnight. Protein A-agarose beads (GE Healthcare, USA) were incubated with the supernatant at 4°C for 2 hr. Beads were washed three times with RIPA buffer, resuspended in 3× Laemmli sample buffer and analysed by SDS-PAGE followed by western blotting.

For experiments on HeLa we incubated Protein G-agarose beads (GE Healthcare, USA) with antibodies at 4°C for 2 hr. Beads were washed three times with RIPA buffer. Hela lysates in RIPA were incubated with Protein G-agarose beads at 4°C for 1 hr for lysate pre-clearing. The recovered supernatant was then incubated with the antibody-conjugated beads at 4°C overnight, washed three times in RIPA buffer, and resuspended in 3× Laemmli sample buffer and analysed by SDS-PAGE followed by western blotting.

## GST pulldown

GST-fusion proteins were prepared by growing transformed BL21 *E. coli* and inducing recombinant protein expression by adding IPTG (0.5 mM final concentration) for 2 hr. Bacteria were pelleted, and the GST-fusion protein was purified employing standard procedures using glutathione Sepharose beads (Thermo Scientific).

Hippocampi and cortices dissected from adult rat brains were pooled together, lysed in RIPA buffer by homogenisation in a tephlon-glass homogeniser, rotated for 1 hr at 4°C and then centrifuged at 10,000 × $g$ for 30 min at 4°C. Supernatants were incubated with glutathione Sepharose beads for 3 hr at 4°C and then washed and resuspended in 3× sample buffer and analysed by SDS-PAGE followed by western blotting.

## Western blots

Proteins were transferred from the acrylamide gel onto the nitrocellulose membrane (0.22 μm, GE Healthcare). Membranes were incubated with the primary antibodies (α-TSPAN5, Aviva Systems Biology #AV46640, 1:500; α-transferrin receptor, Thermo Fisher Clone H68.4, 1:500; α-tubulin, Sigma-Aldrich T5168, 1:40,000; α-AP4σ, gift from Dr Margaret Robinson, 1:500; α-AP4ε, BD Biosciences 612018, 1:1000; α-GluA1, Cell Signaling #13185, 1:1000; α-GluA2/3, gift from Dr Cecilia Gotti, 1:2000; α-Stargazin, Cell Signaling #8511, 1:1000; α-EEA1, BD Transduction Laboratories Clone 14, 1:2000; α-Rab11 BD Transduction Laboratories Clone 47, 1:1500; α-Rab7, SySy 320003, 1:700; α-Vglut1, SySy 135303, 1:2000; α-GFP, MBL #598, 1:2,500; α-GluN2A, Neuromab N327/95, 1:1000; α-CD81, Santa Cruz Biotech #166029, 1:1000; α-GFP, MBL 598, 1:1000; α-HA, Cell Signaling #3724, 1:500) at room temperature for 2–3 hr or overnight at 4°C in TBS Tween-20 (0.1%), milk (5%). After washing, the blots were incubated at room temperature for 1 hr with horseradish peroxidase-conjugated α-rabbit, α-mouse, or α-rat antibodies (1: 2000) in TBS Tween-20 (0.1%), milk (5%). Immunoreactive bands on blots were visualised by enhanced chemiluminescence (Chemidoc XRS+, Bio-Rad) or standard film development.

## Immunocytochemistry

Cultured hippocampal neurons were washed in PBS supplemented with 0.1 mM $CaCl_2$ and 1 mM $MgCl_2$ and fixed in paraformaldehyde (PFA) (4%, Sigma-Aldrich)/sucrose (4%, Sigma-Aldrich) for 10 min at room temperature and incubated with primary antibodies (α-TSPAN5, Aviva System Biology

#AV46640, 1:50; α-GluA2/3, gift of Dr Cecilia Gotti, 1:500) in GDB1X solution (2×: 0.2% gelatin, 0.6% Triton X-100, 33 mM Na$_2$HPO$_4$, 0.9 M NaCl, pH 7.4) for 2 hr at room temperature.

For surface staining, antibodies (α-GluA2, Merck clone 6C4, 1:200; α-GluA1, Cell Signaling #13185, 1:150, α-myc, Sigma #M5546, 1:1000) were applied to neurons for 10 min at room temperature followed by a washing step in PBS supplemented with 0.1 mM CaCl$_2$ and 1 mM MgCl$_2$ and PFA fixation.

After three washes with high salt buffer (500 mM NaCl, 20 mM NaPO$_4^{2-}$, pH 7.4), coverslips were incubated with secondary antibodies (Alexa-conjugated: 1:400; DyLight-conjugated: 1:300) in GDB1X solution for 1 hr at room temperature.

Internalisation experiments were performed as described by *Bassani et al., 2012*. Briefly, neurons were incubated with the anti-GluA2 surface epitope antibody at 10 µg/ml in culture medium for 10 min at room temperature. Excess antibody was then removed by washing with PBS c/m. The antibody-bound receptors were then allowed to undergo internalisation for 0, 5, or 10 min in the original media at 37°C. After PFA fixation, a secondary antibody labelled with Alexa Fluor 555 was incubated in non-permeabilising condition (PBS supplemented with 10% goat serum) for 1 hr at room temperature, thus labelling receptor-antibody remained on the surface. After washing, the coverslips were incubated with a secondary antibody labelled with DyeLight-649 in permeabilising condition (GDB1X) for 1 hr at room temperature to label the internalised receptor antibody.

Coverslips were washed with high salt buffer and mounted with Mowiol (Sigma-Aldrich).

Quantification was performed as signal measured in the 649 channel (corresponding to internalised AMPARs, I$_{AMPARs}$) divided by the sum between the signal in the 649 channel and the signal in the 555 channel (corresponding to the extracellular AMPARs E$_{AMPARs}$): I$_{AMPARs}$/(I$_{AMPARs}$ + E$_{AMPARs}$).

PLA was performed according to the manufacturer's protocol (DuoLink In Situ PLA, Merck Millipore) using DNA probes-conjugated secondary antibodies and DuoLink Fluorescent Detection reagents red or far red.

Colocalisation of the PLA signal and different DsRed-Rabs in *Figure 3* was performed using the ImageJ plugin JACOP.

Fluorescence images were acquired with an LSM800 Meta confocal microscope (Carl Zeiss) and a 63× oil-immersion objective (numerical aperture 1.4) with sequential acquisition settings, at 1024×1024 pixels resolution. Images were collected as Z-stack series projections of approximately 6–10 images, each averaged four times and taken at depth intervals of 0.75 µm.

Dendritic spines were counted on all GFP-positive neuronal dendritic arbor excluding the soma and classified with NeuronStudio software (NeuronStudio) according to the following parameters: general parameters for spine identification: length >0.2 µm and <3.0 µm, max width 3.0 µm, stubby spines size >10 voxels, non-stubby spines size >5 voxels. For spine-type classifications, the following logical tests were used: if neck ratio (head/neck diameter)>1.100 then a spine was classified as thin (if also spine length/head diameter >2.5) or mushroom (if also head diameter was >0.35 µm). A spine is classified as stubby if it fails any of the precedent logical tests.

For quantification, a mask was drawn on the GFP or mCherry channel and the immunofluorescence signal for the different antibodies was quantified as mean intensity. For the analysis of surface GluA2 or GluA1 in *Figure 4*, dendritic spine regions were identified via NeuronStudio as stated above and the quantification performed only on the corresponding areas.

## FRAP-FLIP imaging of SEP-GluA2

Neurons transfected with pCl-SEP-GluA2 and either scrambled, Sh-TSPAN5, or rescue mCherry constructs were incubated for 15 min in equilibrated Tyrode's buffer (15 mM D-glucose, 108 mM NaCl, 5 mM KCl, 2 mM MgCl$_2$, 2 mM CaCl$_2$, and 25 mM HEPES-NaOH, pH 7.4) and coverslips were mounted in an open Inox chamber (Life Imaging Services). For recycling only experiments, neurons were previously incubated with 200 µg/ml cycloheximide (Life Technologies) for 2 hr with cycloheximide also present in the recording Tyrode's buffer. An LSM800 confocal microscope equipped with an environmental chamber (37°C, 5% CO$_2$) was used. A secondary dendrite from neurons positive for both mCherry and SEP signal was selected and a portion of the dendrite (ROI) was initially bleached with high 488 nm laser power (80%) and then sequentially bleached at the extremities of the ROI and imaged every 500 ms. The fluorescence intensity of SEP-GluA2 on individual dendritic spines was measured for individual time points and normalised as F$_n$−F$_0$ (ΔF)/F$_{prebleach}$. The area under the curve was measured via GraphPad Prism 8 as area below a curve fitted by regression on the average values.

For the experiment in *Figure 6E–I*, the exponential curve $\Delta F/F_{pre} = A \left( 1 - e^{-\frac{t}{\tau}} \right)$ was fitted on the first eight time points and then the values of A and $\tau$ were extrapolated from the fitted curve.

## Real-time PCR

mRNA was extracted from cultured rat hippocampal neurons using Nucleozol Reagent following the manufacturer's instructions (Macherey Nagel).

For each condition, 1.5 µg of extracted mRNA was used to synthesise cDNA using SuperScript VILO cDNA Synthesis Kit (Thermo Fisher).

The target sequences of AP4B, AP4E, and β-actin (endogenous control) were amplified from 60 ng of cDNA in the presence of SYBR Green PCR Master Mix (Applied Biosystems) using Applied Biosystems 7000 Real-Time thermocycler. Primer sequences were as follows: AP4B Fw (AGTTGCTGGGAC TTCGACAA), AP4B Rv (CCGTGGACCCCAAGTAACC), AP4E Fw (TTCTGGATGGTTTTGTGGCTG), AP4E Rv (CCAGTGAAGCCAGATGAAGAAAA), β-actin Fw (AGATGACCCAGATCATGTTTGAGA), β-actin Rev (CCTCGTAGATGGGCACAGTGT).

Each sample was run in triplicate, and results were calculated using the ΔΔCT method to allow normalisation of each sample to the internal standard and comparison with the calibrator of each experiment.

## Experiments with ARIAD constructs

Ninety min after addition of the ARIAD ligand (2 µM), anti-myc antibody (Sigma, #M5546, 1:1000) was added to the media. Neurons were directly fixed with 4% PFA, 4% sucrose, and then incubated with a secondary antibody anti-mouse Alexa Fluor 565. Images were taken with a Leica DM5000 microscope with a 40× objective. Quantification was performed with ImageJ to quantify the surface receptor mean intensity.

Intracellular transport videos were acquired on an inverted Leica microscope (DMI6000B) at the Bordeaux Imaging Center at DIV18–19. This microscope, controlled with Metamorph (Molecular Devices, Sunnyvale, CA, USA), is equipped with a confocal spinning-disk system (Yokogawa CSU-X1, laser: 491 nm, 561 nm), an EMCCD camera (Photometrics Quantem 512), a FRAP scanner (Roper Scientific, Evry, France, 561 nm), and an oil objective HCX PL Apo 100X1.4 NA. The coverslips were mounted in a Ludin chamber with 1 ml of Tyrode medium (15 mM glucose, 100 mM NaCl, 5 mM KCl, 2 mM MgCl$_2$, 2 mM CaCl$_2$, 10 mM HEPES, 247 mosm/l) with 2 µM of ARIAD ligand to release the proteins of interest from the ER, and placed at 37°C in a Life Imaging Services chamber. Videos were acquired between 30 and 60 min of incubation with the ligand using the following acquisition sequence (*Hangen et al., 2018*): 10 images are acquired (100 ms exposure), followed by the photobleaching of ~60 µm² of proximal dendrite (5 repetitions, 70% laser), followed by video acquisition (1 min at 1 Hz, 300 images, 100 ms exposure). Co-transfection with the sh-RNAs or control was confirmed by the acquisition of an image in the green channel (488 nm) prior to the video recording.

The videos were analysed by generating kymographs, thanks to the ImageJ plugin KymoToolBox (*Hangen et al., 2018*). The vesicles' pathways were traced by the deep learning software KymoButler (*Jakobs et al., 2019*). From those traces, the number of vesicles and mean speed were calculated.

## Schematic figure

The schematics in *Figures 6A, E, 7A, B and 8* were prepared using BioRender software (https://biorender.com/).

## Quantification

All statistical analyses were done with GraphPad Prism 8 software.

Two-tailed unpaired t-test was performed to assess statistical significance between two independent groups (*Figures 1A and 2D*). One-way ANOVA, followed by Newman-Kuls post hoc multiple comparison test, was used to assess statistical significance between three or more groups (*Figures 1B, 3C, D, 4A, B, D, 5B, C, D, 6D, H, I, 7C, E and F*, *Figure 5—figure supplement 1*; *Figure 5—figure supplement 2*, *Figure 6—figure supplement 1*, *Figure 7—figure supplement 1*).

Statistical details of the experiments can be found in the figure legends (exact mean values, standard errors of the mean [SEM], and n).

Western blots were repeated at least three times from three independent experiments. Imaging experiments on cultured neurons were performed on at least three independent cultures.

## Acknowledgements

We sincerely thank Skye Stuart for critical reading of the manuscript. We thank Robert Malinow for the pCI-SEP GluR2 (Addgene plasmid #24001). We thank Michisuke Yuzaki for the GST-Ct-Stargazin plasmid. We thank Margaret Robinson for the anti AP4σ antibody. We thank Cecilia Gotti for the anti-GluA2/3 antibody. We thank Richard Pagano for plasmids encoding DsRed-Rab5 WT (Addgene plasmid #13050), DsRed-Rab7 WT (Addgene plasmid #12661), and DsRed-Rab11 WT (Addgene plasmid #12679). Rab4-GFP, Rab7-GFP, and Rab11-GFP are kind gifts from Prof G Schiavo. The financial support of Fondazione Telethon, Italy (GGP17283) is gratefully acknowledged. Part of this work was supported by PRIN (Progetti di rilevante interesse nazionale – Bando 2017), 20172C9HLW, Fondazione Cariplo, Italy (2019-3438) and Fondazione Cariplo and Telethon Alliance (GJC21035).

Funding: Fondazione Telethon, Italy (GGP17283), PRIN (Progetti di ricerca di rilevante interesse nazionale – Bando 2017), 20172C9HLW Fondazione Cariplo, Italy (2019-3438), Fondazione Cariplo and Telethon Alliance, Italy (GJC21035)

## Additional information

### Funding

| Funder | Grant reference number | Author |
|---|---|---|
| Fondazione Telethon | GGP17283 | Maria Passafaro |
| Ministero dell'Università e della Ricerca | 20172C9HLW | Maria Passafaro |
| Fondazione Cariplo | 2019-3438 | Maria Passafaro |
| Fondazione Telethon-Cariplo Alliance | GJC21035 | Maria Passafaro Edoardo Moretto |
| Fondazione Telethon | GGP19181 | Lorenzo A Cingolani |

The funders had no role in study design, data collection and interpretation, or the decision to submit the work for publication.

### Author contributions

Edoardo Moretto, Conceptualization, Formal analysis, Investigation, Writing - original draft, Writing – review and editing; Federico Miozzo, Lorenzo A Cingolani, Investigation, Writing – review and editing; Anna Longatti, Francoise Coussen, Fanny Jaudon, Formal analysis, Investigation, Writing – review and editing; Caroline Bonnet, Formal analysis, Investigation; Maria Passafaro, Conceptualization, Supervision, Funding acquisition, Writing – review and editing

### Author ORCIDs

Edoardo Moretto http://orcid.org/0000-0002-3546-6797
Federico Miozzo http://orcid.org/0000-0003-0818-9525
Anna Longatti http://orcid.org/0000-0002-1636-8550
Francoise Coussen http://orcid.org/0000-0002-3194-3058
Fanny Jaudon http://orcid.org/0000-0001-7648-0977
Lorenzo A Cingolani http://orcid.org/0000-0001-9538-1659
Maria Passafaro http://orcid.org/0000-0002-0045-5676

### Ethics

Animal procedures were performed in accordance with the European Community Council Directive of November 24, 1986 (86/609/EEC) on the care and use of animals. Animal procedures were approved by the Italian Ministry of Health (Protocol Number N° 2D46AN.463).

Decision letter and Author response
Decision letter https://doi.org/10.7554/eLife.76425.sa1
Author response https://doi.org/10.7554/eLife.76425.sa2

## Additional files

### Supplementary files
• Transparent reporting form

### Data availability
All data generated during this study are included in the manuscript and supporting files. Source data files have been provided for all figures.

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
