## [Editor Report]

Glutamate receptor trafficking to synapses plays a crucial role in adjusting the efficacy of information flow in the nervous system. Here the authors show in hippocampal neurons that TSPAN5, a tetraspanin family protein facilitates the delivery of AMPA-type glutamate receptors to the cell surface by interacting with adaptor proteins, AP-4, and stargazin. The work provides evidence supporting a novel mechanism that contributes to the regulation of AMPA receptor traffic and is of interest to the molecular neuroscience and cell biology community.

---

## [Decision Letter]

**Decision letter after peer review:**

[Editors’ note: the authors submitted for reconsideration following the decision after peer review. What follows is the decision letter after the first round of review.]

Thank you for submitting your work entitled "The tetraspanin TSPAN5 regulates AMPARs exocytosis by interacting with AP-4 complex" for consideration by *eLife*. Your article has been reviewed by 3 peer reviewers, and the evaluation has been overseen by a Reviewing Editor and a Senior Editor. The reviewers have opted to remain anonymous.

Our decision has been reached after consultation between the reviewers. Based on these discussions and the individual reviews below, we regret to inform you that your work will not be considered further for publication in *eLife*.

All three reviewers agree that the intracellular regulatory mechanisms underlying the cell surface delivery of AMPAR is an important problem. However, the present study raises more questions than answers, and much further experimental work is required in support of the claimed model. We hope you find the reviewer comments constructive.

*Reviewer #1:*

Moretto et al. have recently reported a developmental role for a member of tetraspanins TSPAN5, in morphological maturation of dendritic spines. Here, in a follow up study the authors identify an additional role for TSPAN5 in trafficking of GluA2 to the cell surface in mature synapses. TSPAN5 interacts with AP-4 and also with stargazin and GluA1 and GluA2/3 that can be found in the recycling endosomes. Knocking down TSPAN5 reduces cell surface GluA2 as well as total levels of GluA2 while cell surface GluA1 is increased along with its total levels. TSPAN5 knock-down also compromises exocytosis of GluA2 to the spine surface, an effect which is lost upon blocking protein synthesis. Based on these observations, it is concluded that TSPAN5 specifically regulates exocytosis of newly synthesized GluA2 from recycling endosomes. While the findings are potentially interesting, as the manuscript stands, the conclusions are not compellingly supported by the data shown. Additional experiments will be needed in support of the model presented, and further insights into the mechanism by which the interaction of TSPAN5 with AP-4 and stargazin control the surface delivery specifically of GluA2-AMPAR will enhance the impact of the study.

1. Lines 66, 69, 149 and elsewhere. The use of the term "extracellular" is confusing. It should be referred to as the plasma membrane or the cell surface pool.

2. Figure 2C. The sucrose gradient separation analysis appears incomplete in that the information is limited to the relative efficiency in separating early, late and recycling endosomes and synaptic vesicles. The purity of the indicated fractions remain unclear. Other potential membrane sources, such as the Golgi membranes, lysosomes and autophagosomes should be assessed using suitable markers.

3. Figure 2D. The representative images are not of sufficient resolution and raise questions about the quantification shown. What is the scale bar? Also, it would be more meaningful to assess the degree of co-localization of endogenous proteins.

4. Figure 3A,B. Given that TSPAN5 knock-down and rescue have significant effects on spine density (Figure 1B), the changes in surface AMPAR distribution should be assessed by discriminating non-synaptic and synaptic AMPARs. To this end, it would be important to show the changes in GluA1 and GluA2 levels at synapses by confining the analysis to GluA1 and GluA2 puncta that are associated with presynaptic markers (and ideally also associated with a postsynaptic marker such as PSD-95 or Homer). Again, the image quality should be improved. Moreover, given that biochemical experiments (Figure 3C,D) show a change in total levels of GluA1 and GluA2 upon TSPAN5 knock-down, such an observation should also be supported by immunofluorescence labelling experiments to discriminate intracellular and cell surface pools of GluA1 and GluA2. Such experiments (synaptic/non-synaptic, intracellular/cell surface) are crucial for interpreting the single spine targeted FRAP-FLIP experiments shown in Figure 5.

5. Figure 4A. How does leupeptin treatment affect the total GluA1 and GluA2 measured biochemically? The labelling for GluA2/3 is so weak that the quantification is not convincing.

6. Figure 4B. Given that the experiment shown here monitors the fate of surface labelled GluA2 after washing of excess unbound antibodies and then incubating the neurons at 37C for 0, 5 and 10 min, it is not clear why 0 min time point does not show strong surface labelling. In fact, the 10 min time point shows much stronger surface signal, and it is not clear what the source is. Again, the signal quality needs to be improved to support the claim of impaired cell surface recycling of GluA2 upon knock-down of TSPAN5.

7. Figure 5. The design of the FRAP-FLIP protocol is not explained in sufficient detail in the methods, and it is difficult to interpret the figure. The very weak SEP-GluA2 signal poses an issue. Could the authors confirm the presence of intracellular SEP-GluA2 by applying preventing acidification? In addition, it would be helpful to include a control using a cell surface protein whose exo-endocytic recycling is not affected by TSPAN5 knock-down. No evidence is provided here that the relative reduction in GluA2 exocytosis observed upon TSPAN5 knock-down is due to the newly synthesized GluA2. If such were the case, then it is expected that for the scrambled group, the fluorescence intensity measurements representing GluA2 exocytosis should be higher in the absence of cycloheximide compared to cycloheximide treated group, but that is not observed. Moreover, there could be other translated proteins that participate in TSPAN5-dependent regulation of GluA2 traffic, whose activity is compromised by cycloheximide.

*Reviewer #2:*

Moretto, Longatti et al. found a significant fraction of TSPAN5 exists in the intracellular membrane compartment. There, it interacts directly with AP-4 sigma subunit. TSPAN5 regulates the surface amount of GluR2/3 and GluR1 differently. While TSPAN5 downregulation by shRNA reduces the surface amount of GluR2/3, the same manipulation increases GluR1. The endocytosed GluR2/3 undergo degradation pathway that can be inhibited by leupeptin. Finally, they tested if the recycled and newly synthesized receptors are affected by live imaging. They found that only newly synthesized receptor population was affected by the downregulation of TSPAN5.

Overall, this work is a bit piecemeal. I even feel that it is just a collection of the data they did not publish in their earlier work. They initially made an interesting observation that TSPAN5 interacts with AP-4. But in the rest of study, they did not study about it at all. The data have some ambiguity and often the differences are small to argue anything. They used only single treatment (for example, cycloheximide and leupeptin) to make conclusion. This is also weak.

Another serious concern is that the TSPAN5 antibody they used has multiple extra bands on western blotting on brain tissue (Figure S1 of Moretto et al., 2019). The immunostaining with such antibody cannot be trusted.

English must be edited by a native speaker. I see a lot of grammatical mistakes.

Specific comments.

Figure 1A. The authors should describe in detail how they quantified the extracellular proportion. I see an increase in high-molecular weight population in the presence of the cross-linker. However, strangely, there is no reduction in the monomer band (~35 kd). If the top band is truly cross-linked population, there should be a reduction in the monomer but this is not the case here. Also, it is not clear how they obtained the numbers in graph. Why total is less than Intra+Extra?

Line 86. SEL and LEL domain. "domain" should be plural. But "domain" is often used to refer a protein region which has solid structure but not flexible structure, such as loop between transmembrane domain. I would say "SEL and LEL" is sufficient. Besides the meaning of this sentence of not clear. What do the authors mean by "rely"? Need to rewrite.

Figure 1C. This is not useful. 3+ is not informative. What was the overall result? What genes were identified for how many clones? Among them how many were independent clones? Which domain of AP-4 sigma was identified?

Figure 1D. Add blotting with sigma subunit antibody.

Figure 1E. Something strange in this result. Generally, in a coimmunoprecipitation experiment, the immunoprecipitated protein (for which antibody is used) is much more than coimmunoprecipitated protein. But if one look at the blot, IP with TSPAN5 antibody did not recover more than TSPAN5 protein than IP with AP-4 epsilon antibody. Indeed, there is more TSPAN5 in IP with AP-4epsilon antibody. Also, in the TSPAN5 blot, do not cut the top portion of the band.

Figure 2A. In TSPAN5 blot, there is a clear band in IgG control lane. The authors put an asterisk in IP Stargazin lane. Do they mean this thin band is TSPAN5? It is hard to convince the readers.

Figure 2C. Blotting with Rab4 should be included as it was used in the next panel.

Figure 2D. I have no idea why they could see statistical significance between Rab4 and Rab7. There is only n = 6. Even it is statistically significant, I do not think the authors can make a strong argument as the average is almost the same. Also, they should test significance between Rab4 and Rab11. The authors should explain why TSPAN5 shows a significant colocalization with Rab4 comparable to Rab7 while it was not recovered in early endosome fraction in Figure 2C.

Figure 3. The authors found opposing results for GluR1 and GluR2/3. However, their finding indicates both GluR1 and GluR2/3 equally interacts with TSPAN5. Indeed, the most of GluR1 in hippocampal tissue has both GluR1/2 or GluR2/3 heterooligomer. Only ~8% is GluR1 homomer (Wenthold, 1996). This quantification include interneurons which does not express GluR2 so the amount of GluR1 homomer in pyramidal neurons is even less. The authors seem to have some explanation but merely speculation. This should be experimentally addressed. Here the involvement of AP-4 must be experimentally tested, for example, with Crispr/CAS9 or shRNA.

Figure 3C. The entire blot should be shown for GluR1 and 2/3, not just a part of it.

Figure 5. When comparing the condition with and without cycloheximide, the amount of recovery is almost the same in scrambled control. It is strange because in the absence of cycloheximide, both newly synthesized and recycled receptors are inserted at the synapse while in the presence, only the latter contributes. Also, cycloheximide stops all protein synthesis so the observed difference may or may not be due to newly synthesized GluR2/3. This should be experimentally addressed or the authors should tone down their argument.

*Reviewer #3:*

This manuscript reports the potential roles of intracellular TSPAN5 in dendritic spines upon neuronal maturation. The authors asked whether the intracellular TSPAN5 that is associated with AP-4 and recycling endosomes regulates AMPARs trafficking with different sets of experiments. In support of this, the authors used ShRNA already characterized to show that acute TSPAN5 downregulation in mature neurons affects the surface and total levels of GluA1 or GluA2. The authors also claimed that the intracellular pool of the tetraspanin TSPAN5 specifically promotes exocytosis of newly synthesized GluA2-containing AMPA receptor without affecting internalization or recycling.

Overall, this is an interesting study. However, I have some concerns which should be thoroughly addressed given that TSPAN5 involvement in dendritic spine maturation has been linked to its membrane clustering with neuroligin-1.

1) The annotation and the quantification of crosslinking experiment are not clear to me. "Intra" means "intracellular not crosslinked" and "extra" means "plasma membrane bound and crosslinked", I guess. Line 68-69, The sentence "here the vast majority of TSPAN5 is extracellular" is not correct.

Quantifications do show an increase in the intracellular pool at DIV19 that is not at all clear on the blots. The last panel in Figure 1A is not commented in the text. Can the authors discuss these points?

A negative control like another TSPAN should be used to show that not all TSPAN have the same profile.

The blots at DIV19 are strangely similar to those published in Figure S1C of Moretto et al., Cell Reports.

The sentence line 71 "This observation suggested that the intracellular pool of TSPAN5 could have a completely unrelated function" seems also misleading. Of course, the intracellular pool is different from the surface one but it can just mean that before being at the surface the TSPAN5 is in intracellular vesicles.

2) Two-hybrid results should be added in supplemental section

3) It would have been nice to illustrate the two different pools using confocal imaging in complement of the ShRNA experiment Figure 1B.

4) In Figure 1E, blots are cut short. Can the authors include a different TSPAN for IP? A negative control for Western blot should be included. I would have been nice to have a comparison with TSPAN7.

5) The interaction or complex between TSPAN5, GluA2 and Stargazin is a quite important point. Is the interaction between TSPAN5 and GluAs or between Stargazin and TSPAN? The co-immunoprecipitation of TSPAN with Stargazin is not convincing. Blots need to be ran a bit longer to separate bands. Alternatively, TSPAN5 antibodies should be used for immunoprecipitation. Can the authors test other antibodies like AMPAR ones? Again, negative controls should be included like NMDAR.

6) Figure 1D, Figure 3A and 3B and Figure 4A and 4B. I am not sure what we should see but images are not convincing. Authors should better illustrate what they are quantifying. How the analysis was done? Details need to be included in the method section

7) Why blots are cut in Figure 3C for GluA2/3 and GluA1? Intra and extra – as define – cannot be quantified in D if blots are developed the same way.

8) In the FRAP-FLIP protocol, neurons are treated with "with cycloheximide to remove newly synthesized receptor". However, in the abstract, the author mentioned that "TSPAN5 specifically promotes exocytosis of newly synthesized GluA2-containing AMPA receptor without affecting its internalization or recycling." This sounds not logical. How can the authors study newly synthetized receptors if they block their synthesis? What about lateral diffusion? Moreover, I do not see in the paper any experiment showing a specific role of TSPAN5 on newly synthesized receptors. GluA1 should be included as a control experiment since it does not behave the same way.

9) Graphs in all figures should be scatter plots or whisker-plots to let the readers see the distribution of the data. Combining column scatter plot and a box-and-whiskers plot on the same graph is a good way to display data.

10) The last paragraph of the discussion is very speculative and the authors should at least include data showing the effect of AP-4 loss on AMPAR exocytosis and trafficking process to claim that the paper "provides a possible mechanism for the intellectual disability symptoms that occur in AP-4 deficiency syndrome."

[Editors’ note: further revisions were suggested prior to acceptance, as described below.]

Thank you for resubmitting your work entitled "The tetraspanin TSPAN5 regulates AMPARs exocytosis by interacting with the AP-4 complex" for further consideration by *eLife*. Your revised article has been evaluated by Richard Aldrich (Senior Editor) and a Reviewing Editor and consultations with three reviewers. Please note that two reviewers have been newly invited.

The reviewers find the study to be potentially interesting although further extensive revisions with new data are requested:

Essential revisions

1) The precise nature of TSPAN5 interaction with AP-4, especially the binding sites and the intracellular compartment where they interact, and the requirement of the TSPAN5-AP-4 interaction for AMPAR trafficking need to be compellingly shown with additional experiments. In particular, please address the comments below:

Reviewer 1, points 2, 3;

Reviewer 2, under weaknesses – points 6-8

Reviewer 3, points 1, 2

2) The claim for the exclusive regulation of newly synthesized GluA2-AMPARs by TSPAN5 should be removed unless the authors are able to provide concrete experimental evidence. The western blot data in Figure 2A should be quantified, and along with the quantification results, if in vitro interaction experiments in point 1 above provide some insights into possible specificity for GluA2 regulation by TSPAN5, the authors are welcome to speculate in the discussion.

3) For experiments involving fluorescence imaging, image quality needs to be improved.

4) In addition to the above points (1)-(3), the individual reviewer concerns that involve data re-analysis and editing of the text should be carefully addressed.

The individual reviewer comments are appended below.

*Reviewer #1 (Recommendations for the authors):*

The revised manuscript has been substantially improved with additional control experiments, new experiments, and careful editing. In particular, the monitoring newly synthesized GluA2 using the ARIAD system is elegant. However, several issues remain that require consideration as listed below.

Specific points

P 3, lines 64-69, and discussion. What other proteins do AP-4 interact with? Is TSPAN5 a major interactor or are there others? What is the basis for claiming that AP-4 interaction with TSPAN5 is the crucial one representing the intellectual disability associated with AP-4 deficiency?

Figure 2C. Biochemical fractionation – in the Western blot, TSPAN5 level seems to be the highest in late endosomes enriched for Rab7, along with stargazing, GluA1 and GluA2/3, but clearly devoid of AP-4. This point requires an explanation.

Figure 2D. Representative images of co-localization are difficult to see, which raises questions about the quantification. Higher resolution images should be shown. (This point has been raised in the previous version.)

Figure 5. That the requirement for TSPAN5 in controlling surface levels of GluA2 via exocytosis is likely limited to newly synthesized GluA2 should be confirmed also using bulk surface labelling experiments using surface antibody labelling and/or BS3 cross linking.

P 10, lines 267-270. The logic for the statement is not clear. That a lack of change in the speed or the number of newly ER-exited GluA2 containing vesicles traversing along the dendrite upon KD of TSPAN5 suggest for a lower amount of vesicles containing newly synthesized GluA2 being targeted for exocytosis or they are directed for degradation do not seem to match with the claim that less than 20-30% of total exocytosis is due to recycling GluA2.

Figure 6G,H. To support the conclusion that in the absence of TSPAN5 newly synthesized GluA2 is rerouted for degradation, one should compare the effect of leupeptin treatment on surface vs. intracellular ARIAD-tdTomato-GluA2 with or without TSPAN5. The experiment shown here is limited to demonstrating the effect of leupeptin treatment on the steady-state levels of total GluA2/3, whose reduction in the absence of TSPAN5 is recovered by blocking proteolysis. The pool of GluA2/3 (i.e. newly synthesized GluA2/3 destined for surface delivery) being examined is not clear.

*Reviewer #2 (Recommendations for the authors):*

Strengths: The authors provide solid data for some of their conclusions.

Figure 1C: GST pulldown of TSPAN5 C-terminus provides further evidence of Y2H interaction with AP-4.

Figure 2A: Greater TSPAN5 pulldown of GluA2 vs. GluA1, but this result should be quantified as this result would improve their argument for a specificity for GluA2.

Figure 3A & 3B Knock down of TSPAN5 decreases GluA2 while increases GluA1 surface intensity via imaging. Figure 3C & 3D uses BS3 crosslinking and qualitatively obtains the same result. However, BS3 cross-linking is a non-standard approach to measure surface receptor (see Weaknesses below).

Figure 4C: The lack of additive/synergistic effect of double knockdown of TSPAN5 and AP-4 implies these molecules act in the same pathway.

Figure 5: TSPAN5 influences surface expression of *newly synthesized* GluA2.

Figure 6: Surface expression is lower in shRNA TSPAN5 using Ariad drug which releases newly synthesized GluA2 from ER (Figure 6C) but this is not due to alterations in rate of trafficking (Figures6D-F), thus the authors use leupeptin to inhibit degradation and see a (modest) change. Thus, the authors suggest that TSPAN5 may increase GluA2 expression by preventing lysosomal degradation (model).

Weaknesses: The strengths above are diminished by significant weaknesses described below. Some conclusions are not supported by experimental evidence.

Figure 1A: the authors use a cross-linking approach with a membrane impermeant cross-linker to distinguish between intracellular and surface TSPAN5. This is a non-standard method as it assumes that there is no monomeric surface TSPAN5 (which would not be subject to cross-linking). A more standard approach to distinguish intracellular from surface protein is using biotinylation studies, which relies on the same chemical properties (ie., formation of stable amide bonds through reactive primary amines). At minimum, caveats of the approach should be stated because the fraction referred to as 'intracellular' is likely an over-estimation. In addition, an explanation of the multiple bands in the immunoblots should be provided as well as which specific bands were included in the graphs shown at right.

Alternatively, this result could be eliminated as attempting to demonstrate an increase in the intracellular pool at DIV 19 does not add significantly to the impact, and in fact, the issues raised above reduce impact. For example, the authors go on to state that, "To test if the increase of intracellular TSPAN5 could be related to a different function compared to its previously described role in dendritic spines maturation (Moretto et al., 2019a)." However, there is no way to distinguish between the intracellular and extracellular pools with the knock-down approach as this would target all TSPAN5.

Figure 1B: The rescue condition appears to be over-expression as the result is above baseline.

Figure 1D: the immunoprecipitation results are not convincing as only AP-4e was pulled down by GST (Figure 1C) and it is highly abundant in the input and not enriched in the co-IP with antibodies for TSPAN5.

Lin 105-108: The statement that the C-terminal region of TSPAN5 present in intracellular vesicles is facing the cytosol is confusing as all TSPAN5 C-terminal regions would face the cytosol regardless of whether present in vesicles, the plasma membrane or along the secretory pathway.

Figure 2A: Is the pull-down through AP-4? Figure 1 implies an interaction of TSPAN5 with AP-4 thus one would expect that AP-4 is present in the pull-down.

Figure 2B: the results are not convincing as the amount pull-downed is very small (much less than 2.5% of the input) and the Ponceau staining indicates more GST-Ct Stargazin protein present compared with GST alone.

Figure 2C: Blots should be quantified to support the conclusion that TSPAN5 is enriched with recycling endosomes as the blot appears to indicate a continuous amount of protein in all of the heavier fractions that overlaps with multiple markers. Indeed, the sucrose gradient fractionation suggests that TSPAN5 could be most highly enriched with Rab7 (late endosomes), which would necessitate revising the model proposed.

Figure 4D: The graph indicates 'GluA2 intracellular/Total mean intensity' as a function of time but the Methods section indicate that primary antibodies were used to label surface receptors followed by 0, 5 or 10 min of internalizaiton followed by non-permeable labeling of secondary antibodies. Thus, it is not at all clear how this methods labels intracellular GluA2 as indicated.

Line 148: The statement that, "As expected for a transmembrane protein that also localises to the plasma membrane, TSPAN5 had a high degree of colocalisation with all three Rabs analyzed" is confusing as the Rab proteins are intracellular proteins. The results in Figure 2D also indicate a modest increase in colocalization with Rab11 compared with the other Rabs, and together with the fractionation experiment in Figure 2C.

The authors should consider eliminating the focus on an intracellular pool of TSPAN5 being the dominant mechanism as the experimental approach does not distinguish between these two pools and the evidence for an increase in the intracellular pool is weak. Identifying the binding motif in the C-terminus of TSPAN5 that binds to AP-4 would strengthen the conclusions. In addition, the study could be improved if the authors can demonstrate a direct link with the AP-4 deficiency syndrome associated with intellectual disability symptoms.

*Reviewer #3 (Recommendations for the authors):*

In this manuscript, Moretto and Longatti et al. report a new role for TSPAN5 in regulating the trafficking of AMPA receptors. Specifically, the authors claimed that TSPAN5 promotes the exocytosis of the GluA2-containing AMPA receptors through its interaction with AP-4 and stargazin. The study is novel and is of interest to the general neuroscience and cell biology community. However, I have several major concerns, many of which are related to experimental design and data quality/analysis.

• By using GST pull-down assays, the authors showed that TSPAN-5 C-tail interacts with components of the AP-4 subunits, the GluA1 and GluA2 subunits and stargazin. A previous study by the Yuzaki lab (Matsuda et al., Neuron 2008) has reported the interaction between AP-4 and GluA1 through stargazin. However, the contribution of TSPAN-5 on the complex formation is not clear. Co-immunoprecipitation experiments in the heterologous system by overexpressing components of these complex, or if possible, direct binding assays with purified proteins, should provide a better understanding of the relationship between TSPAN5, AP-4, stargazing and AMPA receptors. Importantly, does TSPAN5 knockdown uncouple AMPA receptors-stargazin from the AP-4 complex in neurons? This could be done through proximity ligation assay (PLA – between GluA2/3 with AP-4) in wild-type vs TSPAN5 knockdown neurons.

• The authors have demonstrated using various assays that TSPAN5 is required for efficient trafficking of AMPA receptors, and that TSPAN5 and AP-4 are likely to operate on the same pathway. However, it is not clear if the interaction between TSPAN5 and AP-4 is required for AMPA receptor trafficking. This can be done by performing the rescue experiment with a TSPAN5 mutant that fails to bind to AP-4 (which requires further refinement of AP-4 binding on TSPAN5), or at the very least with TSPAN5 that lacks the C-terminal tail (δ C-tail).

• I am not convinced by the data presented in Figure 3 that TSPAN5 specifically regulates GluA2-AMPARs. In Figure 3B, I can't really see any differences in the levels of surface GluA1 between wild-type and knock-down neurons. Furthermore, the quantification of GluA1 bands from the cross-linking experiments contains only 3-4 data points with large variations among groups. It will be better if the authors consider performing the ARIAD assay or the FRAP-FLIP assay (no CHX) using the GluA1 construct.

• The localisation of TSPAN-5 requires refinement. The images in Figure 3D do not really match the quantitation on the graph that shows a high level of colocalisation between TSPAN5 and endosomal markers. Importantly, where do TSPAN5 interact with AMPARs? These can be performed with PLA assay in neurons co-expressing those Rabs.

• I suggest that the authors re-analyse the FRAP/FLIP data by measuring the amplitude and the kinetics of fluorescence recovery, instead of measuring the area under the curve. For example, data shown in Figure 5G show that the extent of SEP-GluA2 recovery (amplitude) is comparable between wild-type and TSPAN5 knockdown cells, although slightly slower. Importantly, the fluorescent of SEP-GluA2 drops quickly in TSPAN5 knockdown neurons, suggesting a defect of receptor stabilisation post-exocytosis. Not simply a defect in the rate of receptor exocytosis.

• Also, the experiments performed in the presence of cycloheximide cannot rule out the potential involvement of other newly translated proteins that are required for SEP-GluA2 exocytosis. Other experiments are required to conclude that TSPAN5 is required for the trafficking of newly synthesised GluA2 in neurons.

[Editors’ note: further revisions were suggested prior to acceptance, as described below.]

Thank you for resubmitting your work entitled "The tetraspanin TSPAN5 regulates AMPARs exocytosis by interacting with the AP-4 complex" for further consideration by *eLife*. Your revised article has been evaluated by Richard Aldrich (Senior Editor) and a Reviewing Editor.

The manuscript has been improved but there are some remaining issues that need to be addressed, as outlined below:

1. As the authors acknowledge, the present findings do not directly address whether the interaction of AP-4 with Stargazin and TSPAN5 and its regulation of AMPA receptor traffic is involved in intellectual disability associated with AP-4 deficiency syndrome. The final sentence should be removed from the abstract.

2. Figure 5B, Lines 203-205: The statistical comparison between scrambled and rescue with deltaC TSPAN5 shows only a weak difference, which is quite noticeable compared to the comparison between scrambles and sh-TSPAN5 knock-down. One could argue that there is a considerable recovery of GluA2 intensity. How do sh-TSPAN5 and rescue with deltaC TSPAN5 conditions compare?

3. Figure 6, Lines 281-282. The experiments shown here do not strongly support the claim that the exocytosis of newly synthesized GluA2 receptors is regulated by TSPAN5. The authors should indicate the possibility that rather, factors that are rapidly turned over are needed to promote GluA2 exocytosis.

4. Figure 7, while the ARIAD experiment shows that trafficking of newly synthesized GluA2 is dependent on TSPAN5, the possibility that TSPAN5 also facilitates the recycling of pre-existing GluA2-containing AMPA receptors is not excluded here.

5. Combining the points raised in 3 and 4 above, the authors should tone down the claim that TSPAN5 promotes exocytosis of newly synthesized AMPA receptors or rephrase such that it may not be selective for newly synthesized AMPA receptors. This could be a general mechanism for targeting both new and recycling AMPA receptors to the plasma membrane.

---

## [Author Response]

[Editors’ note: the authors resubmitted a revised version of the paper for consideration. What follows is the authors’ response to the first round of review.]

Our decision has been reached after consultation between the reviewers. Based on these discussions and the individual reviews below, we regret to inform you that your work will not be considered further for publication in eLife.All three reviewers agree that the intracellular regulatory mechanisms underlying the cell surface delivery of AMPAR is an important problem. However, the present study raises more questions than answers, and much further experimental work is required in support of the claimed model. We hope you find the reviewer comments constructive.Reviewer #1:Moretto et al. have recently reported a developmental role for a member of tetraspanins TSPAN5, in morphological maturation of dendritic spines. Here, in a follow up study the authors identify an additional role for TSPAN5 in trafficking of GluA2 to the cell surface in mature synapses. TSPAN5 interacts with AP-4 and also with stargazin and GluA1 and GluA2/3 that can be found in the recycling endosomes. Knocking down TSPAN5 reduces cell surface GluA2 as well as total levels of GluA2 while cell surface GluA1 is increased along with its total levels. TSPAN5 knock-down also compromises exocytosis of GluA2 to the spine surface, an effect which is lost upon blocking protein synthesis. Based on these observations, it is concluded that TSPAN5 specifically regulates exocytosis of newly synthesized GluA2 from recycling endosomes. While the findings are potentially interesting, as the manuscript stands, the conclusions are not compellingly supported by the data shown. Additional experiments will be needed in support of the model presented, and further insights into the mechanism by which the interaction of TSPAN5 with AP-4 and stargazin control the surface delivery specifically of GluA2-AMPAR will enhance the impact of the study.

We thank the reviewer for the interest in our findings. As suggested, in addition to further controls for the experiments that were already presented, we have included new experiments directly demonstrating the involvement of AP-4 and TSPAN5 in mediating exocytosis of newly synthesised GluA2-containing AMPARs.

1. Lines 66, 69, 149 and elsewhere. The use of the term "extracellular" is confusing. It should be referred to as the plasma membrane or the cell surface pool.

We agree with the reviewer. This has now been changed to “plasma membrane pool” throughout the manuscript.

2. Figure 2C. The sucrose gradient separation analysis appears incomplete in that the information is limited to the relative efficiency in separating early, late and recycling endosomes and synaptic vesicles. The purity of the indicated fractions remain unclear. Other potential membrane sources, such as the Golgi membranes, lysosomes and autophagosomes should be assessed using suitable markers.

We agree with the reviewer that this experiment does not generate pure fractions, as indicated also by the partial overlap of the different markers. However, it provides an indication of the localisation of the proteins of interest. Immuno-isolation would be the only way to obtain pure populations of organelles. The experiment presented, together with the colocalisation analysis in Figure 2D, provides an indication of the possible identity of the organelle involved. A precise characterisation of this will be the focus of future research. In this work, we aimed to identify the role of TSPAN5 and AP-4 in regulating AMPAR exocytosis.

3. Figure 2D. The representative images are not of sufficient resolution and raise questions about the quantification shown. What is the scale bar? Also, it would be more meaningful to assess the degree of co-localization of endogenous proteins.

We apologise to the reviewer for the low resolution of the uploaded figure. We have provided higher resolution images for this and the other microscopy images presented in the manuscript. The scale bar is now indicated in the figure legend.

Concerning the second point, RabGTPases are notoriously difficult to stain for as endogenous proteins, at least in neurons, hence the use of overexpression with fluorescent tags is widely used and accepted in the literature (Pavlos et al., 2010).

RabGTPases are notoriously complicated to stain as endogenous proteins, at least in neurons, hence the use of overexpression with fluorescent tags is widely used and accepted in the literature.

4. Figure 3A,B. Given that TSPAN5 knock-down and rescue have significant effects on spine density (Figure 1B), the changes in surface AMPAR distribution should be assessed by discriminating non-synaptic and synaptic AMPARs. To this end, it would be important to show the changes in GluA1 and GluA2 levels at synapses by confining the analysis to GluA1 and GluA2 puncta that are associated with presynaptic markers (and ideally also associated with a postsynaptic marker such as PSD-95 or Homer). Again, the image quality should be improved. Moreover, given that biochemical experiments (Figure 3C,D) show a change in total levels of GluA1 and GluA2 upon TSPAN5 knock-down, such an observation should also be supported by immunofluorescence labelling experiments to discriminate intracellular and cell surface pools of GluA1 and GluA2. Such experiments (synaptic/non-synaptic, intracellular/cell surface) are crucial for interpreting the single spine targeted FRAP-FLIP experiments shown in Figure 5.

We apologise to the reviewer for the low quality of the figure. We have provided enlarged, higher resolution images of the dendrites. Images of the full neurons have now been moved to the supplementary figure related to Figure 3.

We thank the reviewer for the suggested experiments. To discriminate synaptic and extrasynaptic pools of AMPARs we would need to implement super resolution microscopy techniques, to which unfortunately, we do not have direct access. As an alternative, in the revised manuscript, we have now presented an additional analysis on the same images in Figure 3 restricting the analysis on dendritic spines or dendritic shafts using the GFP channel as a mask. We observed that the changes detected upon TSPAN5 knockdown are present in both compartments for both GluA2 and GluA1, suggesting that this mechanism is not specifically restricted to synapses.

The reduction in total GluA2 levels observed biochemically in BS3 crosslinking experiments has also been confirmed by our analysis on total staining presented now in figure 6G, H.

5. Figure 4A. How does leupeptin treatment affect the total GluA1 and GluA2 measured biochemically? The labelling for GluA2/3 is so weak that the quantification is not convincing.

We apologise to the reviewer for the low quality of the uploaded figure, we have provided higher resolution images in the revised manuscript. The Leupeptin experiment has now been moved to figure 6G-H.

6. Figure 4B. Given that the experiment shown here monitors the fate of surface labelled GluA2 after washing of excess unbound antibodies and then incubating the neurons at 37C for 0, 5 and 10 min, it is not clear why 0 min time point does not show strong surface labelling. In fact, the 10 min time point shows much stronger surface signal, and it is not clear what the source is. Again, the signal quality needs to be improved to support the claim of impaired cell surface recycling of GluA2 upon knock-down of TSPAN5.

We apologise to the reviewer for the low quality of the figure. Again, we have provided higher resolution images.

In these experiments, the labelling was performed at room temperature to avoid damaging the cells (as this would occur if incubated on ice). This does not completely stop internalisation, hence the signal, although minimal, is present intracellularly even at time 0. This intracellularly labelled GluA2 will undergo recycling and contribute to the extracellular signal (e.g. in time 10). This is why in these experiments the internalisation is always shown as a ratio between the intracellular and total labelled protein, to correct for these effects.

Concerning the low intensity of the surface signal at time point 0, we have provided a more representative image of the data shown.

7. Figure 5. The design of the FRAP-FLIP protocol is not explained in sufficient detail in the methods, and it is difficult to interpret the figure. The very weak SEP-GluA2 signal poses an issue. Could the authors confirm the presence of intracellular SEP-GluA2 by applying preventing acidification? In addition, it would be helpful to include a control using a cell surface protein whose exo-endocytic recycling is not affected by TSPAN5 knock-down. No evidence is provided here that the relative reduction in GluA2 exocytosis observed upon TSPAN5 knock-down is due to the newly synthesized GluA2. If such were the case, then it is expected that for the scrambled group, the fluorescence intensity measurements representing GluA2 exocytosis should be higher in the absence of cycloheximide compared to cycloheximide treated group, but that is not observed. Moreover, there could be other translated proteins that participate in TSPAN5-dependent regulation of GluA2 traffic, whose activity is compromised by cycloheximide.

We apologise for not explaining the experiments in sufficient detail and have expanded on this in the revised manuscript, also providing a schematic in Figure 5A and E.

As suggested by the reviewer, we have now provided images showing the correct disappearance of SEP signal upon application of acidic (pH 6) imaging media and reappearance upon application of 50 mM NH_4_Cl to alkalinise the intracellular compartments. This is now shown in the supplementary figure related to Figure 5.

Regarding the second point; as stated in the methods section, the fluorescence recovery plots of the experiments are presented as ∆Fluorescence (Fn – F postbleach)/ F(prebleach) to normalise the overexpression levels which are inherently different among cells. In addition, the two experiments (now presented in Figure 5B-D and 5F-H) were performed separately and the SEP-GluA2 signal was acquired at different laser powers and as such, the intensities in the two experiments cannot be directly compared in their values.

Concerning the last point, we thank the reviewer for pointing this out. To more directly address the possible role of TSPAN5 in regulating exocytosis of newly synthesised AMPA receptor, we performed experiments (presented now in Figure 6A-C) taking advantage of the ARIAD system (Rivera et al., 2000; Hangen et al., 2018). In this system, GluA2 is expressed fused to a conditional aggregation domain (CAD) that induces aggregation and trapping of GluA2 in the endoplasmic reticulum, which is the site of AMPAR synthesis. This pool can be released by application of an ariad ligand and exocytosis of this newly synthesised GluA2 can be observed by surface staining with an anti-myc antibody (myc tag inserted in the extracellular portion of GluA2). As shown in Figure 6A-C, this experiment confirms that TSPAN5 regulates exocytosis of newly synthesised GluA2.

We agree with the reviewer that blocking protein synthesis could remove other proteins involved in TSPAN5-mediated AMPAR recycling, we have now pointed this out in the manuscript. However, although possible, this is very unlikely because either the protein in question would need to be completely (or almost) degraded in the timeframe of the experiment (120 min since addition of cycloheximide) and only around 6% of all proteins have a half-life of less than 90 min (Chen et al., 2016), or such protein would need to be newly synthesised to exert its function.

Even if we were in such situation, this does not affect the main conclusion of the paper, that TSPAN5 regulates exocytosis of newly synthesised receptors. According to previous literature (e.g. Passafaro et al., 2001), in the timeframe of the experiment in Figure 5F-H (5 min), new synthesis and recycling accounts for a recovery of 15% of the total surface levels, whereas recycling alone is only between 0% (at time point 0min) and 5.7% (at time point 10 minutes). As such, even if we account for the maximum level (5.7%) and if AMPAR recycling was completely inhibited in absence of TSPAN5 (and this cannot be the case as we still observed some degree of recycling in the experiment in Figure 5A-E, roughly 5%), this could only account for a reduction of 30% (5%/15%), whereas here we observe a reduction of approximately 50% (Figure 5H).

As such, we believe our data demonstrate that TSPAN5 regulates the exocytosis of newly synthesised receptors.

Reviewer #2:Moretto, Longatti et al. found a significant fraction of TSPAN5 exists in the intracellular membrane compartment. There, it interacts directly with AP-4 sigma subunit. TSPAN5 regulates the surface amount of GluR2/3 and GluR1 differently. While TSPAN5 downregulation by shRNA reduces the surface amount of GluR2/3, the same manipulation increases GluR1. The endocytosed GluR2/3 undergo degradation pathway that can be inhibited by leupeptin. Finally, they tested if the recycled and newly synthesized receptors are affected by live imaging. They found that only newly synthesized receptor population was affected by the downregulation of TSPAN5.Overall, this work is a bit piecemeal. I even feel that it is just a collection of the data they did not publish in their earlier work. They initially made an interesting observation that TSPAN5 interacts with AP-4. But in the rest of study, they did not study about it at all. The data have some ambiguity and often the differences are small to argue anything. They used only single treatment (for example, cycloheximide and leupeptin) to make conclusion. This is also weak.

In the revised version of the manuscript, in addition to more control experiments, we have now provided new experiments that reinforce the evidence that TSPAN5 and AP-4 participate in regulating the exocytosis of newly synthesised GluA2-containing AMPARs.

Another serious concern is that the TSPAN5 antibody they used has multiple extra bands on western blotting on brain tissue (Figure S1 of Moretto et al., 2019). The immunostaining with such antibody cannot be trusted.

The localisation of tetraspanins in membranes and their complex interaction web produces different patterns of bands depending on the lysis condition. In our previous paper we have shown that upon lysis in RIPA buffer, only one band is detected in brain tissue which is completely lost in mice knockout for TSPAN5 (Figure S1A, Moretto et al., 2019). Upon lysis with buffers including SDS, tetraspanins are retained in oligomeric forms (Figure S1D and F, Moretto et al., 2019). These bands are all reduced in intensity by an ShRNA against TSPAN5 in neuronal cultures. In addition, the ShRNA against TSPAN5 reduces the signal of the anti TSPAN5 antibody also in immunostaining (Figure S1H, Moretto et al., 2019).

The localisation of tetraspanins in membranes and their complex interaction web produces different patterns of bands depending on the lysis condition. In our previous paper, we have shown that upon lysis in RIPA buffer, only one band is detected in brain tissue, which is completely lost in TSPAN5 knockout mice (Figure S1A, Moretto et al., 2019). Upon lysis with buffers including SDS, such as in the experiment in Figure 1A, tetraspanins are retained in oligomeric form (Figure S1D and F, Moretto et al., 2019). These bands are all reduced in intensity by an ShRNA against TSPAN5 in neuronal cultures (Figure S1D, Moretto et al., 2019), the efficiency of which was also confirmed via RT-PCR (Figure S1G, Moretto et al., 2019). In addition, the ShRNA against TSPAN5 reduces the signal of the anti-TSPAN5 antibody in immunostaining (Figure S1H, Moretto et al., 2019).

We believe this is sufficient evidence to demonstrate the specificity of the antibody.

English must be edited by a native speaker. I see a lot of grammatical mistakes.

We apologise to the reviewer for this. We have had the manuscript edited by a professional editing service.

Specific comments.Figure 1A. The authors should describe in detail how they quantified the extracellular proportion. I see an increase in high-molecular weight population in the presence of the cross-linker. However, strangely, there is no reduction in the monomer band (~35 kd). If the top band is truly cross-linked population, there should be a reduction in the monomer but this is not the case here. Also, it is not clear how they obtained the numbers in graph. Why total is less than Intra+Extra?

We apologise to the reviewer for not clearly explaining how these experiments were quantified; we have also noticed a mistake in the quantification of these experiments.

All bands detected by the anti-TSPAN5 antibody are specific (see previous point above). Hence, they will all have to be considered; in the data we initially presented, we had only quantified the monomer, we have now re-quantified including all bands in the analysis.

The data are presented as intensity of TSPAN5 (total: all bands in the non-crosslinked lane normalised on tubulin; extracellular: intensity of the high molecular weight band that appears only in the BS3 crosslinked lane, normalised to tubulin; intracellular: intensity of all the bands except the high molecular weight crosslinked one, normalised to tubulin). This is the best way to analyse BS3 crosslinking experiments as the affinity of the antibody might be different between the crosslinked protein and the monomeric one (Boudreau et al., 2012). As such, only the different fractions can be compared between themselves (e.g. intracellular at DIV12 vs intracellular at DIV19) and it is not possible to infer what proportion of the protein is actually extracellular or intracellular. This is also likely to be the reason why the reduction in the non-crosslinked bands is not directly visible in the experiment.

Line 86. SEL and LEL domain. "domain" should be plural. But "domain" is often used to refer a protein region which has solid structure but not flexible structure, such as loop between transmembrane domain. I would say "SEL and LEL" is sufficient. Besides the meaning of this sentence of not clear. What do the authors mean by "rely"? Need to rewrite.

We apologise to the reviewer for this. We have rephrased the sentence as follows: “Given that TSPAN5 in the intracellular pool would have its SEL and LEL facing the lumen of vesicles, we hypothesised that this fraction of TSPAN5 could participate in intracellular trafficking through interactions of its cytosol-exposed C-terminus as previously shown for other tetraspanins”.

Figure 1C. This is not useful. 3+ is not informative. What was the overall result? What genes were identified for how many clones? Among them how many were independent clones? Which domain of AP-4 sigma was identified?

We apologise to the reviewers for this. We have provided the list of proteins in the supplementary figure related to Figure 1 and specified the region of AP-4 sigma identified, which corresponded to the first 102 amino acids.

Figure 1D. Add blotting with sigma subunit antibody.

Unfortunately, AP-4 sigma runs very close to the GST protein making it impossible to identify a clear signal in this experiment. However, AP-4 is an obligate tetramer and the presence of a AP-4 epsilon demonstrates that the whole complex is co-precipitated (Hirst et al., 2013).

Figure 1E. Something strange in this result. Generally, in a coimmunoprecipitation experiment, the immunoprecipitated protein (for which antibody is used) is much more than coimmunoprecipitated protein. But if one look at the blot, IP with TSPAN5 antibody did not recover more than TSPAN5 protein than IP with AP-4 epsilon antibody. Indeed, there is more TSPAN5 in IP with AP-4epsilon antibody. Also, in the TSPAN5 blot, do not cut the top portion of the band.

Immunoprecipitation experiments rely on the affinity of antibodies which is inherently different between different antibodies. The anti-AP-4 epsilon antibody looks more efficient in immunoprecipitating AP-4 than the anti-TSPAN5 antibody appears to be in immunoprecipitating TSPAN5, as we used the same amount of antibody (2µg). A comparison of the band intensities in immunoprecipitation experiments performed with different antibodies is therefore not possible.

Figure 2A. In TSPAN5 blot, there is a clear band in IgG control lane. The authors put an asterisk in IP Stargazin lane. Do they mean this thin band is TSPAN5? It is hard to convince the readers.

Unfortunately, the anti-TSPAN5 antibody detects a high signal arising from both the heavy and light chain of the anti-Stargazin antibody. To circumvent this issue, we have now provided a new, cleaner experiment where we used GST fused to the C-terminal tail of Stargazin, which was identified to interact with AP-4 (Matsuda et al., 2008). Using this GST fusion protein, we were able to show precipitation of TSPAN5 together with AMPAR subunit GluA2/3.

Figure 2C. Blotting with Rab4 should be included as it was used in the next panel.

We apologise to the reviewer for this, we have been unable to find a reliable anti-Rab4 antibody to perform this experiment. However, EEA1, which is presented in Figure 2C, is a reliable and routinely used marker of early endosomes.

Figure 2D. I have no idea why they could see statistical significance between Rab4 and Rab7. There is only n = 6. Even it is statistically significant, I do not think the authors can make a strong argument as the average is almost the same. Also, they should test significance between Rab4 and Rab11.

We apologise to the reviewer, we made a mistake in this graph. The significant difference is between Rab4 and Rab11 and there is no statistically significant difference between Rab4 and Rab7, this has been corrected in the revised manuscript.

The authors should explain why TSPAN5 shows a significant colocalization with Rab4 comparable to Rab7 while it was not recovered in early endosome fraction in Figure 2C.

Some partial overlap between EEA1 and TSPAN5 can be seen in lanes 4 and 5 of the fractionation experiment shown in Figure 2C. The high level of colocalisation observed in the imaging experiment could potentially be explained by the overexpression of Rabs, which has been previously shown to enhance their activity thus reducing the maturation of organelles through the endolysosomal pathway (Furusawa et al., 2018). This was necessary as endogenous RabGTPases are notoriously difficult to stain for, at least in neurons, hence the use of overexpression with fluorescent tags is widely used and accepted in the literature (Pavlos et al., 2010).

As such, we believe that the imaging experiment in figure 2D should mostly be considered a comparison between conditions rather than in its absolute values and the results considered together with the fractionation experiment presented in Figure 2C.

Figure 3. The authors found opposing results for GluR1 and GluR2/3. However, their finding indicates both GluR1 and GluR2/3 equally interacts with TSPAN5. Indeed, the most of GluR1 in hippocampal tissue has both GluR1/2 or GluR2/3 heterooligomer. Only ~8% is GluR1 homomer (Wenthold, 1996). This quantification include interneurons which does not express GluR2 so the amount of GluR1 homomer in pyramidal neurons is even less. The authors seem to have some explanation but merely speculation. This should be experimentally addressed.

Although we agree with the reviewer about the importance of AMPA receptor subunit composition, our experiments do not provide any evidence regarding the composition of different oligomers. With our experiments, both in staining, western blot and FRAP-FLIP we are detecting either GluA1 or GluA2 or GluA2/3 as individual proteins. They will then mainly be present in GluA1/2 or GluA2/3 heteromers with a smaller component of GluA1 homomers; however, our experiments cannot resolve this. This is a common approach in the AMPA receptor field mainly because only quite complicated experiments can address the actual abundance of the different oligomers given the presence of GluA2 in both GluA1/2 or GluA2/3.

It is likely that in our experiments, given the reduction of GluA2 and the increase in GluA1, GluA2/3 are the heteromers mainly reduced and it is possible that the remaining GluA2 is redirected to GluA1/2 heteromers and that we will have an increase of GluA1 homomers. We will expand the discussion on this part.

Here the involvement of AP-4 must be experimentally tested, for example, with Crispr/CAS9 or shRNA.

As suggested by the reviewer, we have generated CRISPR/Cas9 constructs to knockdown the expression of AP-4 and analysed the levels of surface GluA2 in this setting, identifying a reduction comparable to that observed upon knockdown of TSPAN5. In addition, the simultaneous knockdown of both TSPAN5 and AP-4 did not lead to a further reduction in GluA2 plasma membrane levels, which strongly supports the participation of these two proteins in the same pathway. This is shown in Figure 4C of the revised manuscript.

Figure 3C. The entire blot should be shown for GluR1 and 2/3, not just a part of it.

We have now added the full blots for GluA2/3 and GluA1 in the supplementary figure related to Figure 3 of the revised manuscript.

Figure 5. When comparing the condition with and without cycloheximide, the amount of recovery is almost the same in scrambled control. It is strange because in the absence of cycloheximide, both newly synthesized and recycled receptors are inserted at the synapse while in the presence, only the latter contributes.

As stated in the methods section, the fluorescence recovery plots of the experiments are presented as ∆Fluorescence (Fn – F postbleach)/ F (prebleach) to normalise the overexpression levels which are inherently different among cells. This would also normalise the levels to prebleach conditions in the presence of cycloheximide (the treatment was carried out for two hours before imaging). In addition, the two experiments (now shown in Figure 5A-D and 5E-H) were performed separately and acquired with different laser powers. Hence the two experiments cannot be directly compared in their absolute values.

Also, cycloheximide stops all protein synthesis so the observed difference may or may not be due to newly synthesized GluR2/3. This should be experimentally addressed or the authors should tone down their argument.

We thank the reviewer for pointing this out.

To more directly address the possible role of TSPAN5 in regulating exocytosis of newly synthesised AMPA receptors, we performed experiments (presented now in Figure 6A-C) taking advantage of the ARIAD system (Hangen et al., 2018; Rivera et al., 2000). In this system, GluA2 is expressed fused to a conditional aggregation domain (CAD) that induces aggregation and traps GluA2 in the endoplasmic reticulum, the site of AMPAR synthesis. This pool can be released by application of an ariad ligand and exocytosis of this newly synthesised GluA2 can be observed by surface staining with an anti-myc antibody (myc tag inserted in the extracellular portion of GluA2). As shown in Figure 6C, this experiment confirms that TSPAN5 regulates exocytosis of newly synthesised GluA2.

We agree with the reviewer that blocking protein synthesis could remove other proteins involved in TSPAN5-mediated AMPAR recycling, we have now pointed this out in the manuscript. However, although possible, this is very unlikely because either the protein in question would need to be completely (or almost) degraded in the timeframe of the experiment (120 min since addition of cycloheximide) and only around 6% of all proteins have a half-life of less than 90 min (Chen et al., 2016), or such protein would need to be newly synthesised to exert its function.

Even if we were in one of these situations, this would not affect the main conclusion of the paper, that TSPAN5 regulates exocytosis of newly synthesised receptors. According to previous literature (e.g. Passafaro et al., 2001), in the timeframe of the experiment in Figure 5F-H (5 min), new synthesis and recycling accounts for a recovery of 15% of the total surface levels, whereas recycling alone is only between 0% (at time point 0min) and 5.7% (at time point 10 minutes). As such, even if we account for the maximum level (5.7%) and if AMPAR recycling was completely inhibited in absence of TSPAN5 (and this cannot be the case as we still observed some degree of recycling in the experiment in Figure 5A-E, roughly 5%), this could only account for a reduction of 30% (5%/15%), whereas here we observe a reduction of approximately 50% (Figure 5H).

Reviewer #3:This manuscript reports the potential roles of intracellular TSPAN5 in dendritic spines upon neuronal maturation. The authors asked whether the intracellular TSPAN5 that is associated with AP-4 and recycling endosomes regulates AMPARs trafficking with different sets of experiments. In support of this, the authors used ShRNA already characterized to show that acute TSPAN5 downregulation in mature neurons affects the surface and total levels of GluA1 or GluA2. The authors also claimed that the intracellular pool of the tetraspanin TSPAN5 specifically promotes exocytosis of newly synthesized GluA2-containing AMPA receptor without affecting internalization or recycling.Overall, this is an interesting study. However, I have some concerns which should be thoroughly addressed given that TSPAN5 involvement in dendritic spine maturation has been linked to its membrane clustering with neuroligin-1.1) The annotation and the quantification of crosslinking experiment are not clear to me. "Intra" means "intracellular not crosslinked" and "extra" means "plasma membrane bound and crosslinked", I guess. Line 68-69, The sentence "here the vast majority of TSPAN5 is extracellular" is not correct.

We apologise to the reviewer for the confusion, their interpretation is correct. We have explained the rationale of BS3 crosslinking experiments in the main text. Concerning the quantification, the data are presented as intensity of TSPAN5 (total: all bands in the non-crosslinked lane normalised on tubulin; extracellular: intensity of the high molecular weight band that appears only in the BS3 crosslinked lane, normalised on tubulin; intracellular: intensity of all the bands except the high molecular weight crosslinked one, normalised on tubulin). This is the best way to analyse BS3 crosslinking experiments as the affinity of the antibody might be different between the crosslinked protein and the non-crosslinked one (Boudreau et al., 2012). As such, only the different fractions can be compared between themselves (e.g. intracellular at DIV12 vs intracellular at DIV19) and it is not possible to infer what proportion of the protein is actually extracellular or intracellular.

For this reason, we agree with the reviewer that the sentence in line 68-69 is incorrect as we cannot compare extracellular and intracellular pools. We have removed it from the revised manuscript.

Quantifications do show an increase in the intracellular pool at DIV19 that is not at all clear on the blots.

We apologise to the reviewer; we have now provided a more representative image. In addition, we have noticed a mistake in the quantification of these experiments.

All bands detected by the anti-TSPAN5 antibody are specific (as seen in (Moretto et al., 2019)). Hence, they will all have to be considered in the quantification. In the data we presented we had only quantified the monomer. This new analysis makes no difference to the final results and is now provided in the revised version of the manuscript. In addition, we have changed the blot for TSPAN5 in Figure 1A for one that is more representative of the new quantification.

The last panel in Figure 1A is not commented in the text. Can the authors discuss these points?

We apologise for not mentioning the Transferrin receptor panels in the text. Transferrin is routinely used as a control cell surface protein for receptor trafficking experiments in neurons. Here, it is used as a control to show that not all plasma membrane proteins follow the same pattern of expression and localisation upon neuronal maturation.

A negative control like another TSPAN should be used to show that not all TSPAN have the same profile.

We apologize for not mentioning the panels about Transferrin receptor in the text. Transferrin is routinely used as a control cell surface protein for receptor trafficking experiments in neurons. It is here used as a control to show that not all plasma membrane proteins follow the same pattern of expression and localization upon neuronal maturation.

The blots at DIV19 are strangely similar to those published in Figure S1C of Moretto et al., Cell Reports.

We have now provided a different and more representative blot for this figure in the revised manuscript. The blots at DIV19 in Figure 1A were exactly the same blots from Figure S1C of Moretto et al., 2019, and we had reported the re-use of this in the submission process.

The sentence line 71 "This observation suggested that the intracellular pool of TSPAN5 could have a completely unrelated function" seems also misleading. Of course, the intracellular pool is different from the surface one but it can just mean that before being at the surface the TSPAN5 is in intracellular vesicles.

We have rephrased the sentence as follows: “We observed an increase in the intracellular levels of TSPAN5 from DIV12 to DIV19, which was not followed by a concomitant increase in plasma membrane levels (Figure 1A), suggesting that increased intracellular levels of TSPAN5 does not necessarily imply increased delivery of this protein to the plasma membrane. The transferrin receptor showed a more stable distribution across these time points”.

If the intracellular pool was only representing TSPAN5 trafficking towards the plasma membrane, we should have observed an increase between DIV12 and DIV19 in both the intracellular and extracellular fraction. Of course, other explanations are possible, for example the intracellular pool could be increased in order to be released in response to synaptic plasticity events, as it is the case for AMPA receptors (Passafaro et al., 2001). However, this prompted us to investigate if TSPAN5 could have another intracellular function, which we identified and described in the following experiments.

2) Two-hybrid results should be added in supplemental section

We have added the list of genes identified with the two-hybrid screen in Supplementary Figure 1. In the main text, we have also specified that the AP-4 sigma clones identified were coding for the first 102 amino acids of the protein.

3) It would have been nice to illustrate the two different pools using confocal imaging in complement of the ShRNA experiment Figure 1B.

We agree with the reviewer; unfortunately, the TSPAN5 antibody does not work for surface staining experiments, and therefore the suggested experiment cannot be performed.

4) In Figure 1E, blots are cut short. Can the authors include a different TSPAN for IP? A negative control for Western blot should be included. I would have been nice to have a comparison with TSPAN7.

We apologise to the reviewer for this, we have now provided a larger image of the blots in Figure 1D. In addition, we have inserted negative controls in the GST pulldown experiments in Figure 2A, B showing that another tetraspanin, CD81, is not precipitated by the C-terminal tail of Stargazin and that the NMDAR subunit 2A is not precipitated by the C-terminal tail of TSPAN5.

5) The interaction or complex between TSPAN5, GluA2 and Stargazin is a quite important point. Is the interaction between TSPAN5 and GluAs or between Stargazin and TSPAN? The co-immunoprecipitation of TSPAN with Stargazin is not convincing. Blots need to be ran a bit longer to separate bands. Alternatively, TSPAN5 antibodies should be used for immunoprecipitation. Can the authors test other antibodies like AMPAR ones? Again, negative controls should be included like NMDAR.

Unfortunately, the anti-TSPAN5 antibody detects a high signal arising from both the heavy and light chain of the anti-Stargazin antibody. To circumvent this issue, we have now provided a new, cleaner experiment in Figure 2B, where we used GST fused to the C-terminal tail of Stargazin, which was identified to interact with AP-4 (Matsuda et al., 2008). Using this GST fusion protein, we were able to show precipitation of TSPAN5 together with AMPAR subunit GluA2/3, but not of the tetraspanin CD81.

In addition, we now show in Figure 2A that the GST pulldown using the C-terminal tail of TSPAN5 does not precipitate the NMDAR subunit 2A.

In general, experiments such as immunoprecipitation or GST-pull down cannot exactly assess direct interactions, but rather demonstrate association in a protein complex. As such, we cannot define the specific interactions. However, the interaction between TSPAN5 and AP-4 is likely to be direct, as we identified it in yeast two-hybrid. The interaction between AP-4 and GluAs is mediated by Stargazin and does not occur without it (Matsuda et al., 2008).

6) Figure 1D, Figure 3A and 3B and Figure 4A and 4B. I am not sure what we should see but images are not convincing. Authors should better illustrate what they are quantifying. How the analysis was done? Details need to be included in the method section

We apologise for the resolution of these images. We have provided higher resolution images in the revised version of the manuscript. The quantifications in Figure 3A, B and 4A, B were done by measuring the mean intensity of the signals in the GFP positive areas. We have added this in the revised manuscript methods section. In addition, now in figure 3A, B we have also analysed the images as suggested by Reviewer #1, restricting the analysis to dendritic spines or dendritic shafts, as isolated using the GFP channel and the NeuronStudio software.

7) Why blots are cut in Figure 3C for GluA2/3 and GluA1? Intra and extra – as define – cannot be quantified in D if blots are developed the same way.

We apologise for not explaining this properly. We have now provided the full blots in the supplementary figure related to figure 3.

Similarly to the experiment in Figure 1A, the quantification and comparison is only made between the same pool between conditions (extracellular in Scrambled vs extracellular in Sh-TSPAN5, both normalised on tubulin) to avoid confusion coming from different affinities of the antibody for crosslinked versus noncrosslinked proteins (Boudreau et al., 2012).

8) In the FRAP-FLIP protocol, neurons are treated with "with cycloheximide to remove newly synthesized receptor". However, in the abstract, the author mentioned that "TSPAN5 specifically promotes exocytosis of newly synthesized GluA2-containing AMPA receptor without affecting its internalization or recycling." This sounds not logical. How can the authors study newly synthetized receptors if they block their synthesis? What about lateral diffusion?

We apologise to the reviewer for not being clear. The rationale behind the experiments in Figure 5 can now be found in the schematic in Figure 5A and 5E. During the project, we hypothesised that TSPAN5 and AP-4 could actually participate in the recycling of AMPARs given the possible localisation in Rab11 positive organelles. As such, we performed the experiment with cycloheximide (Figure 5A-D), to restrict the analysis to recycling AMPARs but we did not observe any difference. We then repeated the FRAP-FLIP experiment without cycloheximide (Figure 5E-H) to include newly synthesised receptors and detected a significant difference. Hence, we hypothesised that the reduction of surface GluA2 in TSPAN5 knockdown is caused by defects in the exocytosis of newly synthesised receptors and not by an effect on receptor recycling. This has been further confirmed by the new experiment in Figure 6A-C (see point below).

Lateral diffusion was excluded in these experiments by using continuous bleaching at the edges of the ROI, as stated in the methods section. This is also explained in the schematics in Figure 5A and 5E.

Moreover, I do not see in the paper any experiment showing a specific role of TSPAN5 on newly synthesized receptors. GluA1 should be included as a control experiment since it does not behave the same way.

We thank the reviewer for pointing this out. To more directly address the possible role of TSPAN5 in regulating exocytosis of newly synthesised AMPA receptors we performed experiments (presented now in Figure 6A-C) taking advantage of the ARIAD system (Hangen et al., 2018; Rivera et al., 2000). In this system, GluA2 is expressed fused to a conditional aggregation domain (CAD) that induces aggregation and trapping of GluA2 in the endoplasmic reticulum, the site of synthesis for most AMPARs. This pool can be released by application of an ariad ligand and exocytosis of this newly synthesised GluA2 can be observed by surface staining with an anti-myc antibody (myc tag inserted in the extracellular portion of GluA2). As shown in Figure 6A-C, this experiment confirms that TSPAN5 regulates exocytosis of newly synthesised GluA2.

9) Graphs in all figures should be scatter plots or whisker-plots to let the readers see the distribution of the data. Combining column scatter plot and a box-and-whiskers plot on the same graph is a good way to display data.

We agree with the reviewer, we have replaced all the bar graphs with column scatter plots showing the distribution of data with the exception of the graphs in Figure 5D and H, as the Area under the curve was calculated as area under the curve fitted on the average of all replicates.

10) The last paragraph of the discussion is very speculative and the authors should at least include data showing the effect of AP-4 loss on AMPAR exocytosis and trafficking process to claim that the paper "provides a possible mechanism for the intellectual disability symptoms that occur in AP-4 deficiency syndrome."

In the revised manuscript, we have generated CRISPR/Cas9 constructs to knockdown the expression of AP-4 and analysed the levels of surface GluA2 in this setting, identifying a reduction comparable to that observed upon knockdown of TSPAN5. In addition, the simultaneous knockdown of both TSPAN5 and AP4 did not lead to a further reduction in GluA2 plasma membrane levels, which strongly supports the participation of these two proteins in the same pathway. This is shown in Figure 4C of the revised manuscript.

References

Boudreau, A. C., Milovanovic, M., Conrad, K. L., Nelson, C., Ferrario, C. R., & Wolf, M. E. (2012). A protein cross-linking assay for measuring cell surface expression of glutamate receptor subunits in the rodent brain after in vivo treatments. Current Protocols in Neuroscience, 1(SUPPL.59). https://doi.org/10.1002/0471142301.ns0530s59

Chen, W., Smeekens, J. M. & Wu, R. (2016). Systematic study of the dynamic and half-lives of newly synthesized proteins in human cells. Chem. Sci. (7), 1393-1400 https://doi.org/10.1039/C5SC03826J

Furusawa, K., Takasugi, T., Chiu, Y.W., Hori, Y., Tomita, T., Fukuda, M., and Hisanaga, S. (2019). CD2associated protein (CD2AP) overexpression accelerates amyloid precursor protein (APP) transfer from early endosomes to the lysosomal degradation pathway. Journal of biological chemistry. 294(28) 10886 –10899. https://doi.org/10.1074/jbc.RA118.005385

Hangen, E., Cordelières, F. P., Petersen, J. D., Choquet, D., & Coussen, F. (2018). Neuronal Activity and Intracellular Calcium Levels Regulate Intracellular Transport of Newly Synthesized AMPAR. Cell Reports, 24(4). https://doi.org/10.1016/j.celrep.2018.06.095

Hirst, J., Irving, C., & Borner, G. H. H. (2013). Adaptor Protein Complexes AP-4 and AP-5: New Players in Endosomal Trafficking and Progressive Spastic Paraplegia. Traffic, 14(2). https://doi.org/10.1111/tra.12028

Matsuda, S., Miura, E., Matsuda, K., Kakegawa, W., Kohda, K., Watanabe, M., & Yuzaki, M. (2008). Accumulation of AMPA Receptors in Autophagosomes in Neuronal Axons Lacking Adaptor Protein AP-4. Neuron. https://doi.org/10.1016/j.neuron.2008.02.012

Moretto, E., Longatti, A., Murru, L., Chamma, I., Sessa, A., Zapata, J., Hosy, E., Sainlos, M., Saint-Pol, J., Rubinstein, E., Choquet, D., Broccoli, V., Schiavo, G., Thoumine, O., & Passafaro, M. (2019). TSPAN5 Enriched Microdomains Provide a Platform for Dendritic Spine Maturation through Neuroligin-1 Clustering. Cell Reports. https://doi.org/10.1016/j.celrep.2019.09.051

Passafaro, M., Piãch, V., & Sheng, M. (2001). Subunit-specific temporal and spatial patterns of AMPA receptor exocytosis in hippocampal neurons. Nature Neuroscience, 4(9), 917–926. https://doi.org/10.1038/nn0901-917

Pavlos, N. J., Gronborg, M., Riedel, D., Chua, J. J. E., Boyken, J., Kloepper, T. H., Urlaub, H., Rizzoli, S. O., & Jahn, R. (2010). Quantitative Analysis of Synaptic Vesicle Rabs Uncovers Distinct Yet Overlapping Roles for Rab3a and Rab27b in ca^2+^-Triggered Exocytosis. Journal of Neuroscience, 30(40). https://doi.org/10.1523/JNEUROSCI.0907-10.2010

Rivera, V. M., Wang, X., Wardwell, S., Courage, N. L., Volchuk, A., Keenan, T., Holt, D. A., Gilman, M., Orci, L., Cerasoli, F., Rothman, J. E., & Clackson, T. (2000). Regulation of Protein Secretion Through Controlled Aggregation in the Endoplasmic Reticulum. Science, 287(5454). https://doi.org/10.1126/science.287.5454.826

Shi, S.-H., Hayashi, Y., Esteban, J. A., & Malinow, R. (2001). Subunit-Specific Rules Governing AMPA Receptor Trafficking to Synapses in Hippocampal Pyramidal Neurons. Cell, 105, 331–343. https://doi.org/10.1016/S0092-8674(01)00321-X

Wu, L., Shi, Y., Jackson, A. C., Bjorgan, K., During, M. J., Sprengel, R., Seeburg, P. H. and Nicoll, R. A. (2009). Subunit composition of synaptic AMPA receptors revealed by a single-cell genetic approach. Neuron 62(2):254-68.

[Editors’ note: what follows is the authors’ response to the second round of review.]

Reviewer #1 (Recommendations for the authors):The revised manuscript has been substantially improved with additional control experiments, new experiments, and careful editing. In particular, the monitoring newly synthesized GluA2 using the ARIAD system is elegant. However, several issues remain that require consideration as listed below.Specific pointsP 3, lines 64-69, and discussion. What other proteins do AP-4 interact with? Is TSPAN5 a major interactor or are there others? What is the basis for claiming that AP-4 interaction with TSPAN5 is the crucial one representing the intellectual disability associated with AP-4 deficiency?

As suggested by the reviewer, we have included information about other interactors of AP-4 identified in neurons in the Introduction (lines 61-65). The paragraph reads as follows: “AP-4 was previously found to regulate the intracellular trafficking and sorting of several transmembrane proteins in neurons including the stargazin-AMPA receptors complex (26), the glutamate receptor δ2 (27), the autophagy regulator ATG9 (28), and DAGLB, an enzyme involved in the production of the endocannabinoid 2-AG (29).”. We do not claim that this interaction is crucial for the intellectual disability phenotype associated with AP-4 deficiency. We just point out that AMPA receptor levels are crucial for synaptic transmission and as such for neuronal functioning and that defects in AMPA receptor levels have been associated with other pathologies characterised by intellectual disability. This suggests that this pathway might be involved in the pathology but we do not demonstrate it in this work.

Figure 2C. Biochemical fractionation – in the Western blot, TSPAN5 level seems to be the highest in late endosomes enriched for Rab7, along with stargazing, GluA1 and GluA2/3, but clearly devoid of AP-4. This point requires an explanation.

As suggested by the reviewer, we have quantified the intensity of the bands of the different proteins analysed, now presented in Figure 3B of the revised manuscript.

It is true that there is a significant pool of TSPAN5, Stargazin and AMPA-Rs in the fractions with the highest signal for Rab7; this is likely the pool of these proteins that is being directed towards degradation, as normally occurring for transmembrane proteins (Vanlandingham and Ceresa, 2009). The absence of AP-4 in this fraction suggests that the complex under study in this work is not present in late endosomes.

A significant pool of TSPAN5, Stargazin and AMPA-Rs is present also in the heaviest fractions 8-10, positive for TfR, where also AP4 is visible. This, together with the new data showing TSPAN5-GluA2 PLA colocalisation being the highest with Rab11, presented in Figure 3D of the revised manuscript, strongly points at the TSPAN5-AP-4-Stargazin-AMPA-Rs complex being present in this compartment.

Figure 2D. Representative images of co-localization are difficult to see, which raises questions about the quantification. Higher resolution images should be shown. (This point has been raised in the previous version.)

We apologise to the reviewer for this. We have changed the images, now in Figure 3C, with images more representative of the quantification and shown with higher resolution (the figure was at 300ppi in the previous version, now at 600ppi). Colocalising puncta are now highlighted with white arrowheads.

Figure 5. That the requirement for TSPAN5 in controlling surface levels of GluA2 via exocytosis is likely limited to newly synthesized GluA2 should be confirmed also using bulk surface labelling experiments using surface antibody labelling and/or BS3 cross linking.

We thank the reviewer for this comment. Studying trafficking of endogenously newly synthesised AMPA-Rs would require very complicated experiments.

Although based on over expression systems, we believe that our experiments in Figures 6 and 7 clearly demonstrate that lowering TSPAN5 levels induces a reduction in the exocytosis of newly synthesised GluA2 AMPA-Rs. In particular, the experiment in figure 7 is based on surface labelling of newly synthetised AMPA receptors and clearly demonstrates that knockdown of TSPAN5 reduces their levels. Overexpressed AMPARs have been extensively used in the field to study their trafficking properties.

P 10, lines 267-270. The logic for the statement is not clear. That a lack of change in the speed or the number of newly ER-exited GluA2 containing vesicles traversing along the dendrite upon KD of TSPAN5 suggest for a lower amount of vesicles containing newly synthesized GluA2 being targeted for exocytosis or they are directed for degradation do not seem to match with the claim that less than 20-30% of total exocytosis is due to recycling GluA2.

We apologise to the reviewer for not being clear in this point. In the experiment presented in Figure 7A-C, we observed a reduction in the surface level of newly synthesised GluA2 but did not detect a change in the speed of transport of the vesicles containing GluA2 nor a change in the number of vesicles travelling. In particular, the reduction in GluA2 surface levels but a normal number of GluA2 positive vesicles could be explained in two ways: (1) these vesicles contain a lower amount of GluA2 (i.e. amount of GluA2 in each vesicle) which we were unable to measure in this experiment or (2) a portion of these vesicles reaches dendritic spines but fails to deliver their content to the plasma membrane. The sentence has been changed as follows: (lines 296-299) “These results suggest that there is either a lower amount of GluA2 loaded into each of these vesicles directed for exocytosis or that these vesicles can reach their destination but fail to deliver their content to the plasma membrane of dendrites and might be directed for degradation as a result.”

Figure 6G,H. To support the conclusion that in the absence of TSPAN5 newly synthesized GluA2 is rerouted for degradation, one should compare the effect of leupeptin treatment on surface vs. intracellular ARIAD-tdTomato-GluA2 with or without TSPAN5. The experiment shown here is limited to demonstrating the effect of leupeptin treatment on the steady-state levels of total GluA2/3, whose reduction in the absence of TSPAN5 is recovered by blocking proteolysis. The pool of GluA2/3 (i.e. newly synthesized GluA2/3 destined for surface delivery) being examined is not clear.

We agree with the reviewer on this point and have downplayed the conclusions of this experiment. The paragraph now reads as follows: (lines 300-307) “To test this second possibility, we assessed the total levels of GluA2/3 via immunofluorescence in DIV20 neurons transfected at DIV12 with either Scrambled or ShTSPAN5 and treated with the lysosomal inhibitor leupeptin (Figure 7D, E), since AMPARs are mostly degraded via this pathway (41). Leupeptin treatment increased GluA2/3 to similar levels in Scrambled and Sh-TSPAN5 transfected neurons, suggesting that AMPARs degradation is increased in the absence of TSPAN5 (Figure 7D-E).

However, this experiment does not directly demonstrate that newly synthesised GluA2 are rerouted towards degradation.”. We believe this does not diminish the importance of our findings and instead open up new studies to understand what is the fate of AMPA-Rs in the absence of TSPAN5 and/or AP-4.

Reviewer #2 (Recommendations for the authors):Strengths: The authors provide solid data for some of their conclusions.Figure 1C: GST pulldown of TSPAN5 C-terminus provides further evidence of Y2H interaction with AP-4.Figure 2A: Greater TSPAN5 pulldown of GluA2 vs. GluA1, but this result should be quantified as this result would improve their argument for a specificity for GluA2.Figure 3A & 3B Knock down of TSPAN5 decreases GluA2 while increases GluA1 surface intensity via imaging. Figure 3C & 3D uses BS3 crosslinking and qualitatively obtains the same result. However, BS3 cross-linking is a non-standard approach to measure surface receptor (see Weaknesses below).Figure 4C: The lack of additive/synergistic effect of double knockdown of TSPAN5 and AP-4 implies these molecules act in the same pathway.Figure 5: TSPAN5 influences surface expression of *newly synthesized* GluA2.Figure 6: Surface expression is lower in shRNA TSPAN5 using Ariad drug which releases newly synthesized GluA2 from ER (Figure 6C) but this is not due to alterations in rate of trafficking (Figures6D-F), thus the authors use leupeptin to inhibit degradation and see a (modest) change. Thus, the authors suggest that TSPAN5 may increase GluA2 expression by preventing lysosomal degradation (model).Weaknesses: The strengths above are diminished by significant weaknesses described below. Some conclusions are not supported by experimental evidence.Figure 1A: the authors use a cross-linking approach with a membrane impermeant cross-linker to distinguish between intracellular and surface TSPAN5. This is a non-standard method as it assumes that there is no monomeric surface TSPAN5 (which would not be subject to cross-linking). A more standard approach to distinguish intracellular from surface protein is using biotinylation studies, which relies on the same chemical properties (ie., formation of stable amide bonds through reactive primary amines). At minimum, caveats of the approach should be stated because the fraction referred to as 'intracellular' is likely an over-estimation. In addition, an explanation of the multiple bands in the immunoblots should be provided as well as which specific bands were included in the graphs shown at right.

We agree that BS3 would fail to crosslink monomeric TSPAN5 that is not engaging in any interaction even if present on the surface. However, we hypothesise that the level of this pool is likely to be extremely low, considering that tetraspanins exert their function by homo and heterotypic interactions in the so called Tetraspanin Enriched Microdomains. We have mentioned this limitation in the main text as follows: (lines 89-93) “It needs to be mentioned that it is possible that a fraction of TSPAN5 present on the plasma membrane does not interact with any other protein. This fraction would not be crosslinked and run as a monomer. However, this eventuality is quite unlikely, especially considering that the main function of tetraspanins is exerted by homo and heterotypic interactions (Charrin et al., 2014).” Thus, we believe that our claim about an increase in the intracellular pool could still be valid.

In addition, as suggested by the reviewer, we have explained that we considered all bands that appear in the western blot as we previously demonstrated that they are all specific as they are decreased by the ShRNA against TSPAN5 (Moretto et al., 2019, supplementary Figure 1F). We have explained this in the main text which reads as follows: (lines 81-85) “As shown in Figure 1A, TSPAN5 appears as a complex pattern of bands. This is probably due to the association of this protein with cholesterol rich membranes which makes it poorly soluble in standard lysis buffers (3). We previously demonstrated that all these bands are specific (11) and thus they were all included in the quantification.”

Alternatively, this result could be eliminated as attempting to demonstrate an increase in the intracellular pool at DIV 19 does not add significantly to the impact, and in fact, the issues raised above reduce impact. For example, the authors go on to state that, "To test if the increase of intracellular TSPAN5 could be related to a different function compared to its previously described role in dendritic spines maturation (Moretto et al., 2019a)." However, there is no way to distinguish between the intracellular and extracellular pools with the knock-down approach as this would target all TSPAN5.

Although the point made by the reviewer about BS3 cannot be ruled out by our experiments, we find it extremely unlikely. We agree that this result is not crucial for the impact of the paper but it genuinely represent our thinking process during the development of the project.

This increase in the intracellular pool between DIV12 and DIV19 might suggest that the function of intracellular TSPAN5 is likely more important in more mature neurons, whereas the “extracellular” function of TSPAN5 on dendritic spine maturation that we previously described (Moretto et al., 2019) is crucial at younger stages as these are when most of synaptogenesis occurs in culture (Chanda et al., 2017).

Considering this point, we designed the strategy followed throughout the manuscript, which consisted in silencing TSPAN5 (and/or AP4) at DIV13 to reduce the protein levels when synaptogenesis is well underway, to reduce the interference of the “extracellular” synaptogenic function of TSPAN5 and better isolate the other function described in this manuscript. It is true that knockdown would reduce levels of TSPAN5 both intracellularly and extracellularly, but we are showing that TSPAN5 knockdown at DIV13 minimally impact synaptogenesis.

Although these experiments do not incontrovertibly demonstrate that the increase in TSPAN5 levels is due to the increase in the intracellular pool, they strongly suggest it, and we think are interesting and worth sharing with the scientific community, highlighting the rightful caveats raised by the reviewer (see point above).

Figure 1B: The rescue condition appears to be over-expression as the result is above baseline.

It is true that in many of our experiments, the rescue condition seems to induce a potentiation of TSPAN5 function. We believe that the importance of the rescue experiments in this work is to show the specificity of the function isolated with ShRNA-based knockdown and exclude off-target effects of this approach. This is demonstrated by the restoration of the defects induced by ShRNA expression by concomitant expression of an ShRNA-resistant form of TSPAN5.

The potentiation of TSPAN5 function is potentially linked to the fact that TSPAN5 expression in the Rescue condition is under the control of strong constitutive promoters and is not controlled by the endogenous TSPAN5 promoter and regulatory elements thus resulting in a relative overexpression.

Figure 1D: the immunoprecipitation results are not convincing as only AP-4e was pulled down by GST (Figure 1C) and it is highly abundant in the input and not enriched in the co-IP with antibodies for TSPAN5.

We respectfully disagree with this point. We only explored the presence of AP-4ε in the pulldown experiment (now shown in Figure 2A) because this is the only subunit for which good commercially available antibodies exist.

Regarding the Co-IP (now in Figure 2B), this experiment is used to demonstrate that an association exists between the analysed proteins. Using Co-Ips to judge the strength of an interaction is very complicated. Although we used the same amount (in µg) of the different antibodies for precipitation, these will have different affinities for their target proteins. In addition, each antibody might have different affinity for the protein in its native folded configuration compared to the denatured form of the protein that is detected in the western blot. As such we believe that, although the levels of AP-4 epsilon appear to be low in the TSPAN5 immunoprecipitate compared to the input, this experiment shows only that TSPAN5, AP4 epsilon and AP4 sigma (here detected with an homemade antibody from J Hirst, which was not available at the time of the pulldown experiment) are present in the same macromolecular complex. The interaction is specific as none of the proteins is present in the IgG control.

Line 105-108: The statement that the C-terminal region of TSPAN5 present in intracellular vesicles is facing the cytosol is confusing as all TSPAN5 C-terminal regions would face the cytosol regardless of whether present in vesicles, the plasma membrane or along the secretory pathway.

We agree with the reviewer that this sentence was confusing. We have changed it as follows: (lines 110113) “The only portions of TSPAN5 exposed to the cytosol are the N and C termini (1). The C-terminus of other tetraspanins have been shown to regulate the intracellular trafficking of other proteins (12). We thus decided to perform a yeast two-hybrid screen using the C-terminal tail of TSPAN5 as bait.”

Figure 2A: Is the pull-down through AP-4? Figure 1 implies an interaction of TSPAN5 with AP-4 thus one would expect that AP-4 is present in the pull-down.

The interaction shown here (now figure 2C) is indeed mediated by AP-4, as demonstrated by the reduction in PLA signal between TSPAN5 and GluA2 upon AP-4 knockdown (shown in Figure 5D). We just did not think it was necessary to show AP-4 in this blot as we show it in Figure 2A in the same experimental paradigm (pulldown with GST-C Terminus of TSPAN5).

Figure 2B: the results are not convincing as the amount pull-downed is very small (much less than 2.5% of the input) and the Ponceau staining indicates more GST-Ct Stargazin protein present compared with GST alone.

We respectfully disagree with the reviewer. In this figure (now Figure 2E), the Ponceau shows one band in the empty GST and two major bands in the GST-Ct-Stargazin. The lower band in the GST-Ct-Stargazin is likely to be GST that has lost the CT-Stargazin given that it runs at the same molecular weight as the band in the empty GST lane. This is quite common in this type of experiments (for example, a similar phenomenon can be seen in the Ponceau in Figure 2A).

Hence, only the upper of the two bands in the GST-Ct-Stargazin lane would be pulling down interactors of Stargazin. The intensity of this upper band is comparable to that in the empty GST lane. In addition, although it is true that the amount of proteins pulled down is lower than the input, we do not think this hamper our conclusions. The experiment shows that these proteins can associate and that the interactions are specific, which is demonstrated by the higher signal compared to the empty GST and the absence of pulled down CD81. In addition, GluA2/3, an extremely well characterised interactor of stargazin, is precipitated in a very similar way, with a level lower in the pulldown compared to the input, and a weak signal in the empty GST control.

Figure 2C: Blots should be quantified to support the conclusion that TSPAN5 is enriched with recycling endosomes as the blot appears to indicate a continuous amount of protein in all of the heavier fractions that overlaps with multiple markers. Indeed, the sucrose gradient fractionation suggests that TSPAN5 could be most highly enriched with Rab7 (late endosomes), which would necessitate revising the model proposed.

As suggested by the reviewer, we have quantified the intensity of the bands of the different proteins analysed, now presented in Figure 3B of the revised manuscript.

It is true that there is a significant pool of TSPAN5, Stargazin and AMPA-Rs in the fractions positive for Rab7; this is likely the pool of these proteins that is being directed towards degradation, as normally occurring for transmembrane proteins (Vanlandingham and Ceresa, 2009). The absence of AP-4 in this fraction suggests that the complex under study in this work is not present in late endosomes. Of course, this does not exclude a possible separate function for this pool, but this is not the focus of the present study.

A significant pool of TSPAN5, Stargazin and AMPA-Rs is present also in the heaviest fractions 8-10, positive for TfR, where also AP4 is visible. This, together with the new data showing TSPAN5-GluA2 PLA colocalisation being the highest with Rab11, presented in Figure 3D of the revised manuscript, strongly points to the TSPAN5-AP-4-Stargazin-AMPA-Rs complex being present in this compartment.

Following the reviewer’s comment, we have however removed the claim that the proteins are enriched in TfR-positive fractions stating only their concomitant presence.

Figure 4D: The graph indicates 'GluA2 intracellular/Total mean intensity' as a function of time but the Methods section indicate that primary antibodies were used to label surface receptors followed by 0, 5 or 10 min of internalizaiton followed by non-permeable labeling of secondary antibodies. Thus, it is not at all clear how this methods labels intracellular GluA2 as indicated.

We apologise with the reviewer for not explaining this clearly. This has been now better explained in the method section as follows: (lines 545-564) “Internalisation experiments were performed as described by Bassani and colleagues (12). Briefly, neurons were incubated with the anti-GluA2 surface epitope antibody at 10 μg/ml in culture medium for 10 min at room temperature. Excess antibody was then removed by washing with PBS c/m. The antibody-bound receptors were then allowed to undergo internalisation for 0, 5 or 10 min in the original media at 37°C. After paraformaldehyde fixation, a secondary antibody labelled with AlexaFluor 555 was incubated in non-permeabilising condition (PBS supplemented with 10% goat serum) for 1 h at room temperature, thus labeling receptor-antibody remained on the surface. After washing, the coverslips were incubated with a secondary antibody labelled with DyeLight-649 in permeabilising condition (GDB1X containing 0.3% Triton X-100) for 1 h at room temperature to label the internalized receptor-antibody.

Coverslips were washed with high salt buffer and mounted with Mowiol (Sigma Aldrich).

Quantification was performed as signal measured in the 649 channel (corresponding to internalised AMPA receptors, I _AMPARs_) divided by the sum between the signal in the 649 channel and the signal in the 555 channel (corresponding to the extracellular AMPA receptors E AMPARs): I _AMPARs_/ (I _AMPARs_ + E _AMPARs_).”

Line 148: The statement that, "As expected for a transmembrane protein that also localises to the plasma membrane, TSPAN5 had a high degree of colocalisation with all three Rabs analyzed" is confusing as the Rab proteins are intracellular proteins. The results in Figure 2D also indicate a modest increase in colocalization with Rab11 compared with the other Rabs, and together with the fractionation experiment in Figure 2C.

We agree with the reviewer that this sentence was not clear. We have changed it as follows: (lines 159-161) “TSPAN5 showed a high level of colocalisation with all three Rabs. This is not surprising as TSPAN5 is likely to be transported in the endolysosomal pathway, similarly to many other transmembrane proteins that can localise in the plasma membrane.” To strengthen the conclusion of TSPAN5 colocalisation with recycling endosomes we have quantified the levels of all the proteins analysed in the fractionation experiment, now shown in figure 3A,B which shows high level of TSPAN5 in the TfR positive fraction, and have included a Proximity ligation assay (PLA) showing that the association between TSPAN5 and GluA2 mostly takes place in Rab11 positive organelles, shown in Figure 3D of the revised manuscript.

Reviewer #3 (Recommendations for the authors):In this manuscript, Moretto and Longatti et al. report a new role for TSPAN5 in regulating the trafficking of AMPA receptors. Specifically, the authors claimed that TSPAN5 promotes the exocytosis of the GluA2-containing AMPA receptors through its interaction with AP-4 and stargazin. The study is novel and is of interest to the general neuroscience and cell biology community. However, I have several major concerns, many of which are related to experimental design and data quality/analysis.• By using GST pull-down assays, the authors showed that TSPAN-5 C-tail interacts with components of the AP-4 subunits, the GluA1 and GluA2 subunits and stargazin. A previous study by the Yuzaki lab (Matsuda et al., Neuron 2008) has reported the interaction between AP-4 and GluA1 through stargazin. However, the contribution of TSPAN-5 on the complex formation is not clear. Co-immunoprecipitation experiments in the heterologous system by overexpressing components of these complex, or if possible, direct binding assays with purified proteins, should provide a better understanding of the relationship between TSPAN5, AP-4, stargazing and AMPA receptors. Importantly, does TSPAN5 knockdown uncouple AMPA receptors-stargazin from the AP-4 complex in neurons? This could be done through proximity ligation assay (PLA – between GluA2/3 with AP-4) in wild-type vs TSPAN5 knockdown neurons.

We thank the reviewer for raising this point. Indeed our reasoning is that TSPAN5 binds AP-4 sigma, as identified by yeast two hybrid. As shown in Matsuda et al., 2008, AP-4 can in turn interact with Stargazin, via its cargo binding subunit AP-4µ and through this interaction form a complex with AMPA-Rs. As such, the complex would be formed by a linear interaction TSPAN5-AP4 sigma -AP4µ-Stargazin-AMPARs.

As suggested, to clarify the organisation of this interaction complex, we have performed PLA experiments in hippocampal neurons identifying interaction between TSPAN5 and GluA2 (Figure 5D of the revised manuscript). Removing AP-4 by CRISPR/Cas9 knockdown of AP-4β or ε dramatically reduced the interaction between TSPAN5 and GluA2, as it would be expected if AP-4 was required for the interaction (Figure 5D).

We did not perform PLA between AP-4 and GluA2, as suggested, because the formation of the complex AP-4-Stargazin-AMPA-Rs was shown to occur in heterologous systems with little to no expression of TSPAN5 (Matsuda et al., 2008, Figure 7A-C). As such TSPAN5 should not participate in the formation of the complex but rather allow the loading of this complex to the right organelle for its delivery to the plasma membrane. This has now also been included in the main text (line 145) and in the Discussion paragraph.

In addition, we showed the existence of this complex by co-immunoprecipitation experiments in Hela cells, transfected with TSPAN5-GFP, Stargazin-HA and GluA2, whereas AP-4 is endogenously expressed in these cells (Figure 2 —figure supplement 1).

• The authors have demonstrated using various assays that TSPAN5 is required for efficient trafficking of AMPA receptors, and that TSPAN5 and AP-4 are likely to operate on the same pathway. However, it is not clear if the interaction between TSPAN5 and AP-4 is required for AMPA receptor trafficking. This can be done by performing the rescue experiment with a TSPAN5 mutant that fails to bind to AP-4 (which requires further refinement of AP-4 binding on TSPAN5), or at the very least with TSPAN5 that lacks the C-terminal tail (δ C-tail).

We thank the reviewer for the suggestion. We have performed the suggested experiment by analysing the surface levels of GluA2 in neurons transfected with a construct carrying both the ShTSPAN5 and the ShRNAresistant cDNA of the human TSPAN5 without the C-terminus (Rescue ΔC), as this is the region we identified to be binding to AP-4. The data are presented in Figure 5A, B of the revised manuscript. TSPAN5 lacking the C-terminus was unable to rescue the levels of surface GluA2 upon endogenous TSPAN5 knockdown, demonstrating that TSPAN5-AP-4 interaction is necessary for the trafficking of GluA2 to the plasma membrane.

• I am not convinced by the data presented in Figure 3 that TSPAN5 specifically regulates GluA2-AMPARs. In Figure 3B, I can't really see any differences in the levels of surface GluA1 between wild-type and knock-down neurons. Furthermore, the quantification of GluA1 bands from the cross-linking experiments contains only 3-4 data points with large variations among groups. It will be better if the authors consider performing the ARIAD assay or the FRAP-FLIP assay (no CHX) using the GluA1 construct.

We thank the reviewer for this point. We have replaced the image of surface levels of GluA1 as assessed by immunocytochemistry with panels more representative of the quantification, and these are presented in Figure 4B and Figure 4 —figure supplement 1, panel B. In addition, although the variation is large in the western blot data, the differences reach statistical significance and are in agreement with our imaging experiments.

We have also followed the reviewer’s suggestion and performed the experiment using the ARIAD system on GluA1 to assess exocytosis of newly synthesised receptor upon modulation of TSPAN5. This experiment is now presented in Figure 7F of the revised manuscript. We found that reducing TSPAN5 levels induced lower levels of newly synthetised surface GluA1, an effect rescued by the expression of an ShRNA-resistant TSPAN5. This confirms that the role of TSPAN5 is not restricted to the exocytosis of newly synthesised GluA2 but also of newly synthesised GluA1. This is also a consequence of the fact that the interaction between TSPAN5 and GluA is mediated by Stargazin, which can interact with both GluA1 and GluA2. The increased steady state levels of surface GluA1, shown in Figure 4, is likely a compensatory mechanism which could be due to increased expression, reduced internalisation or reduced degradation.

• The localisation of TSPAN-5 requires refinement. The images in Figure 3D do not really match the quantitation on the graph that shows a high level of colocalisation between TSPAN5 and endosomal markers. Importantly, where do TSPAN5 interact with AMPARs? These can be performed with PLA assay in neurons co-expressing those Rabs.

We have changed the images showing colocalisation of TSPAN5 with the endosomal markers, now shown in Figure 3C of the revised manuscript, with images more representative of the quantification and also to provide increased resolution. Colocalising puncta are now highlighted with white arrowheads in the images.

In addition, as suggested by the reviewer we have performed PLA with antibodies against TSPAN5 and GluA2 and analysed the colocalisation of the PLA signal with endosomal markers. This experiment is shown in Figure 3D of the revised manuscript. The PLA signal shows very high level of colocalisation with Rab11 and much less so with Rab5 and Rab7, confirming that the TSPAN5-GluA2 association takes place mostly in Rab11 positive organelles.

• I suggest that the authors re-analyse the FRAP/FLIP data by measuring the amplitude and the kinetics of fluorescence recovery, instead of measuring the area under the curve. For example, data shown in Figure 5G show that the extent of SEP-GluA2 recovery (amplitude) is comparable between wild-type and TSPAN5 knockdown cells, although slightly slower. Importantly, the fluorescent of SEP-GluA2 drops quickly in TSPAN5 knockdown neurons, suggesting a defect of receptor stabilisation post-exocytosis. Not simply a defect in the rate of receptor exocytosis.

We thank the reviewer for this suggestion. By taking advantage of the previously used equation describing the recovery of fluorescence of SEP-GluA2 (Hildick et al., 2012), we fitted our data onto the equation ΔF/Fpre=A(1−etr) and extrapolated the values of A, corresponding to the steady state ΔF/Fpre, and τ, which represents a time constant related to the kinetic of exocytosis. These are now presented in Figure 6I of the revised manuscript. Both parameters are affected by silencing TSPAN5 demonstrating both reduced kinetics and steady state levels.

Although it is true that the drop in SEP GluA2 signal shown in Figure 6G could be different between conditions, this would be influenced also by the bleaching of the SEP fluorescence over continuous imaging. We thus decided to exclude the decay phase from the analysis.

• Also, the experiments performed in the presence of cycloheximide cannot rule out the potential involvement of other newly translated proteins that are required for SEP-GluA2 exocytosis. Other experiments are required to conclude that TSPAN5 is required for the trafficking of newly synthesised GluA2 in neurons.

Following the suggestion of one of the reviewer in the previous round of revision, which had raised a very similar point, with which we agree, we had included the experiment presented in figure 7A-F of the revised manuscript. In this experiment, we evaluate only the signal arising from newly synthesised GluA2, thanks to the ARIAD system. Here myc-tagged newly synthetised GluA2 is trapped in the ER, and can be released for secretion by application of the ARIAD ligand. We then measured the levels of this GluA2 in the plasma membrane by surface staining for myc. This experiment shows that knockdown of TSPAN5 reduces the levels of this pool of GluA2 in the surface, thus demonstrating its role in the exocytosis of newly synthesised receptors.

References

Charrin S, Jouannet S, Boucheix C, Rubinstein E. Tetraspanins at a glance. J Cell Sci. 2014 Sep 1;127(Pt 17):3641-8. doi: 10.1242/jcs.154906. Epub 2014 Aug 15. PMID: 25128561.

Chanda S, Hale WD, Zhang B, Wernig M, Südhof TC. Unique versus Redundant Functions of Neuroligin Genes in Shaping Excitatory and Inhibitory Synapse Properties. J Neurosci. 2017 Jul 19;37(29):6816-6836. doi: 10.1523/JNEUROSCI.0125-17.2017. Epub 2017 Jun 12. PMID: 28607166; PMCID: PMC5518416.

Hildick KL, González-González IM, Jaskolski F, Henley JM. Lateral diffusion and exocytosis of membrane proteins in cultured neurons assessed using fluorescence recovery and fluorescence-loss photobleaching. J Vis Exp. 2012 Feb 29;(60):3747. doi: 10.3791/3747. PMID: 22395448; PMCID: PMC3315441.

Matsuda S, Miura E, Matsuda K, Kakegawa W, Kohda K, Watanabe M, Yuzaki M. Accumulation of AMPA receptors in autophagosomes in neuronal axons lacking adaptor protein AP-4. Neuron. 2008 Mar 13;57(5):730-45. doi: 10.1016/j.neuron.2008.02.012. PMID: 18341993.

Moretto E, Longatti A, Murru L, Chamma I, Sessa A, Zapata J, et al. TSPAN5 Enriched Microdomains Provide a Platform for Dendritic Spine Maturation through Neuroligin-1 Clustering. Cell Rep. 2019 Oct 29;29(5):1130–1146.e8.

Vanlandingham PA, Ceresa BP. Rab7 regulates late endocytic trafficking downstream of multivesicular body biogenesis and cargo sequestration. J Biol Chem. 2009 May 1;284(18):12110-24. doi: 10.1074/jbc.M809277200. Epub 2009 Mar 5. PMID: 19265192; PMCID: PMC2673280.

[Editors’ note: what follows is the authors’ response to the third round of review.]

The manuscript has been improved but there are some remaining issues that need to be addressed, as outlined below:1. As the authors acknowledge, the present findings do not directly address whether the interaction of AP-4 with Stargazin and TSPAN5 and its regulation of AMPA receptor traffic is involved in intellectual disability associated with AP-4 deficiency syndrome. The final sentence should be removed from the abstract.

According to the editors’ suggestion, we have removed this sentence from the abstract.

2. Figure 5B, Lines 203-205: The statistical comparison between scrambled and rescue with deltaC TSPAN5 shows only a weak difference, which is quite noticeable compared to the comparison between scrambles and sh-TSPAN5 knock-down. One could argue that there is a considerable recovery of GluA2 intensity. How do sh-TSPAN5 and rescue with deltaC TSPAN5 conditions compare?

According to the editors’ suggestion we have performed a statistical analysis between the Sh-TSPAN5 and Rescue deltaC TSPAN5. No statistically significant difference is detectable between these conditions either in the OneWay ANOVA (already shown) or in a Student T test (p = 0.0669).

3. Figure 6, Lines 281-282. The experiments shown here do not strongly support the claim that the exocytosis of newly synthesized GluA2 receptors is regulated by TSPAN5. The authors should indicate the possibility that rather, factors that are rapidly turned over are needed to promote GluA2 exocytosis.

We agree with the editors on this point. We have added the following sentence in line 275-278 at the end of the section discussing Figure 6 experiments: “However, our experiments do not exclude the possibility that TSPAN5 could also regulate the recycling of GluA2-containing AMPA-Rs, an effect that would be masked by the application of cycloheximide in the experiment presented in Figure 6A-D which could cause the loss of rapidly turning over factors needed for this process.”

4. Figure 7, while the ARIAD experiment shows that trafficking of newly synthesized GluA2 is dependent on TSPAN5, the possibility that TSPAN5 also facilitates the recycling of pre-existing GluA2-containing AMPA receptors is not excluded here.

We thank the editors for raising this point. We have added the following sentence in line 314-315 at the end of the section discussing Figure 7 experiments: “It is important to note that these experiments still do not exclude a possible regulation of TSPAN5 on recycling AMPA-Rs.“

5. Combining the points raised in 3 and 4 above, the authors should tone down the claim that TSPAN5 promotes exocytosis of newly synthesized AMPA receptors or rephrase such that it may not be selective for newly synthesized AMPA receptors. This could be a general mechanism for targeting both new and recycling AMPA receptors to the plasma membrane.

We thank the reviewer for this comment. In addition to the sentences included in the points above, we have added the following sentence in the Discussion section (lines 332-334): “In addition, our experiments cannot fully exclude that TSPAN5 could also regulate the recycling of AMPA-Rs. It thus possible that TSPAN5 could modulate the delivery to the plasma membrane of both newly synthesised and recycling AMPA-Rs.”

We have also removed “newly synthesised” from both the impact statement and the abstract (line 25), in the chapter title (line 233), in the discussion (line 342) and in the figure 6 title (line 956) to make more clear that AMPA-Rs recycling could also be regulated by TSPAN5.